environmental science

multi-isotope analysis, provenancing, diet, ancestry estimation, Tudor England, late medieval/early modern

**Author for correspondence:**
Richard Madgwick
e-mail: madgwickrd3@cardiff.ac.uk

# Diversity aboard a Tudor warship: investigating the origins of the *Mary Rose* crew using multi-isotope analysis

Jessica Scorrer[1], Katie E. Faillace[1], Alexzandra Hildred[2], Alexandra J. Nederbragt[3], Morten B. Andersen[3], Marc-Alban Millet[3], Angela L. Lamb[4] and Richard Madgwick[1]

[1]School of History, Archaeology and Religion, Cardiff University, Cardiff CF10 3EU, UK
[2]Mary Rose Trust, HM Naval Base, Portsmouth PO1 3LX, UK
[3]School of Earth and Environmental Sciences, Cardiff University, Cardiff CF10 3AT, UK
[4]National Environmental Isotope Facility, British Geological Survey, Keyworth, Nottinghamshire NG12 5GG, UK

JS, 0000-0002-4047-1507; KEF, 0000-0001-9858-5817;
MBA, 0000-0002-3130-9794; M-AM, 0000-0003-2710-5374;
ALL, 0000-0003-1809-4327; RM, 0000-0002-4396-3566

The great Tudor warship, the *Mary Rose*, which sank tragically in the Solent in 1545 AD, presents a rare archaeological opportunity to research individuals for whom the precise timing and nature of death are known. A long-standing question surrounds the composition of the Tudor navy and whether the crew were largely British or had more diverse origins. This study takes a multi-isotope approach, combining strontium ($^{87}Sr/^{86}Sr$), oxygen ($\delta^{18}O$), sulfur ($\delta^{34}S$), carbon ($\delta^{13}C$) and nitrogen ($\delta^{15}N$) isotope analysis of dental samples to reconstruct the childhood diet and origins of eight of the *Mary Rose* crew. Forensic ancestry estimation was also employed on a subsample. Provenancing isotope data tentatively suggests as many as three of the crew may have originated from warmer, more southerly climates than Britain. Five have isotope values indicative of childhoods spent in western Britain, one of which had cranial morphology suggestive of African ancestry. The general trend of relatively high $\delta^{15}N$ and low $\delta^{13}C$ values suggests a broadly comparable diet to contemporaneous British and European communities. This multi-isotope approach and the nature of the archaeological context has allowed the reconstruction of the biographies of eight Tudor individuals to a higher resolution than is usually possible.

# 1. Introduction

This study presents new multiple isotope analyses (strontium ($^{87}$Sr/$^{86}$Sr), oxygen ($\delta^{18}$O), sulfur ($\delta^{34}$S), carbon ($\delta^{13}$C) and nitrogen ($\delta^{15}$N)) to investigate the geographical origins and childhood diet of eight members of the *Mary Rose* crew. The overarching aim is to extend the biographies of these men to provide wider insights into Tudor life under the reign of King Henry VIII (r. 1509–1547), at the individual level. The known timing and circumstances of death, the excellent preservation of their skeletal remains and the fact that there is evidence for their professions (electronic supplementary material, table S1) present a very rare opportunity to apply a multi-isotope and ancestry estimation study to provide higher resolution information on individual osteobiographies than most archaeological studies allow.

## 1.1. The *Mary Rose*

The *Mary Rose* was a successful warship for Henry VIII (r. 1509–1547) for 34 years, from 1511 until 1545 (figure 1). Her keel was laid in 1509 and her construction was completed in time for the first French war of 1512–1514. Despite a second war with France, the *Mary Rose,* the flagship of the fleet, was kept in reserve between 1522 and 1536, first at Portsmouth and then refitted on the River Thames. In the early 1540s, she was armed for another impending war against France [1].

On 19 July 1545, during the Battle of the Solent, off the south coast of England, the *Mary Rose* sank resulting in the deaths of the vast majority of her crew [2–4]. Over a few months, half of the hull infilled with estuarine silts encasing much of her contents, including the crew [5]. Over 400 years later, the remaining half of the *Mary Rose* hull and her contents were recovered, conserved and displayed in Portsmouth Historic Dockyard. The preservation of the ship and her contents is exceptional and the recovery of remains of at least 179 crew members has led to considerable research [6–10].

## 1.2. Tudor mobility and diet

As a backdrop to the scientific data produced in this study, it is necessary to consider current scholarship on Tudor mobility and diet. The reign of Henry VIII (r. 1509–1547) marks the end of the medieval period and the beginning of the early modern era in England. This period is characterized by substantial social and economic change, notably the break from Catholicism. However, this is a largely arbitrary historical division rather than a major transition in terms of the lives of the populace. Many medieval customs continued; diet remained mostly unchanged and established trade links with continental Europe and the Mediterranean region continued [11]. Consequently, and as isotopic research on the Tudor period is limited, data on medieval diet and mobility provide a useful comparative resource.

In the Tudor period, England had trade links with The Netherlands, France, Italy, Germany and the Mediterranean [12]. Although regular trade beyond the Mediterranean did not occur until Queen Elizabeth I's reign (r. 1558–1603), there is evidence of contact with African countries, such as Morocco, as early as the late fifteenth century [13]. Trade between countries facilitated the movement of people as well as goods. Between 1500 and 1540 historical documents record the names of over 3000 immigrants settling in England (mostly in the south), originating (in descending numbers) from France, The Netherlands, Germany, Scotland, Belgium, Italy, Spain, the Channel Islands and Portugal [14]. Even within Britain movement of people was common at this time; vagrancy was rife due to population growth and bad harvests [15]. As well as the movement of Europeans, there is evidence for people from further afield settling in Europe. Studies have found that the number of individuals of African ancestry in Tudor England was greater than previously assumed; the names of over 360 individuals have been identified between 1500 and 1640 [16–21]. Historians studying the presence of Africans in Tudor England often use 'African' to describe both direct migrants from Africa and people born in England of Black African descent, as Tudor parish records did not always note the place of birth [16] (see [22,23] for discussions on race in medieval Europe). The majority of records of residents described as 'African' are from port towns in southern England [20]. Interestingly, the papers from the High Court of Admiralty document that one of the first endeavours to salvage the wreck of the *Mary Rose* soon after she sank was attempted by a West African diver, Jacques Francis [19].

Historical research into the diet of British medieval populations has largely produced generalizations of nation-wide diet and have mainly focused on the English perspective [11]. Cereals, in the form of bread, pottage and ale, formed the main component of medieval diets, irrespective of social

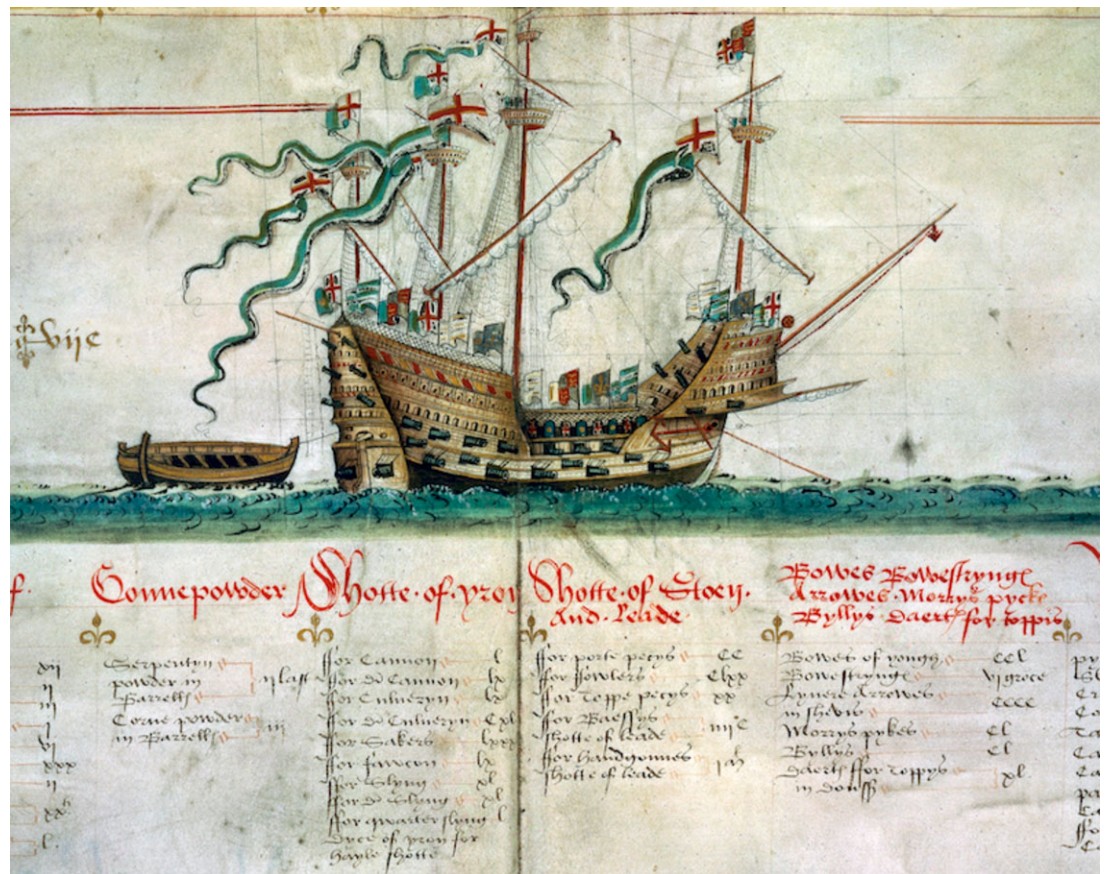

**Figure 1.** The *Mary Rose* depicted on the Anthony Roll (1546), an illustrated inventory of King Henry VIII's navy (by permission of the Pepys Library, Magdalene College, Cambridge, UK).

status [24]. Meats (most commonly beef and pork) and fish (both marine and freshwater) were consumed in different quantities by all levels of late medieval society, as evidenced by historical, isotopic and zooarchaeological research [25–27]. From at least the fourteenth century, the Church had placed greater emphasis on avoidance of meat for fasting rituals, leading to increased fish consumption [25], which is suggested in the widespread elevation of $\delta^{13}C$ and $\delta^{15}N$ values at sites from this period [28,29], but the impact of food insecurities may have also played a role [30,31]. However, by the time of the movement towards the Reformation in the 1530s, with the abandonment of the Lenten fast, there was a decline in fish consumption [25].

## 1.3. The Tudor navy

Henry VIII was an ambitious king, keen to wage war and demonstrate his power in Europe, and the early Tudor period was a time of great development for the royal navy [32]. Although merchant ships continued to be acquired and armed to serve the King in times of war, both Henry VIII and his father commissioned purpose-built warships like the *Mary Rose* [33–35]. In 1545, according to the Anthony Roll (1546 inventory of Henry VIII's navy), the royal navy comprised in excess of 8000 men [36]. Substantial crews of skilled mariners, soldiers and gunners, as well as individuals from wide-ranging professions (cooks, carpenters, pursers, etc.) would have been required during wartime. Retinues (private armies) of English lords would have been called upon during wars, and auxiliaries were sent around Britain where and when needed, for example, the *Elizabeth* of Newcastle was reinforced with a contingent of Devon men in 1513 [33]. Foreign mercenaries were hired during times of war, with contingents from Germany, The Netherlands, Italy and Spain employed to fight the French [37]. Sources from the end of the sixteenth century indicate the presence of individuals of African ancestry aboard English merchant and pirate ships [19]. A palaeopathological study suggests the *Mary Rose* may have been manned by a semi-permanent, professional crew as those sampled were young but suffered many age-related vertebral pathologies potentially caused by continuous work aboard a

warship [7]. However, during the latter half of Henry VIII's reign recruitment of men became more difficult due to the increased size of the royal fleet as well as the lure of profitable privateering [6]. Numbers on royal vessels may have been made up by recruits from merchant ships (in addition to the King acquiring and arming the merchant ships themselves) [38].

There are indications of foreign connections among the *Mary Rose* crew in the form of various ceramic types, including English, French, Low Countries', Rhenish and Iberian wares [3]. These may represent personal belongings brought on board by recruits from overseas, although similar wares would have been available at English ports due to trade connections [3]. In addition, whetstones belonging to the barber surgeon and carpenters have been analysed and may have originated from Brittany, the Channel Islands, Scottish Highlands or regions in southern Europe and North Africa [3].

There has been one isotope study into diet and provenance of the *Mary Rose* crew, using $\delta^{13}C$, $\delta^{15}N$ and $\delta^{18}O$ isotope analysis on 18 individuals [9] (see also [39,40]). The $\delta^{13}C$ and $\delta^{15}N$ analysis on bone collagen provided comparable results to those at other British medieval sites [28,29,41]. Bell *et al.* [9] assert that the $\delta^{18}O$ results demonstrate that at least one-third, and up to almost two-thirds, of the individuals in their sample originated from a more southerly, or warmer, locale than Britain and make the claim that miscommunication between the multi-national crew contributed to the ship's sinking. This paper was heavily criticized by Millard & Schroeder [39], who argued that only one of the 18 sampled individuals (Individual 6) is likely to have spent his childhood outside of Britain, but probably from a more northerly or easterly climatic zone. A reassessment of Bell *et al.*'s [9] data in the light of wider work on $\delta^{18}O$ values in British humans [42,43] supports Millard & Schroeder's [39] conclusion. The historical and scientific evidence discussed above points to diverse naval recruits within the fleet, but direct scientific evidence from human remains from the Tudor period is very limited.

## 1.4. Isotope analysis in mobility and dietary studies

Since the 1970s, isotope analysis has been employed by archaeologists to investigate the diet, origins and mobility of past humans and animals (see §§2.3 and 2.4 for detail on isotope notation). Childhood diet and origins are investigated through isotope analysis of dental samples. Dental tissues do not substantially remodel; therefore, the isotope composition of primary dentine and enamel represents the formation period during childhood [44].

The $^{87}Sr/^{86}Sr$ isotope ratios are incorporated into skeletal tissues from diet and principally relate to the geology of the area where food was produced [45]. As past diets generally comprised locally sourced food, $^{87}Sr/^{86}Sr$ analysis is useful for exploring origins. Time-integrated radioactive decay of the rubidium isotope $^{87}Rb$ to $^{87}Sr$, compared with the stable $^{86}Sr$ isotope, is responsible for the variable $^{87}Sr/^{86}Sr$. Very old and rubidium-rich bedrocks have higher $^{87}Sr/^{86}Sr$, such as granites (up to 0.720 in Britain), whereas younger and low-rubidium bedrocks exhibit lower $^{87}Sr/^{86}Sr$ values, for example, basalts (approx. 0.705) [46].

The $^{18}O/^{16}O$ isotope ratios ($\delta^{18}O$) are primarily derived from ingested fluids, reflecting local drinking water values and are therefore also useful for investigating origins. Values of $\delta^{18}O$ vary according to temperature, altitude, latitude, coastal proximity and precipitation and thus relate to particular climatic zones [47]. A global study of human $\delta^{18}O$ values within archaeological populations by Lightfoot & O'Connell [48] found that the range within most populations is likely to be greater than 3 per mil (‰). This variation could be due to physiological factors (metabolic isotope fractionation), culturally mediated behaviours (e.g. wine and animal milk consumption), food and drink importation and multiple water sources with different $\delta^{18}O$ values [49]. Studies have shown that culinary practices such as brewing, boiling and stewing can elevate the $\delta^{18}O$ values of the water in liquids and food [24,50,51]. There are wide-ranging factors that impact on $\delta^{18}O$ values, and therefore, it is a complex isotope proxy to interpret, a point returned to in the discussion. However, in this study, comparisons will be made with $\delta^{18}O$ data from other medieval sites, where the impact of culinary practices is likely to have been similar.

Combined $^{87}Sr/^{86}Sr$ and $\delta^{18}O$ analyses have long been used in parallel to investigate origins (e.g. [52]) but in the last 15 years, $\delta^{34}S$ ($^{34}S/^{32}S$ isotope ratio) analysis of bone and dentine collagen has also become useful in both dietary and mobility studies in humans and animals [53–57]. Biosphere $\delta^{34}S$ has a wide range; relatively low and negative values are common in inland areas of waterlogged ground and/or impervious lithologies [58]. By contrast, marine resources have relatively high $\delta^{34}S$ values, and food grown and grazed on coastal soils can be influenced by a 'sea-spray effect' which results in $\delta^{34}S$ values which approach marine sulfate composition (+20‰) [59], although local geological and environmental factors can also play a role [60]. Therefore, humans and animals feeding at coastal sites generally exhibit higher $\delta^{34}S$ values than those sourcing their diets from further inland, although estuarial regions also

tend to have high biosphere $\delta^{34}$S [61]. Values above +14‰ may indicate origins within 50 km of the coast [60], with exposed coasts facing prevailing winds producing the highest values [62].

Palaeodietary reconstruction using $\delta^{13}$C ($^{13}$C/$^{12}$C isotope ratio) and $\delta^{15}$N ($^{15}$N/$^{14}$N isotope ratio) values is based on the premise that the isotope values in skeletal tissues reflect the average $\delta^{13}$C and $\delta^{15}$N of consumed food (primarily dietary proteins) [47]. It is beyond this paper's scope to describe all the manifold variables that affect $\delta^{13}$C and $\delta^{15}$N values, as diet is not a major focus. The premises behind these methods have been described in detail elsewhere [47], so are recounted only briefly (and simplistically). Values of $\delta^{13}$C principally vary in relation to the consumption of marine resources and different types of plants ($C_3$ and $C_4$ photosynthesizing plants). Values of $\delta^{15}$N are closely related to the trophic level of the consumer, with humans consuming more animal protein having higher values. The consumption of fish also tends to result in higher $\delta^{15}$N values due to elongated aquatic food chains. As an example, humans that consume mostly marine protein are expected to have $\delta^{13}$C values up to $-12$‰ and high $\delta^{15}$N values up to 15‰ [63]. However, food produced on manured land will also have higher $\delta^{15}$N values [64]. Dietary composition is not the only driver of $\delta^{13}$C and $\delta^{15}$N variation. Origins also have an impact, as varied landscape baseline values mean that the same foodstuffs will have different values depending on where they are produced [65].

More recently, multiple isotope proxies have been employed in combination to investigate archaeological diet and mobility to increase interpretative potential. However, studies that integrate five isotope systems, such as this one, remain rare [53,56,66–69]. Isotope approaches are essentially exclusive and are therefore most suited to discounting potential areas of origin. Consequently, the use of multiple discriminants is more interpretatively powerful in providing the potential to exclude more possible locations. There has been little isotope work on post-medieval human remains in Britain [9,53,70,71]. There is, however, a wealth of data for late medieval Britain, especially in terms of $\delta^{13}$C and $\delta^{15}$N [28,29,72–74], but also for $^{87}$Sr/$^{86}$Sr and $\delta^{18}$O [66,75].

# 2. Material and methods

## 2.1. The Mary Rose 'characters'

The Mary Rose Trust has attributed professions to the skeletal remains of seven individuals who are investigated in this project: Cook (FCS-12), Royal Archer (FCS-70), Archer (FCS-75), Carpenter (FCS-81), Officer (FCS-84), Gentleman (FCS-85) and Purser (FCS-88) (figure 2; electronic supplementary material, table S1). The professions have been ascribed based on context and artefactual associations by the Mary Rose Trust, but are by no means certain as the material could have been displaced [3]. In addition to these characters, an eighth individual (FCS-09, the 'Young Mariner') was investigated as a photogrammetry study (N Owen 2018, personal communication) highlighted the possibility that this individual was potentially of African ancestry.

## 2.2. Sample selection

Enamel samples were extracted from the second molar (M2) for $^{87}$Sr/$^{86}$Sr and $\delta^{18}$O analysis, and dentine was extracted from the adjacent first molar (M1) root for $\delta^{13}$C, $\delta^{15}$N and $\delta^{34}$S analysis. The same sampling method was followed for all individuals. A ca 4 mm strip of enamel was extracted from the lower half (i.e. from the root–enamel junction towards the occlusal surface) of the lingual surface of the M2. The strip was divided into mesial and distal sections, with the larger being analysed for $^{87}$Sr/$^{86}$Sr and the other for $\delta^{18}$O. A whole M1 root was sampled for each individual. Due to the exceptional preservation of the teeth and the fact that maxillary and mandibular molars should produce similar isotope ratios [76], teeth/roots were selected based on the ease of sampling to minimize damage (see electronic supplementary material, table S2).

Each tooth provides information from different periods of childhood [42]. M1 roots and M2 crowns form between 2½ and 9½ years of age, thus sampling dentine and enamel from these zones provides broadly comparable temporal signals [77]. The duration of incorporation of isotope compositions into dental enamel remains poorly understood, especially in relation to mineralization [49,78], and therefore, it is not possible to be precise about the temporal range. Bioapatite (in enamel) does not remodel and collagen (in dentine) undergoes very little remodelling; thus the isotope compositions of teeth represent an archive of childhood diet and geographic origins [45]. All enamel samples cover a period of several years of incorporation, thus minimizing any potential seasonal effect on $\delta^{18}$O values and limiting issues

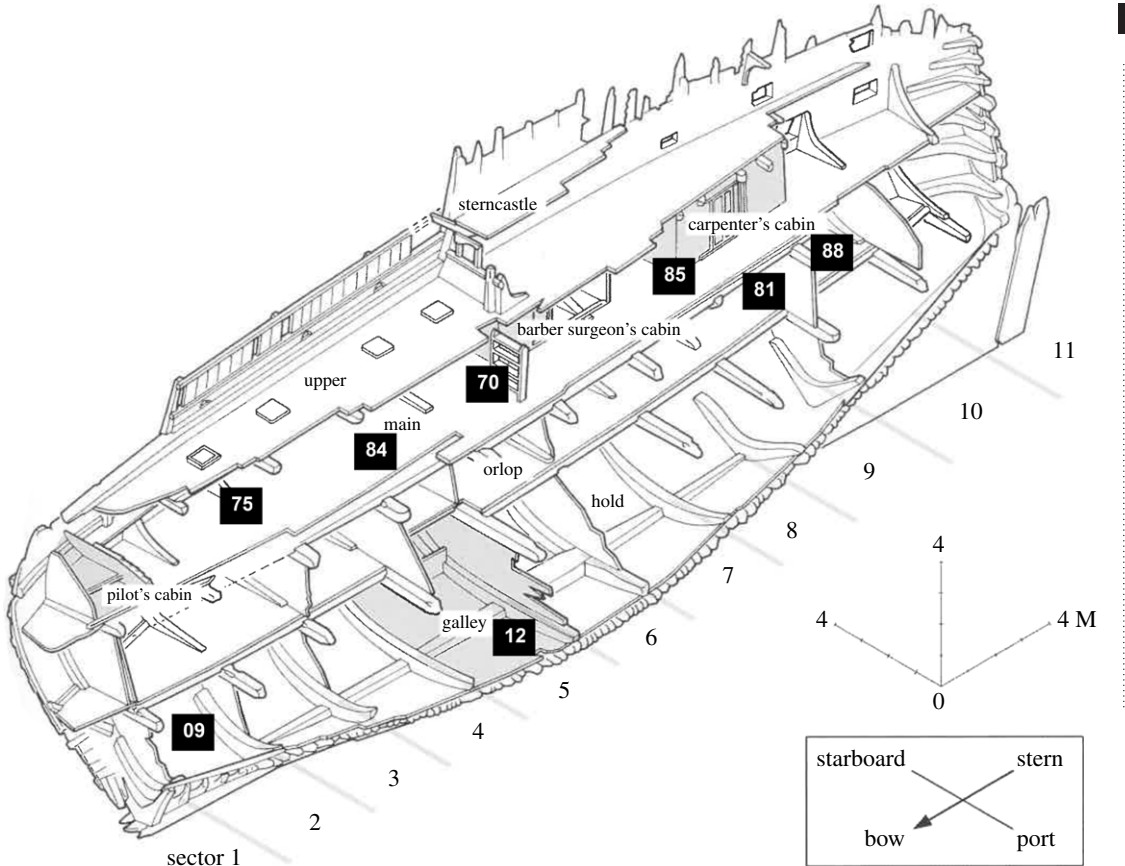

**Figure 2.** Cutaway diagram of the *Mary Rose* ship indicating the locations of individuals sampled for this study (adapted from Marsden & McElvogue [5], © Mary Rose Trust). FCS-09: the Young Mariner, FCS-12: the Cook, FCS-70: the Royal Archer, FCS-75: the Archer, FCS-81: the Carpenter, FCS-84: the Officer, FCS-85: the Gentleman and FCS-88: the Purser.

surrounding intra-individual variation [79]. The youngest remains analysed from the *Mary Rose* were from a 10 year old [3,6], so this strategy should ensure signals relate to pre-recruitment childhood origins. Furthermore, this age bracket should avoid nursing signals, as historical and isotope studies indicate that weaning in this period was generally complete by 2 years [26,80].

In addition to the dental samples, a collagen sample from a rib bone (which, unlike dentine, remodels substantially, [81]) was taken from FCS-09 for $\delta^{13}C$ and $\delta^{15}N$ isotope analysis to investigate diet in the years before death, prompted by the only instance of severe, bilateral buccal cemento-enamel junction caries among the individuals recovered from the *Mary Rose*.

## 2.3. Enamel sample preparation ($^{87}Sr/^{86}Sr$ and $\delta^{18}O$ analysis)

Enamel samples (20–50 mg) were cut from teeth and cleaned using a diamond saw and burr to remove all adhering dentine and greater than 10 μm of the enamel surface. Samples for $^{87}Sr/^{86}Sr$ (approx. 20–40 mg) and $\delta^{18}O$ (approx. 2.5–5 mg) analysis were then divided.

For $^{87}Sr/^{86}Sr$ analysis, samples were ultrasonically cleaned in deionized water and transferred to a clean working area (class 100, laminar flow) for further preparation and $^{87}Sr/^{86}Sr$ analysis at Cardiff Earth Laboratory for Trace element and Isotope Chemistry (CELTIC). Samples were digested in 8 M $HNO_3$ and heated overnight at 120°C. Strontium extraction from enamel samples used Sr Spec™ resin using a revised version of the protocol of Font *et al.* [82]. Samples were loaded into resin columns in 1 ml 8 M $HNO_3$. Matrix elements (including Ca and traces of Rb) were then removed in several washes of 8 M $HNO_3$ before Sr was eluted and collected in 0.05 M $HNO_3$. Samples were placed on a hotplate (120°C) to dry overnight before this process was repeated for a second pass for the effective removal of any remaining traces of Ca. Once dry, purified samples were re-dissolved in 0.3 M $HNO_3$. The $^{87}Sr/^{86}Sr$ ratios were measured using a Nu Plasma II multi-collector inductively coupled plasma mass spectrometer (MC-ICP-MS) at Cardiff University. Samples were introduced using an Aridus II

desolvator introduction system. All data were first corrected for on-peak blank intensities, then mass bias corrected using the exponential law and a normalization ratio of 8.375209 for $^{88}Sr/^{86}Sr$ [83]. Residual krypton (Kr) and rubidium ($^{87}Rb$) interferences were monitored and corrected for using $^{82}Kr$ and $^{83}Kr$ ($^{83}Kr/^{84}Kr = 0.20175$ and $^{83}Kr/^{86}Kr = 0.66474$; without normalization) and $^{85}Rb$ ($^{85}Rb/^{87}Rb = 2.5926$), respectively. Analysis of NIST SRM 987 during the analytical session gave a $^{87}Sr/^{86}Sr$ value of $0.710292 \pm 0.000007$ ($2\sigma_N$, $n = 11$), and all data are corrected to NIST SRM 987 values of 0.710248 [84]. Total procedural blanks are typically less than 20pg of Sr, which is negligible relative to the Sr in samples (greater than 20 ng). Accuracy of the NIST SRM 987 normalization and the chemistry processing was assessed by repeat measurements of $^{87}Sr/^{86}Sr$ ratio in NIST SRM 1400 (Bone Ash, processed through chemistry similar to the unknown samples), giving an average $^{87}Sr/^{86}Sr$ ratio of $0.713111 \pm 0.000014$ ($2\sigma_N$, $n = 5$), which is consistent with the published value ($0.713126 \pm 0.000017$ [85]).

For $\delta^{18}O$ analysis, enamel was powdered using an agate pestle and mortar. The isotope composition of the structural carbonate within the enamel was measured ($\delta^{18}O_{carbonate}$). At the Cardiff University Stable Isotope Facility, samples were acidified for 5 min with greater than 100% ortho-phosphoric acid at 70°C [86] and analysed in duplicate using a Thermo MAT253 dual inlet mass spectrometer coupled to a Kiel IV carbonate preparation device. The resultant isotope values are reported as per mil ($^{18}O/^{16}O$) normalized to the VPDB scale using an in-house carbonate reference material (BCT63) calibrated against NBS19 certified reference material. The $\delta^{18}O_{carbonate}$ values are then converted into the VSMOW scale (VSMOW = $1.0309 \times \delta^{18}O$ VPDB + 30.91 [87]). The long-term reproducibility for $\delta^{18}O$ BCT63 is ±0.03‰ (1σ). The standard deviation of replicate $\delta^{18}O$ measurements is 0.049‰. The $\delta^{18}O_{carbonate}$ values were converted to $\delta^{18}O_{phosphate}$ ($\delta^{18}O_p$) values (following [88]) to allow for comparison with other datasets. The error on the calculated $\delta^{18}O_p$ values is 0.24‰, based on the analytical error and the error in the conversion regression equation [88].

## 2.4. Dentine and bone collagen sample preparation ($\delta^{13}C$, $\delta^{15}N$ and $\delta^{34}S$ analysis)

Collagen was prepared for $\delta^{13}C$, $\delta^{15}N$ and $\delta^{34}S$ analysis following a modified Longin method [89]. Approximately 0.4 g of root dentine and 1 g of rib bone were extracted using a precision drill with diamond wheel attachment. Collagen was extracted from the whole length of roots to provide an average for the period of development (see electronic supplementary material, table S2). Sample surfaces were lightly abraded using a burr to remove contaminants and greater than 10 µm of the cortex. The root and rib samples were then placed in test tubes in 7 ml 0.5 M HCl and stored at 4°C. Acid was changed weekly until full demineralization. Each sample was then washed with deionized water (less than or equal to 5 µS cm$^{-1}$) and gelatinized in a pH3 solution (acidified with HCl) at 75°C for 48 h. The supernatant was then filtered (8 µm Ezee-filter, Elkay, Basingstoke), frozen and freeze dried. Samples were weighed and analysed in duplicate. Ratios of $\delta^{34}S$, $\delta^{13}C$ and $\delta^{15}N$ are reported in per mil (‰) relative to VCDT, VPDB and AIR standards, respectively. The $\delta^{34}S$ analysis was undertaken at the National Environmental Isotope Facility, Keyworth, Nottinghamshire. The instrumentation comprises a ThermoFinnigan EA IsoLink coupled to a Delta V Plus isotope ratio mass spectrometer via a Conflo IV interface. Values of $\delta^{34}S$ were calibrated using IAEA S-1 and S-2 with secondary checks comprising in-house standard M1360P (powdered gelatine) with laboratory-accepted delta value of 2.99‰ and modern cattle bone collagen (laboratory-accepted value of 12.71‰). The average 1σ reproducibility across the standard material was ±0.28‰. M1360P was used to calculate %S (0.77% S, calibrated to IAEA S-1 and S-2). The 1σ standard reproducibility was ±0.25‰. Both $\delta^{13}C$ and $\delta^{15}N$ ratios were measured at the Cardiff University Stable Isotope Facility using a Flash 1112 elemental analyser coupled to a Thermo Delta V Advantage. Ratios of $\delta^{13}C$ and $\delta^{15}N$ were calibrated against caffeine (laboratory grade, 98.5%, Acros Organics, lot A0342883) and an in-house supermarket gelatine standard, which is calibrated against IAEA-600 ($\delta^{13}C$ and $\delta^{15}N$), IAEA-CH-6 ($\delta^{13}C$) and IAEA-N-2 ($\delta^{15}N$). The 1σ ($n = 54$) reproducibility was ±0.06 for $\delta^{13}C$ and ±0.07 for $\delta^{15}N$. Different weights of caffeine were analysed to establish a calibration equation for the abundance of C and N against signal intensity in the mass spectrometer, which was used to calculate %N and %C in actual samples. The content of caffeine (28.85%N, 49.48%C) was calculated from its chemical formula ($C_8H_{10}N_4O_2$). All samples adhered to published quality control indicators [90,91].

## 2.5. Methods of ancestry estimation

During a photogrammetry study of FCS-09 [76], it was suggested that this individual may have been of African ancestry and additional craniometric and morphological analysis was conducted to test this.

Osteological ancestry estimation is not often applied to archaeological remains in Britain (see [92–94] for exceptions). The authors acknowledge the problematic history of ancestry estimation stemming from early pursuits of scientific racism [95]. Although we recognize the methodological and ethical concerns regarding these approaches in forensic contexts [96], we also agree that not seeking evidence of human diversity and phenotypic variation in the archaeological record only obscures it further [94], something historians are actively addressing [16–21]. Ancestry analysis was also performed on FCS-70 and FCS-81, because of their unusual $\delta^{18}O$ values and good cranial preservation. Further information on these methods is provided in the electronic supplementary material.

Craniometrics [97] were analysed via Fordisc 3.1 [98], which compares individual samples with known (recent) skeletal populations. Samples were compared with Black, Hispanic and White males from the Forensic Databank. Posterior probabilities, which estimate group affinity based on relative distance and assumes affinity to one reference group, and typicality probability (Chi-square), which gives a likelihood of group affinity based on the absolute distance of the sample from the group's centroid, are reported alongside group affinity estimates. Morphoscopic analysis followed Decision Tree and Optimized Summed Scores Attributes (OSSA) methods [99]. The skeletal populations these methods are based on are also recent. Therefore, interpretation was conducted with an understanding of the historical, colonial framework which led to these collections and the ancestry classifications they use [100–102], as well as evidence for Tudor migration and diversity discussed above. This framing was particularly important in understanding the designation of 'Hispanic' in a Tudor context. This classification is used in the USA and refers to individuals with origins or ancestral connections to Spanish-speaking regions, particularly from the Americas [103,104]. Applying this linguistic and social distinction to physical remains is practically and ethically challenging, and requires an understanding of the colonialist practices in 'Hispanic' regions, which ultimately led to an admixture of Indigenous populations, (enslaved) African and European settlers [104]. It is also important to recognize that the reference populations of Black Americans are the result of racialization that follows centuries of enslavement and discrimination which failed to recognize the substantial phenotypic diversity of individuals with origins or ancestral connections to the African continent [105,106]. This cannot be assumed to be the same perception of 'Black' in Tudor England though its origins can be traced before this period [22].

# 3. Isotope results

## 3.1. Provenance isotope results

The isotope data from $^{87}Sr/^{86}Sr$, $\delta^{18}O$ and $\delta^{34}S$ analyses are displayed in table 1 with summary statistics in table 2 (see electronic supplementary material, tables S3 and S4 for extended $\delta^{18}O$ and $^{87}Sr/^{86}Sr$ data, including quality control indices). All three proxies produced relatively wide-ranging results. The $^{87}Sr/^{86}Sr$ values range from 0.70872 to 0.71070 (figure 3), with a mean of 0.70966 ($\pm0.00072$, $1\sigma$). Overall, the $\delta^{18}O_P$ values are high for a British dataset, ranging between 18.4‰ and 21.2‰ (figure 3), with a mean of 19.2‰ ($\pm1.0$‰, $1\sigma$). The mean is skewed by one clear outlier, FCS-70 (21.2‰), who has one of the highest values measured in a human from a British archaeological context, and therefore, the median of 18.9‰ provides a more balanced average. Three individuals (FCS-70, FCS-81 and FCS-85) have $\delta^{18}O_P$ values that fall more than two standard deviations outside of the British mean (17.7‰ $\pm1.4$‰, $2\sigma$, $n = 615$, [42]). One individual (FCS-81) is at the very extreme end of human values reported in Britain and another (FCS-70) is well beyond it [42]. The $\delta^{34}S$ results range widely from −3.3‰ to 17.4‰, with a mean of 10.5‰ and a large standard deviation ($\pm8.2$‰, $1\sigma$). Half of the individuals (FCS-9, FCS-12, FCS-75 and FCS-85) have high (greater than 14‰) $\delta^{34}S$ values (15.7 to 17.4‰). One individual (FCS-88) has an intermediate value (12.6‰), and the others (FCS-84, FCS-70 and FCS-81) have low $\delta^{34}S$ values (−3.3 to 6.9‰).

## 3.2. Dietary isotope results

The isotope results are presented in tables 1 and 2. The $\delta^{13}C$ values range between −20.1‰ and −18.3‰, with a mean of −19.4‰ ($\pm0.7$‰, $1\sigma$) and $\delta^{15}N$ values range between 9.4‰ and 12.8‰ (figure 4), with a mean of 11.2‰ ($\pm1$‰, $1\sigma$). Two outliers were observed in terms of $\delta^{15}N$. FCS-88 has the highest $\delta^{15}N$ value (12.8‰) and FCS-75 has a markedly lower value than the rest of the dataset (9.4‰). FCS-12 has the highest $\delta^{13}C$ (−18.3‰) and $\delta^{34}S$ value (17.4‰) as well as a high $\delta^{15}N$ value (11.3‰). When

**Table 1.** Multi-isotope analysis results from dental samples of eight individuals from the *Mary Rose* (see electronic supplementary material, table S1 for contextual information on these individuals). Oxygen values were converted from carbonate ($\delta^{18}O_c$) to phosphate ($\delta^{18}O_p$) using the conversion equation set out in Chenery *et al.* [88]. Sulfur isotope analysis carried out by BGS © UKRI.

| skeleton no. | character | $^{87}Sr/^{86}Sr$ | $\delta^{18}O_c$ | $\delta^{18}O_p$ | $\delta^{34}S$ | %S | C:S | N:S | $\delta^{13}C$ | $\delta^{15}N$ | %C | %N | C:N |
|---|---|---|---|---|---|---|---|---|---|---|---|---|---|
| FCS-09 rib | Young Mariner | | | | | | | | −19.0 | 11.7 | 40.4 | 14.7 | 3.2 |
| FCS-09 | Young Mariner | 0.71070 | 27.2 | 18.4 | 16.7 | 0.30 | 373 | 115 | −18.8 | 11.2 | 41.8 | 15.0 | 3.2 |
| FCS-12 | Cook | 0.70965 | 27.7 | 18.9 | 17.4 | 0.30 | 387 | 119 | −18.3 | 11.3 | 42.6 | 15.3 | 3.2 |
| FCS-70 | Royal Archer | 0.70888 | 30.0 | 21.2 | 0.6 | 0.27 | 421 | 130 | −20.0 | 10.9 | 41.8 | 15.1 | 3.2 |
| FCS-75 | Archer | 0.71055 | 27.4 | 18.6 | 15.7 | 0.30 | 392 | 121 | −19.5 | 9.4 | 42.6 | 15.3 | 3.2 |
| FCS-81 | Carpenter | 0.70872 | 28.5 | 19.8 | −3.3 | 0.28 | 403 | 124 | −20.0 | 11.4 | 42.3 | 15.2 | 3.3 |
| FCS-84 | Officer | 0.70915 | 27.2 | 18.4 | 6.9 | 0.29 | 383 | 119 | −20.1 | 10.8 | 40.9 | 14.8 | 3.2 |
| FCS-85 | Gentleman | 0.70983 | 28.1 | 19.3 | 17.4 | 0.29 | 388 | 119 | −19.0 | 11.7 | 41.3 | 14.8 | 3.3 |
| FCS-88 | Purser | 0.70979 | 27.6 | 18.8 | 12.6 | 0.26 | 437 | 134 | −19.3 | 12.8 | 42.6 | 15.2 | 3.3 |

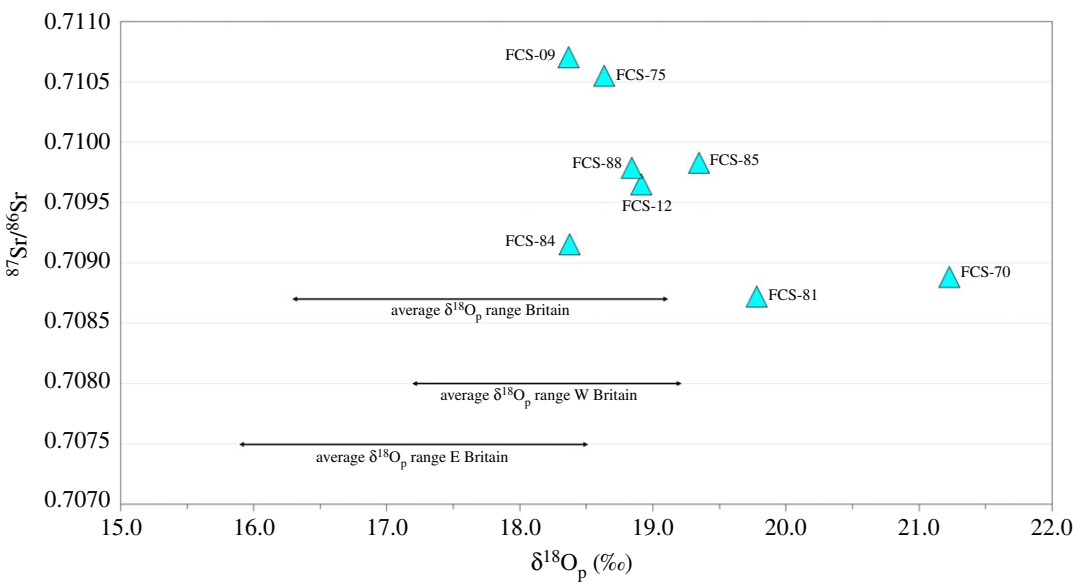

**Figure 3.** Oxygen ($\delta^{18}O_{phosphate}$) and strontium isotope results for the eight *Mary Rose* crew members along with $2\sigma$ ranges for Britain as a whole, and east (E) and west (W) regions, following Evans *et al.* [42].

**Table 2.** Summary statistics for all isotope proxies for eight *Mary Rose* individuals (excl. FCS-09 rib sample). Sulfur isotope analysis carried out by BGS © UKRI.

| isotope proxy | mean | s.d. | median | IQR | min. | max. |
|---|---|---|---|---|---|---|
| $\delta^{18}O_c$ | 27.97 | 0.93 | 27.67 | 0.86 | 27.18 | 29.95 |
| $\delta^{18}O_p$ | 19.19 | 0.95 | 18.88 | 0.89 | 18.37 | 21.23 |
| $^{87}Sr/^{86}Sr$ | 0.709658 | 0.000724 | 0.709719 | 0.000925 | 0.708718 | 0.710701 |
| $\delta^{34}S$ | 10.50 | 8.16 | 14.15 | 11.55 | −3.30 | 17.40 |
| $\delta^{13}C$ | −19.36 | 0.66 | −19.39 | 1.03 | −20.11 | −18.29 |
| $\delta^{15}N$ | 11.19 | 0.95 | 11.23 | 0.59 | 9.40 | 12.79 |

comparing the $\delta^{34}S$ and $\delta^{13}C$ data, the three individuals with the lowest $\delta^{34}S$ values also have the lowest $\delta^{13}C$ values (FCS-70, FCS-81 and FCS-84) and the three individuals with the highest $\delta^{34}S$ values also have the highest $\delta^{13}C$ values (FCS-09, FCS-12 and FCS-85). Although sample sizes are deemed too small for statistical analysis, $\delta^{34}S$ and $\delta^{15}N$ isotope data show no clear relationship.

FCS-09 was the only individual for which different elements were analysed to investigate temporal changes in diet. Only a minor elevation in $\delta^{15}N$ was observed between the early developing dentine and later developing rib samples. The dietary signals were well within the range of the other crew members, and the buccal caries are likely to result from a cultural practice that had a negligible dietary impact.

# 4. Discussion

The general trend of relatively high $\delta^{15}N$ and low $\delta^{13}C$ values for all eight of the crew are characteristic of a largely terrestrial protein-rich childhood diet in a $C_3$ ecosystem and consistent with isotope data from other late medieval British sites (figure 4; see [107–109] for European sites). Individuals with high $\delta^{15}N$ and relatively low $\delta^{13}C$ have been interpreted as representing protein-rich diets involving the substantial consumption of freshwater fish, omnivorous animals or crops and animals raised on manured land [28,41] (but see [30,31]).

The eight *Mary Rose* individuals have $^{87}Sr/^{86}Sr$ values (0.70872 to 0.71070) compatible with British origins, which Evans *et al.* [61] estimates as ranging between 0.7071 and 0.7183. The $^{87}Sr/^{86}Sr$ values are

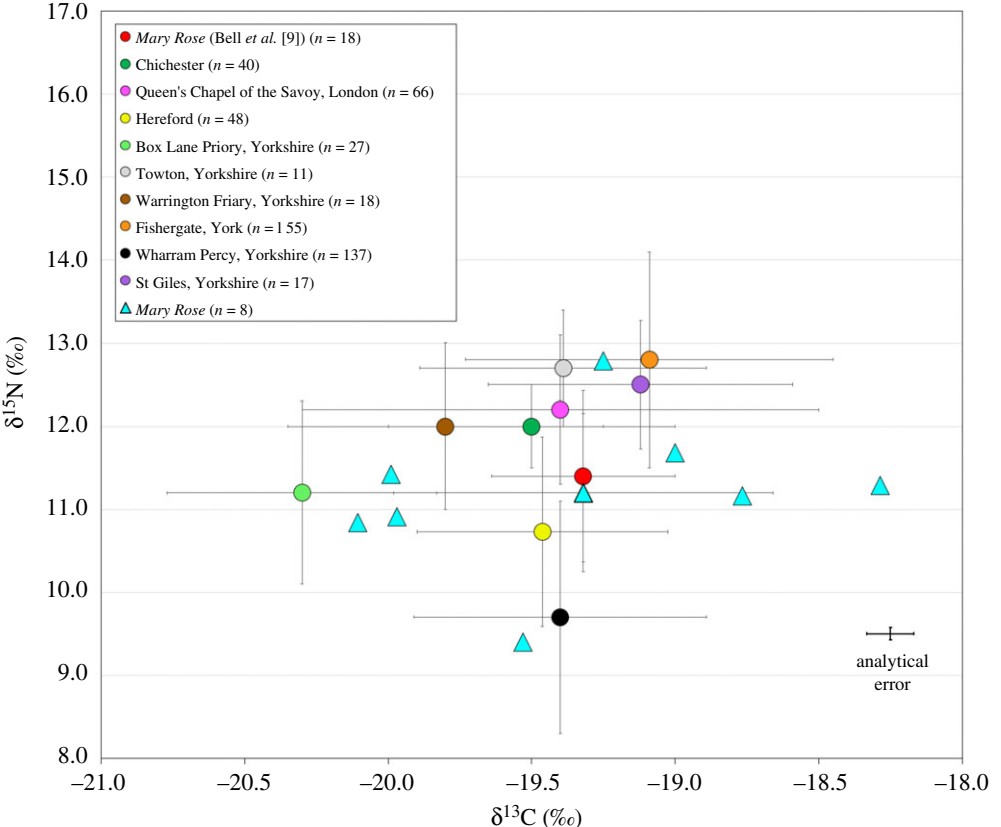

**Figure 4.** Carbon and nitrogen isotope results for the eight *Mary Rose* crew members (individual data points plotted as well as the mean), compared with isotope data from post-medieval sites (*Mary Rose* [9], Chichester [71], Queen's Chapel of the Savoy, London [70]) and medieval sites (Hereford [74], Box Lane Priory, Yorkshire [73], Towton, Yorkshire [28], Warrington Friary, Yorkshire [28], Fishergate, York [29], Wharram Percy, Yorkshire [72] and St Giles, Yorkshire [28]). All error bars = $1\sigma$. The analytical error relates to the *Mary Rose* data from this study. (N.B. comparative data comprises data from both bone and dentine collagen).

not particularly diagnostic, with values in this range potentially deriving from various areas across Britain [61]. Furthermore, $^{87}Sr/^{86}Sr$ values such as these have been recorded at various locations across Europe, the Mediterranean, Africa and the Middle East and are by no means unique to Britain [46,110–118]. Interpretation is difficult as there is no local baseline to compare with for distinguishing locals from non-locals and therefore the integration of multiple proxies is key to interpretation.

The approach to $\delta^{18}O$ analysis was designed to address challenges and ambiguities in the interpretation of provenance, but it is not possible to discount all issues of equifinality. Recommended methods were employed [86], sampling locations were standardized to ensure comparability and to minimize seasonal signals [49,79], drinking water corrections were not used [119], and interpretations were based on comparisons with large British datasets [42,43]. The *Mary Rose* provides a particular challenge, as it cannot be treated as a settled community from which outliers can be identified statistically, as advocated by Lightfoot & O'Connell [48]. The same authors state that pinpointing specific homelands should not usually be attempted. We therefore take an exploratory approach, suggesting potential areas of origin based on integration with other isotope proxies and artefactual evidence.

The $\delta^{18}O$ values suggest that as many as three individuals may be from warmer climates than Britain. FCS-70, FCS-81 and FCS-85 have high $\delta^{18}O_P$ values (21.2‰, 19.8‰ and 19.3‰, respectively) that are more than two standard deviations from the British mean of 17.7‰ ±1.4‰ ($2\sigma$, $n = 615$; [42,43]). The dataset compiled for this estimation included data from medieval sites and therefore accounts for possible elevations in $\delta^{18}O$ of human enamel samples relating to medieval culinary practices, such as brewing, stewing and boiling [24]. Nevertheless, due to the various factors affecting $\delta^{18}O$ values in archaeological populations, $\delta^{18}O$ analysis remains a complex tool for provenancing humans and must be interpreted with caution. Comparison with existing data from the *Mary Rose* must also be made with caution. Bell *et al.*'s [9] data are not directly comparable due to a different sampling methodology and the use of a NaClO pre-treatment, which has been demonstrated to make $\delta^{18}O$

values lower [120]. Once converted to $\delta^{18}O_p$ [88], the mean for the previous dataset ($17.6 \pm 0.87$‰, $1\sigma$) is markedly lower than in this study ($19.2 \pm 0.95$‰, $1\sigma$), although there is considerable overlap in the datasets. This difference may derive from chance sampling, as both studies relate to only a small number of individuals or may relate to varied sampling and pre-treatment methods.

Three of the five 'British' individuals (FCS-09, FCS-12 and FCS-75) have high $\delta^{34}S$ values (greater than 14‰) that suggest coastal origins [60]; FCS-88 has an intermediate value and FCS-84 has a low 'inland' value. Within the group of likely migrants, FCS-85 has a high 'coastal' $\delta^{34}S$ value whereas FCS-70 and FCS-81 have very low $\delta^{34}S$ values suggesting 'inland' residence, potentially in areas of wetland and/or impervious lithology [58]. The three individuals with the highest $\delta^{34}S$ values (FCS-09, FCS-12 and FCS-85) also had the highest $\delta^{13}C$ values, supporting a coastal residence [59].

The two potential sub-sets of the crew are discussed below, with more detailed biographical reconstruction for individuals with isotope values outside of the British range.

## 4.1. Individuals with isotope values consistent with British origins

Five of the individuals (FCS-09, FCS-12, FCS-75, FCS-84 and FCS-88) have isotope values consistent with a childhood spent in Britain. With an $\delta^{18}O_p$ value of 18.4‰ and a $^{87}Sr/^{86}Sr$ value of 0.71070, FCS-09 sits well within British isotope ranges and is discussed in more detail in a later section, as ancestry analyses were also undertaken.

All five have $\delta^{18}O_p$ values (group mean = 18.6‰) closer to the mean value for western Britain ($18.2 \pm 1.0$‰, $2\sigma$, $n = 40$) than for eastern Britain ($17.2 \pm 1.3$‰, $2\sigma$, $n = 83$) [42]. These crew members were therefore probably raised in the west of Britain or perhaps the southern coastal strip, which is also characterized by higher $\delta^{18}O_p$ values [42], although overlap in baseline values across adjacent regions means this cannot be determined with great confidence.

FCS-09 and FCS-75 have very similar isotope values (FCS-09: $\delta^{18}O_p = 18.4$‰, $^{87}Sr/^{86}Sr = 0.71070$, $\delta^{34}S = 16.7$‰; FCS-75: $\delta^{18}O_p = 18.6$‰, $^{87}Sr/^{86}Sr = 0.71055$, $\delta^{34}S = 15.7$‰) and may have originated from the same region. Their $\delta^{18}O_p$ values suggest a western or southern British origin, and their $\delta^{34}S$ values indicate a childhood spent in close proximity (less than 30 km) to the sea [60,61]. Values of $^{87}Sr/^{86}Sr$ close to 0.7110 are commonly found on Palaeozoic rocks (e.g. Devonian, Silurian and Ordovician) that are common on the western side of southern Britain, such as in South Wales, Devon and Cornwall, as well as Carboniferous grits in North Devon [46,61,111]. A coastal settlement in the southwest of Britain is a plausible childhood residence for FCS-09 and FCS-75. Privateering traditions, commercial trade routes and the long medieval wars with France encouraged lively maritime activity in the southwest of Britain in the sixteenth century [121]. Well-known maritime hubs of the period that are consistent with these values include Poole, Weymouth (Dorset), Plymouth, Dartmouth (Devon) and Fowey (Cornwall) [61,122].

Suggesting origins for the remaining three British individuals is more difficult. The $\delta^{18}O$ values indicate likely origins in the west or south. The $^{87}Sr/^{86}Sr$ values of FCS-12 (0.70965), FCS-84 (0.70915) and FCS-88 (0.70979) relate to rocks of various ages across vast swathes of Britain [46]. All are higher than would be expected for chalk bedrocks and therefore large areas of Wessex and southeast England can be discounted. The values are also too low to relate to older, more radiogenic geologies such as Proterozoic and Archaean bedrocks, which cover large parts of Scotland and pockets of northern England [46,111]. Most of Wales can be excluded, as this area is dominated by more radiogenic lithologies [61]. The $^{87}Sr/^{86}Sr$ values may relate to limestone geologies although there are various other possibilities within Britain (and beyond). Values between 0.709 and 0.710 are very common in the British biosphere, leading Montgomery [123] to describe this range as 'the strontium of doom' due to the limited interpretative resolution that can be achieved. Considering the $\delta^{34}S$ values of these three individuals, FCS-12 has a very high $\delta^{34}S$ value (17.4‰) indicative of coastal origins, most likely from more exposed western areas. FCS-88 (12.6‰) has a more intermediate value, whereas FCS-84 (6.9‰) had a more inland childhood abode [61]. Despite similar $^{87}Sr/^{86}Sr$ values, it is clear that all three came from different areas. A range of western coastal zones provides potential areas of origin for FCS-12. FCS-88 could derive from the Thames estuary or other southern/western areas, though probably not from the most exposed coastal zones of the west. FCS-84 could come from the southern end of the English midlands or inland areas of Wessex that are not on chalk lithology (e.g. north Wiltshire). However, pinpointing origins is not possible, and these assignments must be considered as suggestions based on the current mapping.

Due to the size of the *Mary Rose* and the threat of French invasion, she never ventured further than the seas around Britain and coastal areas of northern France. Apart from refitting in the River Thames

prior to the Third French War, she is believed to have mostly kept to the Channel [5]. Recruits may well have been picked up along the southern coast of Britain, although contingents of men were sent around the country to man ships during times of war. In 1514, for example, the crew of the royal ship *Lizard* comprised contingents from the coastal towns of Beaumaris, Plymouth, Hull and Tynemouth [34]. The historical evidence does not highlight particular regions for Tudor naval recruitment; however, the isotope results from this study suggest five western/southern British individuals, four of which may have grown up near the coast. It should be noted that although the individuals discussed above have isotope values compatible with Britain, other regions of Europe cannot be ruled out. Origins in Britain provide the most parsimonious interpretation, but isotope analysis is an exclusive approach and vast swathes of northern Europe cannot be excluded.

## 4.2. Individuals with $\delta^{18}O_p$ values outside the British range

Three individuals (FCS-70, FCS-81 and FCS-85) have $\delta^{18}O$ values beyond the expected British range [42,43]. High $\delta^{18}O_p$ values (greater than 19‰) in humans are common in warmer climates than Britain, such as southern European coasts, southwest Iberia and North Africa [48,124]. It must be stressed that there are multiple factors, other than migration from a warmer climate, which could cause higher $\delta^{18}O_p$ values relative to the other individuals (e.g. altitude, latitude, diet, temperature, water source; [49]). However, considering late medieval and early modern isotope data from Britain [9,26,28,42,61,75,125,126], the $\delta^{18}O_p$ values for these three individuals suggest origins in a warmer region than Britain. This is supported by artefactual evidence from the *Mary Rose*, discussed below.

FCS-85 may well have originated from a southern European coastal site as he has a high $\delta^{18}O_p$ value of 19.3‰, a 'coastal' $\delta^{34}S$ value of 17.4‰ and a $^{87}Sr/^{86}Sr$ value (0.70983) consistent with various Mesozoic and late Palaeozoic sediments found extensively around the Mediterranean coastline, including southern Spain, Italy and Greece [48,60,110,124,127]. There were trade links with cities around the Mediterranean during the Tudor period, which probably resulted in the movement of people as well as goods [12]. FCS-85 is associated with a gentleman's chest which contained a decorated casket panel similar to those produced in northern Italy in the late fourteenth and fifteenth centuries [3]. However, procurement of such items would have been possible through trade and does not necessarily represent Italian origins. A coastal region in southern Europe is a likely place of origin for this individual based on his isotope values and the historical/artefactual evidence.

FCS-70 and FCS-81 have $^{87}Sr/^{86}Sr$ values consistent with various Mesozoic lithologies around the Mediterranean [110,127]. They have higher $\delta^{18}O_p$ values than would generally be expected of residents of southern Europe (FCS-70 = 21.2‰, FCS-81 = 19.8‰), especially FCS-70 [128–130] (but see [131]). They also have very low $\delta^{34}S$ values (0.6‰ and −3.3‰, respectively), which excludes origins within coastal zones and raises the possibility of areas of wetland/impervious geology [58,60,61]. With an $\delta^{18}O_p$ value of 19.8‰, an upbringing in inland southern Europe is a possibility for FCS-81 (the 'Carpenter'). Material evidence from the carpenter's cabin suggests that at least one of the six carpenters on board may have been Spanish. Four adzes with a 'stirrup' design, commonly found in Spain during this period were recovered, and there were a few coins identified as Spanish in an embroidered leather pouch found in a personal chest in the cabin [3]. Combining isotope and artefactual evidence, inland southwest Iberia provides a plausible area of origin.

Although an $\delta^{18}O_p$ value of 21.2‰ is rather high for residence in Europe [128–130] (but see [131]), warmer southern regions of Europe certainly cannot be ruled out. Interestingly FCS-70 (the 'Royal Archer') had a leather wristguard bearing the symbol of a pomegranate, associated with the Moors in the medieval and early modern periods, as a fruit that they had brought to Europe from Africa, and was also a symbol used for Catherine of Aragon's branch of the English royal family [20]. Considering the potential elevation of $\delta^{18}O$ values due to dietary factors, it is possible that this individual was an Iberian Moor. This is supported by the dietary isotopes of FCS-70 which indicate a $C_3$-based diet, consistent with Spanish origins [132], opposed to $C_4$ which is more expected for residents of more arid climates [63], like Africa. Nevertheless, $C_3$ crops were widespread by the medieval period, and so low $\delta^{13}C$ values do not rule out more arid climates than Britain, such as North Africa, where populations cultivated European crops such as wheat, similar to other Mediterranean agricultural communities [133]. Considering the $^{87}Sr/^{86}Sr$ value of FCS-70, regions such as northeastern Morocco [134] and locations along the Continental Intercalaire aquifer which spans the Atlas Mountains in Algeria to the Chotts of southern Tunisia [135] are possibilities, as are locations in Iberia, but pinpointing origins is not possible.

**Table 3.** Ancestry estimation summary for three *Mary Rose* individuals.

| skeleton no. | Fordisc classification | posterior probability | Chi-square typicality | Decision Tree estimate | OSSA estimate |
|---|---|---|---|---|---|
| FCS-09 | Hispanic male | 0.611 | 0.823 | Black | Black |
| FCS-70 | White male | 0.640 | 0.352 | Black | — |
| FCS-81 | — | — | — | Black | — |

## 4.3. Ancestry estimation of subsample

To improve the clarity of the isotope data and with the possibility of North African origins for FCS-70 and FCS-81 and potential African ancestry in FCS-09, further craniometric and morphoscopic analyses were conducted (table 3; electronic supplementary material, figures S1–S3). Further impetus was provided by the presence of a bipartite inca bone in the occipital bone of FCS-70. Inca bones are more prevalent in modern sub-Saharan African and East Asian populations and are rare in European communities [136]. Craniometric analysis via Fordisc 3.1 was possible for FCS-09 and FCS-70. FCS-09 was closest to the Hispanic male population in the Forensic Databank sample, with a posterior probability of 0.611 and a Chi-square typicality of 0.823. FCS-70 was closest to White males, with a posterior probability of 0.640 and a Chi-square typicality of 0.352. Although Jantz & Ousley [98] consider any typicality above 0.05 to be valid, another archaeological study which used Fordisc [92] removed samples with typicality below 0.7 from the final assessment. Due to cranial incompleteness, for FCS-70 and FCS-81, the only morphoscopic analysis that was possible was the Decision Tree [99]. Both FCS-70 and FCS-81 were classified as 'Black.' FCS-09 was analysed using the Decision Tree and OSSA [99]. In both tests, FCS-09 was classified as 'Black'. The overall classification accuracies for the Decision Tree (for Black individuals) and OSSA (for Black and White individuals) methods are 91% and 86.1%, respectively. As highlighted above, applying forensic ancestry estimation methods to archaeological remains is problematic. We have therefore taken a gestalt approach to the interpretation of the ancestry estimation results, isotope results, artefactual evidence and historical context, with the aim of identifying phenotypic geographic diversity, not culturally defined categories [105,106,137]. These interpretations are made in consideration of available data, but may be revised in future studies with the addition of other forms of evidence used in studies of population affinity, such as ancient DNA (aDNA) analysis, post-cranial metrics and dental morphology [138].

Due to exceptional cranial preservation, FCS-09 was analysed using all available methods. The isotope values from FCS-09 are entirely consistent with British origins and suggestive of being raised in coastal southwest Britain. The difference in ancestry estimates between the metric and morphoscopic methods are not incompatible when considered in the context of a deep medieval history between North Africa and the Iberian Peninsula, and are reified with the knowledge of how the 'Hispanic' ethnic category came to be. We therefore support the assignment that FCS-09 has African ancestry but suggest that he grew up in southwest Britain. This finding adds to the growing evidence for Black Africans in professional roles among the Tudor population [16–21]. Most records referring to African residents are from parishes near port towns, such as London, Southampton, Bristol and Plymouth [20], which may have attracted people outside of Britain for trade. FCS-09 may well have been raised in or near a port town in the southwest of England and represents the first direct evidence for mariners of African descent in the navy of Henry VIII.

The evidence from FCS-70 and FCS-81 is less certain. The craniometric ancestry estimate for FCS-70 should be approached with caution given the overall low typicality, which suggests FCS-70 is dissimilar to all three populations, or potentially of diverse heritage. However, this may be because fewer measurements were possible due to poorer preservation than FCS-09. This contrasts with morphoscopic analysis, in which the Decision Tree classified FCS-70 along the same 'branch' as FCS-09, resulting in an estimate of 'Black'. FCS-81 was also estimated as 'Black' using the Decision Tree but followed a different 'branch' from FCS-70 and FCS-09 (electronic supplementary material, figure S3). Furthermore, it was only possible to analyse FCS-81 using one ancestry estimation method, so the results are not as strong. The $\delta^{34}S$ values of both FCS-70 and FCS-81 suggest inland environments and both have high $\delta^{18}O_p$ values, with FCS-70 being higher than is typical for Europe. Therefore, based on the morphoscopic ancestry estimate and the isotope evidence, origins in inland North Africa cannot be discounted for FCS-70, with Iberia also being possible. The isotopic and artefactual evidence for

FCS-81 has been discussed above as being consistent with inland Iberia or possibly inland Italy. Although the ancestry evidence for FCS-81 is limited, it is important to recognize that African ancestry is still possible, even if FCS-81 was not raised in Africa. Historical evidence for 'Iberian Moors' is vast, and we know that the African diver Jacques Francis was a member of a Venetian household, attesting to the presence of African individuals in Italy at this time [19]. Although small samples make ancestry estimation of past populations more difficult due to greater statistical uncertainty, this analysis contributed key evidence for diversity in the Tudor navy and will be expanded in future studies.

# 5. Conclusion

There has been extensive historical research on the Tudor period, yet direct analyses of its population are lacking. The multi-factorial approach taken by this study has combined artefactual and historical evidence with multi-isotope and ancestry estimation analyses to gain a greater insight into eight members of the *Mary Rose* crew. Our findings point to the important contributions that individuals of diverse backgrounds and origins made to the English navy. This adds to the ever-growing body of evidence for diversity in geographic origins, ancestry and lived experiences in Tudor England. Although isotope analysis is a useful tool for investigating provenance, it should be used with caution and suggested interpretations in this research are supported by artefactual evidence. As drivers of isotope variation become better understood and biosphere maps become higher resolution, the confidence with which origins can be reconstructed will improve [139]. As such, these interpretations may require future refinement. Future isotopic (e.g. [140]), ancestry and aDNA analyses on more of the human remains of the *Mary Rose* will result in a more complete narrative of both Henry VIII's navy and the wider Tudor world.

Data accessibility. The datasets supporting this article have been included within the manuscript and uploaded as part of the electronic supplementary material.

Authors' contributions. J.S. and R.M. designed and instigated the research project. K.E.F. designed and delivered the ancestry estimation analysis. J.S. and R.M. sampled the remains, and J.S. prepared all enamel, bone and dentine samples for analysis. J.S. led the production of the manuscript, assisted by R.M. and K.E.F. All authors read, contributed and approved the manuscript. A.J.N. ($\delta^{18}O$, $\delta^{13}C$, $\delta^{15}N$), A.L.L. ($\delta^{34}S$), M.-A.M. and M.B.A. ($^{87}Sr/^{86}Sr$) undertook mass spectrometry. A.H. provided curatorial support, access and wider information on the *Mary Rose* throughout the project.

Competing interests. There are no competing interests.

Funding. The isotope analysis was funded by the Cardiff University Early Career research fund to R.M. and FORDISC software was purchased by Avanti Media.

Acknowledgements. We are grateful to the Mary Rose Trust for their support for this project and granting permission to undertake sampling of the human remains in their care. We also owe thanks to Nick Owen (Swansea University), Sam Robson and Garry Scarlett (University of Portsmouth) for discussions relating to the *Mary Rose* 'characters', and thanks to Rebecca Redfern (Museum of London) for comments on an earlier draft of this paper. We are also grateful to the Oxford Archaeology South Heritage Burials Team and Ceri Boston for providing access to unpublished osteological data, and to Ian Brewer for discussions surrounding southern European lithologies. A.L.L. would like to acknowledge Christopher Brodie, Thermo Fisher, for instrumentation support. We are also grateful to three anonymous reviewers whose comments have made a marked improvement to the paper.

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
