## [Peer Review File · Royal Society Open Science]

Review History

RSOS-202106.R0 (Original submission)

Review form: Reviewer 1

Is the manuscript scientifically sound in its present form?

Yes

Are the interpretations and conclusions justified by the results?

Yes

Is the language acceptable?

Yes

Do you have any ethical concerns with this paper?

No

Have you any concerns about statistical analyses in this paper?

No

Recommendation?

Accept with minor revision (please list in comments)

Comments to the Author(s)

Overall this is an excellent manuscript based on a very sound study. The authors applied multiple isotope analysis (strontium and oxygen of dental enamel; sulfur, carbon and nitrogen of dentine collagen/+1 rib sample) to construct fairly detailed, yet admittedly incomplete, osteobiographies of eight individuals from the famous shipwreck of the Mary Rose. In its current form the manuscript fulfills the main criteria for publication in this journal (Articles should report work that is scientifically sound, in which the methodology is rigorous and the conclusions are fully supported by the data.) and as such is certainly suitable for publication. I was also very impressed by the clear structure and writing, and particularly how the authors were able to carefully yet successfully incorporate the ancestry estimation results into the broader study. On this point, the way that this potentially problematic data type is used, and the further clarification provided in the supplementary document, provides a useful template for future studies attempting to integrate isotopic data and osteological data sets with ancestry estimations. The authors generally struck a good balance between over- and under-interpretation. The study is also especially important for highlighting the presence of non-Europeans in the British/European archaeological record of the late mediaeval/early modern period.

There are two main critiques of the ms, in its current, that if properly addressed would greatly improve this manuscript.

1) While the authors briefly mention the previous study of Bell et al. (2009) and the subsequent archaeological debate that this initiated (Millard & Schroeder 2010; Bell et al. 2010) there is no explicit effort made to engage in this debate nor even to compare the data in this study with that originally produced by Bell and colleagues. It is unclear why this is the case, and understandable that the authors clearly wish to emphasize their own new data. Nonetheless, since the original study by Bell et al. included some of the same isotope systems (namely, oxygen, carbon and nitrogen) applied to the same exact skeleton collection, the complete lack of engagement with, or direct comparison to this earlier study is problematic. If the authors have reason to doubt the reliability of the data reported by Bell and colleagues, then they are obligated to be explicit about this. If not, then it really is necessary to make a clear comparison between the isotope data from Bell et al. (2009) and the isotope data reported herein (e.g. how do they compare? do the ranges overlap? are the results consistent with the previous study or differ significantly?, etc...). The data from both studies should also be plotted together in one or more figures to allow the readers to directly compare and interpret the data themselves.

Also, a more explicit comment on earlier examples of African individuals recovered from (Roman) British archaeological contexts is perhaps merited (Leach et al. 2010).

2) As is the case with most isotopic provenance studies, the approach used is (by necessity) exclusionary. However, the manner of presenting these types of results (rather long descriptions listing all of the places where each individual could NOT have originated from for each isotope proxy) is less than ideal. In most geographic contexts there is no other option than to present the results this way. However, Britain is the most intensively and extensively mapped areas of the world for most isotope systems, and one of the few regions where high quality isoscapes exist for both strontium and oxygen isotopes (e.g. Evans et al. 2012; Pelligrini et al. 2016; see also the British isotope domains dataset and online tool at <https://www.bgs.ac.uk/datasets/biosphere-isotope-domains-gb/>). As such, it is somewhat striking that this study makes no attempt at a more systematic, quantitative, spatial approach to interpreting and presenting the isotopic provenance data. Such an approach would greatly improve the visualization, interpretation, and presentation of the isotope results by simply and effectively illustrating the areas of potential origin for each individual (or at least the British ones). Such an approach is not very complex and can be accomplished with a fairly simple application in ArcGIS, as demonstrated recently by a similar

study combining skeletal isotope data and isoscapes (in a British context!) to trace the origins of individuals buried at Stonehenge (Snoeck et al. 2018).

Based on the concerns detailed above, I recommend that the manuscript be accepted with minor revisions. I hope that the authors seriously consider the proposed suggestions for revision, and look forward to reading the revised and published version of this paper in Royal Society Open Science.

Review form: Reviewer 2 (Olaf Nehlich)

Is the manuscript scientifically sound in its present form?

Yes

Are the interpretations and conclusions justified by the results?

Yes

Is the language acceptable?

Yes

Do you have any ethical concerns with this paper?

No

Have you any concerns about statistical analyses in this paper?

Yes

Recommendation?

Accept with minor revision (please list in comments)

Comments to the Author(s)

The manuscript by Scorrer and colleagues presents a study on the mediaeval warship Mary Rose. The manuscript attempts an isotope study in order to understand the origins of the crew to the ship. The article is a well written summary and the study scientifically sound, so that the whole manuscript gives a detailed insight into the crews diet and possible origins. However, there are some flaws in the study's design, since the team analysed carbon, nitrogen, sulphur, oxygen and strontium isotopes of eight individuals the sample numbers are very low and discrepancies in the data result in highly biased data. Therefore, there are no strong conclusions and the article is adding another layer of information without revealing origins or deeper understanding of the Tudor's warship. In addition the authors decided to include a fully unrelated craniometric ancestry estimation into the article, which fails to connect to the other results and is only another mosaic puzzle piece in the story, which does not fit with other results.

General remarks:

The data table 1 should be merged with table S1 since they both contain valuable information and only together these information are intuitive assessable. Additionally the table description needs to be more substantial.

The Figures need a higher resolution, Figures 1,3, and 4 need a higher resolution and better quality, my printouts were bad and even on screen were no distinctions between symbols.

The introduction to the Tudor kingdom and the mobility of the navy is excellent. It is well written, substantial yet not lengthy. The isotope background is alright, but lacks some

introduction to isotope data from the studied period. There is a multitude of data published and an overview of available data would have been a good start for understanding the setting.

The study design is alright, given the importance and value of the samples, however, seven individuals are very few and therefore the robustness of such data will be limited. This is the major concern of the study, because the results will be only trustworthy, if there are no or limited numbers of outliers, but due to the nature of the historic background we would expect quite a number of outliers. This means there will be no background data for proper interpretation.

Therefore, the general literature review of isotopic mobility studies is most important.

The sample treatment and analytical details are highly detailed and should be cut to a minimum, additionally many of these methods have been published before and these should be acknowledged.

The ancestry estimation by craniometric and morphoscopic methods should be excluded from the manuscript. Though valid to a certain point the sample number and variability in the skeletal remains are quite big and the interpretation results more in speculations than scientific conclusions. In my opinion the interpretation could be added in a side note, but not a full chapter. The conclusion are just a summary and are relativizing the own data. In my opinion the results are only limited and this needs to be addressed. Additionally the authors have started a data set for the Mary Rose material which needs to be expanded in put into the larger context.

Minor comments:

The collagen was not ultrafiltered, this is usually sufficient for carbon and nitrogen isotope results, however, for sulphur isotopic results this could be problematic. Additionally the salt water could have compromised the materials and therefore an additional ultrafiltration step seems wise. My concern is related to the correlation of sulphur isotope values with sulphur content in collagen. The highest sulphur isotope values also revealed the highest sulphur contents. This could be indicative for seawater sulphate intrusion, therefore these data should be questioned and double checked.

The strontium results are nice, but the interpretation and presentation lacks ambition. I would recommend to use literature data for comparison and additional arguments. In my opinion these data have been neglected in the interpretation. Similarly the oxygen isotope data, which are more problematic, but in itself have some value, which needs to be addressed.

The suggested changes are not to major in my opinion and therefore should be addressed and the manuscript altered accordingly. After doing so, in my regards to the high quality of the manuscript's style there is no issue with publication.

Review form: Reviewer 3

Is the manuscript scientifically sound in its present form?

Yes

Are the interpretations and conclusions justified by the results?

Yes

Is the language acceptable?

Yes

Do you have any ethical concerns with this paper?

No

Have you any concerns about statistical analyses in this paper?

Yes

Recommendation?

Accept with minor revision (please list in comments)

Comments to the Author(s)

Scorrer et al. have prepared a well-written manuscript that succinctly presents and very adequately interprets multi-isotope and morphometric skeletal data from a relatively small (but well selected) sample of humans from the Mary Rose wreck. This paper is an important contribution to the growing literature involving these datasets, and more specifically helps to add additional scientific value to our understanding about the life-ways (i.e. origins) of individuals from the Tudor time period. I have added specific comments/corrections to the attached .pdf (Appendix A) of the manuscript, and mention a few of these items here.

There are no major concerns with the publication of this manuscript providing these minor corrections made and/or considered to help substantiate the interpretation of the isotope data. The authors have been particularly thorough in their realistic evaluation of the morphometric determination using the existing software and databases available for this purpose (e.g. Fordisc), which are not specific to archaeological populations. The additional information on this part of the study contained in the supplemental was much appreciated and necessary.

Finally, the balance between offering specific (likely) geographic origins for each of the individuals sampled with a recognition of equifinality inherent in individual, or combined, isotope systems ($\delta^{13}\text{C}$, $\delta^{15}\text{N}$, $\delta^{18}\text{O}$, $\delta^{34}\text{S}$, and $87\text{Sr}/86\text{Sr}$) is reasonably done. I have suggested statistical treatment of the isotope data should be attempted/worked through via appropriate non-parametric methods to help further support the author's suggestion of origins within or outside 'Britain'. Given the additional information available on each of the individuals that are part of this study, one could consider a possible bayesian approach to determining 'local' versus 'non-local' in this context.

Overall I enjoyed the paper and how the authors have approached this research, and I look forward to seeing it published in its revised format.

Decision letter (RSOS-202106.R0)

Dear Ms Scorrer

On behalf of the Editors, we are pleased to inform you that your Manuscript RSOS-202106 "Diversity aboard a Tudor warship: Investigating the origins of the Mary Rose crew using multi-isotope analysis" has been accepted for publication in Royal Society Open Science subject to minor revision in accordance with the referees' reports. Please find the referees' comments along with any feedback from the Editors below my signature.

We invite you to respond to the comments and revise your manuscript. Below the referees' and Editors' comments (where applicable) we provide additional requirements. Final acceptance of

your manuscript is dependent on these requirements being met. We provide guidance below to help you prepare your revision.

Please submit your revised manuscript and required files (see below) no later than 7 days from today's (ie 01-Mar-2021) date. Note: the ScholarOne system will 'lock' if submission of the revision is attempted 7 or more days after the deadline. If you do not think you will be able to meet this deadline please contact the editorial office immediately.

on behalf of Professor Matthew Collins (Associate Editor) and Kevin Padian (Subject Editor)
openscience@royalsociety.org

Associate Editor Comments to Author (Professor Matthew Collins):
Comments to the Author:

Thank your for your MS. All three reviewers agree that this is a fascinating paper which should be published and all suggesting only minor changes.

I how that you would agree that the full data associated with the $87\text{Sr}/86\text{Sr}$ analyses should be included (e.g. in the supplementary section. These data should have 88Sr (V), 85Rb (V), $87\text{Sr}/86\text{Sr}$, $87\text{Sr}/86\text{Sr}$ (1 standard error). Given the methods described in the manuscript for the determination of the $87\text{Sr}/86\text{Sr}$ data. It is best practice to include all the QC data generated with these analyses.

Reviewer #1

Overall this is an excellent manuscript based on a very sound study. The authors applied multiple isotope analysis (strontium and oxygen of dental enamel; sulfur, carbon and nitrogen of dentin collagen/ +1 rib sample) to construct fairly detailed, yet admittedly incomplete, osteobiographies of eight individuals from the famous shipwreck of the Mary Rose. In its current form the manuscript fulfills the main criteria for publication in this journal (Articles should report work that is scientifically sound, in which the methodology is rigorous and the conclusions are fully supported by the data.) and as such is certainly suitable for publication. I was also very impressed by the clear structure and writing, and particularly how the authors were able to carefully yet successfully incorporate the ancestry estimation results into the broader study. On this point, the way that this potentially problematic data type is used, and the further clarification provided in the supplementary document, provides a useful template for future studies attempting to integrate isotopic data and osteological data sets with ancestry estimations. The authors generally struck a good balance between over- and under-interpretation. The study is

also especially important for highlighting the presence of non-Europeans in the British/European archaeological record of the late mediaeval/early modern period.

There are two main critiques of the ms, in its current, that if properly addressed would greatly improve this manuscript.

1) While the authors briefly mention the previous study of Bell et al. (2009) and the subsequent archaeological debate that this initiated (Millard & Schroeder 2010; Bell et al. 2010) there is no explicit effort made to engage in this debate nor even to compare the data in this study with that originally produced by Bell and colleagues. It is unclear why this is the case, and understandable that the authors clearly wish to emphasize their own new data. Nonetheless, since the original study by Bell et al. included some of the same isotope systems (namely, oxygen, carbon and nitrogen) applied to the same exact skeleton collection, the complete lack of engagement with, or direct comparison to this earlier study is problematic. If the authors have reason to doubt the reliability of the data reported by Bell and colleagues, then they are obligated to be explicit about this. If not, then it really is necessary to make a clear comparison between the isotope data from Bell et al. (2009) and the isotope data reported herein (e.g. how do they compare? do the ranges overlap? are the results consistent with the previous study or differ significantly?, etc...). The data from both studies should also be plotted together in one or more figures to allow the readers to directly compare and interpret the data themselves.

Also, a more explicit comment on earlier examples of African individuals recovered from (Roman) British archaeological contexts is perhaps merited (Leach et al. 2010).

2) As is the case with most isotopic provenance studies, the approach used is (by necessity) exclusionary. However, the manner of presenting these types of results (rather long descriptions listing all of the places where each individual could NOT have originated from for each isotope proxy) is less than ideal. In most geographic contexts there is no other option than to present the results this way. However, Britain is the most intensively and extensively mapped areas of the world for most isotope systems, and one of the few regions where high quality isoscapes exist for both strontium and oxygen isotopes (e.g. Evans et al. 2012; Pelligrini et al. 2016; see also the British isotope domains dataset and online tool at <https://www.bgs.ac.uk/datasets/biosphere-isotope-domains-gb/>). As such, it is somewhat striking that this study makes no attempt at a more systematic, quantitative, spatial approach to interpreting and presenting the isotopic provenance data. Such an approach would greatly improve the visualization, interpretation, and presentation of the isotope results by simply and effectively illustrating the areas of potential origin for each individual (or at least the British ones). Such an approach is not very complex and can be accomplished with a fairly simple application in ArcGIS, as demonstrated recently by a similar study combining skeletal isotope data and isoscapes (in a British context!) to trace the origins of individuals buried at Stonehenge (Snoeck et al. 2018).

Based on the concerns detailed above, I recommend that the manuscript be accepted with minor revisions. I hope that the authors seriously consider the proposed suggestions for revision, and look forward to reading the revised and published version of this paper in Royal Society Open Science.

Reviewer #2

The manuscript by Scorer and colleagues presents a study on the medieval warship Mary Rose. The manuscript attempts an isotope study in order to understand the origins of the crew to the ship. The article is a well written summary and the study scientifically sound, so that the whole manuscript gives a detailed insight into the crews diet and possible origins. However, there are some flaws in the study's design, since the team analysed carbon, nitrogen, sulphur, oxygen and strontium isotopes of eight individuals the sample numbers are very low and discrepancies in the data result in highly biased data. Therefore, there are no strong conclusions and the article is adding another layer of information without revealing origins or deeper understanding of the

Tudor's warship. In addition the authors decided to include a fully unrelated craniometric ancestry estimation into the article, which fails to connect to the other results and is only another mosaic puzzle piece in the story, which does not fit with other results.

General remarks:

The data table 1 should be merged with table S1 since they both contain valuable information and only together these information are intuitive assessable. Additionally the table description needs to be more substantial.

The Figures need a higher resolution, Figures 1,3, and 4 need a higher resolution and better quality, my printouts were bad and even on screen were no distinctions between symbols.

The introduction to the Tudor kingdom and the mobility of the navy is excellent. It is well written, substantial yet not lengthy. The isotope background is alright, but lacks some introduction to isotope data from the studied period. There is a multitude of data published and an overview of available data would have been a good start for understanding the setting.

The study design is alright, given the importance and value of the samples, however, seven individuals are very few and therefore the robustness of such data will be limited. This is the major concern of the study, because the results will be only trustworthy, if there are no or limited numbers of outliers, but due to the nature of the historic background we would expect quite a number of outliers. This means there will be no background data for proper interpretation.

Therefore, the general literature review of isotopic mobility studies is most important.

The sample treatment and analytical details are highly detailed and should be cut to a minimum, additionally many of these methods have been published before and these should be acknowledged.

The ancestry estimation by craniometric and morphoscopic methods should be excluded from the manuscript. Though valid to a certain point the sample number and variability in the skeletal remains are quite big and the interpretation results more in speculations than scientific conclusions. In my opinion the interpretation could be added in a side note, but not a full chapter. The conclusion are just a summary and are relativizing the own data. In my opinion the results are only limited and this needs to be addressed. Additionally the authors have started a data set for the Mary Rose material which needs to be expanded in put into the larger context.

Minor comments:

The collagen was not ultrafiltered, this is usually sufficient for carbon and nitrogen isotope results, however, for sulphur isotopic results this could be problematic. Additionally the salt water could have compromised the materials and therefore an additional ultrafiltration step seems wise. My concern is related to the correlation of sulphur isotope values with sulphur content in collagen. The highest sulphur isotope values also revealed the highest sulphur contents. This could be indicative for seawater sulphate intrusion, therefore these data should be questioned and double checked.

The strontium results are nice, but the interpretation and presentation lacks ambition. I would recommend to use literature data for comparison and additional arguments. In my opinion these data have been neglected in the interpretation. Similarly the oxygen isotope data, which are more problematic, but in itself have some value, which needs to be addressed.

The suggested changes are not to major in my opinion and therefore should be addressed and the manuscript altered accordingly. After doing so, in my regards to the high quality of the manuscript's style there is no issue with publication.

Reviewer #3

Scorer et al. have prepared a well-written manuscript that succinctly presents and very adequately interprets multi-isotope and morphometric skeletal data from a relatively small (but well selected) sample of humans from the Mary Rose wreck. This paper is an important

contribution to the growing literature involving these datasets, and more specifically helps to add additional scientific value to our understanding about the life-ways (i.e. origins) of individuals from the Tudor time period. I have added specific comments/corrections to the attached .pdf of the manuscript, and mention a few of these items here.

There are no major concerns with the publication of this manuscript providing these minor corrections made and/or considered to help substantiate the interpretation of the isotope data. The authors have been particularly thorough in their realistic evaluation of the morphometric determination using the existing software and databases available for this purpose (e.g. Fordisc), which are not specific to archaeological populations. The additional information on this part of the study contained in the supplemental was much appreciated and necessary.

Finally, the balance between offering specific (likely) geographic origins for each of the individuals sampled with a recognition of equifinality inherent in individual, or combined, isotope systems ($\delta^{13}\text{C}$, $\delta^{15}\text{N}$, $\delta^{18}\text{O}$, $\delta^{34}\text{S}$, and $87\text{Sr}/86\text{Sr}$) is reasonably done. I have suggested statistical treatment of the isotope data should be attempted/worked through via appropriate non-parametric methods to help further support the author's suggestion of origins within or outside 'Britain'. Given the additional information available on each of the individuals that are part of this study, one could consider a possible bayesian approach to determining 'local' versus 'non-local' in this context.

Overall I enjoyed the paper and how the authors have approached this research, and I look forward to seeing it published in its revised format.

Reviewer comments to Author:

Reviewer: 1

Comments to the Author(s)

Overall this is an excellent manuscript based on a very sound study. The authors applied multiple isotope analysis (strontium and oxygen of dental enamel; sulfur, carbon and nitrogen of dentine collagen/+1 rib sample) to construct fairly detailed, yet admittedly incomplete, osteobiographies of eight individuals from the famous shipwreck of the Mary Rose. In its current form the manuscript fulfills the main criteria for publication in this journal (Articles should report work that is scientifically sound, in which the methodology is rigorous and the conclusions are fully supported by the data.) and as such is certainly suitable for publication. I was also very impressed by the clear structure and writing, and particularly how the authors were able to carefully yet successfully incorporate the ancestry estimation results into the broader study. On this point, the way that this potentially problematic data type is used, and the further clarification provided in the supplementary document, provides a useful template for future studies attempting to integrate isotopic data and osteological data sets with ancestry estimations. The authors generally struck a good balance between over- and under-interpretation. The study is also especially important for highlighting the presence of non-Europeans in the British/European archaeological record of the late mediaeval/early modern period.

There are two main critiques of the ms, in its current, that if properly addressed would greatly improve this manuscript.

1) While the authors briefly mention the previous study of Bell et al. (2009) and the subsequent archaeological debate that this initiated (Millard & Schroeder 2010; Bell et al. 2010) there is no explicit effort made to engage in this debate nor even to compare the data in this study with that originally produced by Bell and colleagues. It is unclear why this is the case, and understandable that the authors clearly wish to emphasize their own new data. Nonetheless, since the original study by Bell et al. included some of the same isotope systems (namely, oxygen, carbon and nitrogen) applied to the same exact skeleton collection, the complete lack of engagement with, or

direct comparison to this earlier study is problematic. If the authors have reason to doubt the reliability of the data reported by Bell and colleagues, then they are obligated to be explicit about this. If not, then it really is necessary to make a clear comparison between the isotope data from Bell et al. (2009) and the isotope data reported herein (e.g. how do they compare? do the ranges overlap? are the results consistent with the previous study or differ significantly?, etc...). The data from both studies should also be plotted together in one or more figures to allow the readers to directly compare and interpret the data themselves.

Also, a more explicit comment on earlier examples of African individuals recovered from (Roman) British archaeological contexts is perhaps merited (Leach et al. 2010).

2) As is the case with most isotopic provenance studies, the approach used is (by necessity) exclusionary. However, the manner of presenting these types of results (rather long descriptions listing all of the places where each individual could NOT have originated from for each isotope proxy) is less than ideal. In most geographic contexts there is no other option than to present the results this way. However, Britain is the most intensively and extensively mapped areas of the world for most isotope systems, and one of the few regions where high quality isoscapes exist for both strontium and oxygen isotopes (e.g. Evans et al. 2012; Pelligrini et al. 2016; see also the British isotope domains dataset and online tool at <https://www.bgs.ac.uk/datasets/biosphere-isotope-domains-gb/>). As such, it is somewhat striking that this study makes no attempt at a more systematic, quantitative, spatial approach to interpreting and presenting the isotopic provenance data. Such an approach would greatly improve the visualization, interpretation, and presentation of the isotope results by simply and effectively illustrating the areas of potential origin for each individual (or at least the British ones). Such an approach is not very complex and can be accomplished with a fairly simple application in ArcGIS, as demonstrated recently by a similar study combining skeletal isotope data and isoscapes (in a British context!) to trace the origins of individuals buried at Stonehenge (Snoeck et al. 2018).

Based on the concerns detailed above, I recommend that the manuscript be accepted with minor revisions. I hope that the authors seriously consider the proposed suggestions for revision, and look forward to reading the revised and published version of this paper in Royal Society Open Science.

Reviewer: 2

Comments to the Author(s)

The manuscript by Scorrer and colleagues presents a study on the mediaeval warship Mary Rose. The manuscript attempts an isotope study in order to understand the origins of the crew to the ship. The article is a well written summary and the study scientifically sound, so that the whole manuscript gives a detailed insight into the crews diet and possible origins. However, there are some flaws in the study's design, since the team analysed carbon, nitrogen, sulphur, oxygen and strontium isotopes of eight individuals the sample numbers are very low and discrepancies in the data result in highly biased data. Therefore, there are no strong conclusions and the article is adding another layer of information without revealing origins or deeper understanding of the Tudor's warship. In addition the authors decided to include a fully unrelated craniometric ancestry estimation into the article, which fails to connect to the other results and is only another mosaic puzzle piece in the story, which does not fit with other results.

General remarks:

The data table 1 should be merged with table S1 since they both contain valuable information and only together these information are intuitive assessable. Additionally the table description needs to be more substantial.

The Figures need a higher resolution, Figures 1,3, and 4 need a higher resolution and better quality, my printouts were bad and even on screen were no distinctions between symbols.

The introduction to the Tudor kingdom and the mobility of the navy is excellent. It is well written, substantial yet not lengthy. The isotope background is alright, but lacks some introduction to isotope data from the studied period. There is a multitude of data published and an overview of available data would have been a good start for understanding the setting.

The study design is alright, given the importance and value of the samples, however, seven individuals are very few and therefore the robustness of such data will be limited. This is the major concern of the study, because the results will be only trustworthy, if there are no or limited numbers of outliers, but due to the nature of the historic background we would expect quite a number of outliers. This means there will be no background data for proper interpretation.

Therefore, the general literature review of isotopic mobility studies is most important.

The sample treatment and analytical details are highly detailed and should be cut to a minimum, additionally many of these methods have been published before and these should be acknowledged.

The ancestry estimation by craniometric and morphoscopic methods should be excluded from the manuscript. Though valid to a certain point the sample number and variability in the skeletal remains are quite big and the interpretation results more in speculations than scientific conclusions. In my opinion the interpretation could be added in a side note, but not a full chapter.

The conclusion are just a summary and are relativizing the own data. In my opinion the results are only limited and this needs to be addressed. Additionally the authors have started a data set for the Mary Rose material which needs to be expanded in put into the larger context.

Minor comments:

The collagen was not ultrafiltered, this is usually sufficient for carbon and nitrogen isotope results, however, for sulphur isotopic results this could be problematic. Additionally the salt water could have compromised the materials and therefore an additional ultrafiltration step seems wise. My concern is related to the correlation of sulphur isotope values with sulphur content in collagen. The highest sulphur isotope values also revealed the highest sulphur contents. This could be indicative for seawater sulphate intrusion, therefore these data should be questioned and double checked.

The strontium results are nice, but the interpretation and presentation lacks ambition. I would recommend to use literature data for comparison and additional arguments. In my opinion these data have been neglected in the interpretation. Similarly the oxygen isotope data, which are more problematic, but in itself have some value, which needs to be addressed.

The suggested changes are not to major in my opinion and therefore should be addressed and the manuscript altered accordingly. After doing so, in my regards to the high quality of the manuscript's style there is no issue with publication.

Reviewer: 3

Comments to the Author(s)

Scorrer et al. have prepared a well-written manuscript that succinctly presents and very adequately interprets multi-isotope and morphometric skeletal data from a relatively small (but well selected) sample of humans from the Mary Rose wreck. This paper is an important contribution to the growing literature involving these datasets, and more specifically helps to add additional scientific value to our understanding about the life-ways (i.e. origins) of individuals from the Tudor time period. I have added specific comments/corrections to the attached .pdf of the manuscript, and mention a few of these items here.

There are no major concerns with the publication of this manuscript providing these minor corrections made and/or considered to help substantiate the interpretation of the isotope data. The authors have been particularly thorough in their realistic evaluation of the morphometric

determination using the existing software and databases available for this purpose (e.g. Fordisc), which are not specific to archaeological populations. The additional information on this part of the study contained in the supplemental was much appreciated and necessary.

Finally, the balance between offering specific (likely) geographic origins for each of the individuals sampled with a recognition of equifinality inherent in individual, or combined, isotope systems ($\delta^{13}\text{C}$, $\delta^{15}\text{N}$, $\delta^{18}\text{O}$, $\delta^{34}\text{S}$, and $87\text{Sr}/86\text{Sr}$) is reasonably done. I have suggested statistical treatment of the isotope data should be attempted/worked through via appropriate non-parametric methods to help further support the author's suggestion of origins within or outside 'Britain'. Given the additional information available on each of the individuals that are part of this study, one could consider a possible bayesian approach to determining 'local' versus 'non-local' in this context.

Overall I enjoyed the paper and how the authors have approached this research, and I look forward to seeing it published in its revised format.

===PREPARING YOUR MANUSCRIPT===

===PREPARING YOUR REVISION IN SCHOLARONE===

Author's Response to Decision Letter for (RSOS-202106.R0)

See Appendix B.

Decision letter (RSOS-202106.R1)

Dear Ms Scorrer,

I am pleased to inform you that your manuscript entitled "Diversity aboard a Tudor warship: Investigating the origins of the Mary Rose crew using multi-isotope analysis" is now accepted for publication in Royal Society Open Science.

on behalf of Prof Kevin Padian (Subject Editor)
openscience@royalsociety.org

Appendix A**ROYAL SOCIETY
OPEN SCIENCE****Diversity aboard a Tudor warship: Investigating the origins
of the Mary Rose crew using multi-isotope analysis**

Journal:	Royal Society Open Science
Manuscript ID	RSOS-202106
Article Type:	Research
Date Submitted by the Author:	19-Nov-2020
Complete List of Authors:	Scorrer, Jessica; Cardiff University, School of History, Archaeology and Religion Faillace, Katie E.; Cardiff University, School of History, Archaeology and Religion Hildred, Alexzandra; Mary Rose Trust Nederbragt, Alexandra ; Cardiff University, School of Earth and Ocean Sciences Andersen, Morten; Cardiff University, School of Earth and Ocean Sciences Millet, Marc-Alban; Cardiff University, School of Earth and Ocean Sciences Lamb , Angela; National Environmental Isotope Facility, British Geological Survey Madgwick, Richard; Cardiff University, School of History, Archaeology and Religion
Subject:	environmental science < BIOLOGY
Keywords:	multi-isotope analysis, provenancing, diet, ancestry estimation, Tudor Britain, late medieval/early modern
Subject Category:	Organismal and Evolutionary Biology

Author-supplied statements

Relevant information will appear here if provided.

Ethics

Does your article include research that required ethical approval or permits?:

This article does not present research with ethical considerations

Statement (if applicable):

CUST_IF_YES_ETHICS :No data available.

Data

It is a condition of publication that data, code and materials supporting your paper are made publicly available. Does your paper present new data?:

Yes

Statement (if applicable):

The datasets supporting this article have been included within the manuscript and uploaded as part of the supplementary material.

Conflict of interest

I/We declare we have no competing interests

Statement (if applicable):

CUST_STATE_CONFLICT :No data available.

Authors' contributions

This paper has multiple authors and our individual contributions were as below

Statement (if applicable):

JS and RM designed and instigated the research project. KF designed and delivered the ancestry estimation analysis. JS and RM sampled the remains and JS prepared all enamel, bone and dentine samples for analysis. JS led the production of the manuscript, assisted by RM and KF. All authors read, contributed and approved the manuscript. AN ($\hat{\delta}^{18}O$, $\hat{\delta}^{13}C$, $\hat{\delta}^{15}N$), AL ($\hat{\delta}^{34}S$), MAM and MA ($^{87}Sr/^{86}Sr$) undertook mass spectrometry. AH provided curatorial support, access and wider information on the Mary Rose throughout the project.

Diversity aboard a Tudor warship: Investigating the origins of the *Mary Rose* crew using multi-isotope analysis

Authors: Jessica Scorrer, Katie E. Faillace, Alexzandra Hildred, Alexandra J. Nederbragt, Morten Andersen, Marc-Alban Millet, Angela L. Lamb and Richard Madgwick

Jessica Scorrer, School of History, Archaeology and Religion, Cardiff University, Cardiff CF10 3EU, UK

Katie E. Faillace, School of History, Archaeology and Religion, Cardiff University, Cardiff CF10 3EU, UK

Alexzandra Hildred, Mary Rose Trust, HM Naval Base, Portsmouth PO1 3LX, UK

Alexandra J. Nederbragt, School of Earth and Ocean Sciences, Cardiff University, Cardiff CF10 3AT, UK

Morten Andersen, School of Earth & Ocean Sciences, Cardiff University, Cardiff CF10 3AT, UK

Marc-Alban Millet, School of Earth & Ocean Sciences, Cardiff University, Cardiff CF10 3AT, UK

Angela L. Lamb, National Environmental Isotope Facility, British Geological Survey, Keyworth, Nottinghamshire NG12 5GG, UK

Richard Madgwick*, School of History, Archaeology and Religion, Cardiff University, Cardiff CF10 3EU, UK

Keywords: multi-isotope analysis, provenancing, diet, ancestry estimation, Tudor Britain, late medieval/early modern

***Author for correspondence** (madgwickrd3@cardiff.ac.uk)

Abstract

The great Tudor warship, the *Mary Rose*, which sank tragically in the Solent in 1545 AD, presents a rare archaeological opportunity to research individuals for whom the precise timing and nature of death are known. A long-standing question surrounds the composition of the Tudor navy and whether the crew were largely British or had more diverse origins. This study takes a multi-isotope approach, combining strontium ($^{87}\text{Sr}/^{86}\text{Sr}$), oxygen ($\delta^{18}\text{O}$), sulphur ($\delta^{34}\text{S}$), carbon ($\delta^{13}\text{C}$) and nitrogen ($\delta^{15}\text{N}$) isotope analysis of dental samples to reconstruct the childhood diet and origins of eight of the *Mary Rose* crew. Forensic ancestry estimation was also employed on a subsample. Provenancing isotope data tentatively suggests as many as three of the crew may have originated from warmer, more southerly climates than Britain. Five have isotope values indicative of childhoods spent in western Britain, one of which had cranial morphology suggestive of African ancestry. The general trend of relatively high $\delta^{15}\text{N}$ and low $\delta^{13}\text{C}$ suggests a broadly comparable diet to contemporaneous British and European communities. This multi-isotope approach and the nature of the archaeological context has allowed reconstruction of the biographies of eight Tudor individuals to a higher resolution than is usually possible.

1. Introduction

This study presents new multiple isotope analyses (strontium ($^{87}\text{Sr}/^{86}\text{Sr}$), oxygen ($\delta^{18}\text{O}$), sulphur ($\delta^{34}\text{S}$), carbon ($\delta^{13}\text{C}$) and nitrogen ($\delta^{15}\text{
[revised manuscript text omitted]

55 $\delta^{15}\text{N}$ and $\delta^{18}\text{O}$ isotope analysis on eighteen individuals [9] (*see also* [39, 40]). The $\delta^{13}\text{C}$ and
56 $\delta^{15}\text{N}$ analysis on bone collagen provided comparable results to those at other British medieval
57 sites [28, 29, 41]. Bell *et al.* [9] assert that the $\delta^{18}\text{O}$ results demonstrate that at least one-third,
58 and up to almost two-thirds, of the individuals in their sample originated from a more southerly,
59 or warmer, locale than Britain and make the claim that miscommunication between the multi-
60

national crew contributed to the ship's sinking. This paper was heavily criticised by Millard and Schroeder [39], who argued that only one of the eighteen sampled individuals (Individual 6) is likely to have spent his childhood outside the British Isles, but likely from a more northerly or easterly climatic zone. A reassessment of Bell *et al.*'s [9] data in light of wider work on $\delta^{18}\text{O}$ values in British humans [42, 43] supports Millard and Schroeder's [39] conclusion. The historical and scientific evidence discussed above points to diverse naval recruits within the Tudor fleet, but direct scientific evidence from human remains from this period is very limited.

1.4. Isotope analysis in mobility and dietary studies

Since the 1970s, isotope analysis has been employed by archaeologists to investigate diet, origins and mobility of past humans and animals. Childhood diet and origins are investigated through isotope analysis of dental samples. Dental tissues do not substantially remodel, therefore the isotope composition of primary dentine and enamel represents the formation period during childhood [44].

The $^{87}\text{Sr}/^{86}\text{Sr}$ isotope ratios are incorporated into skeletal tissues from diet and principally relate to the geology of the area where food was produced [45]. As past diets generally comprised locally sourced food, $^{87}\text{Sr}/^{86}\text{Sr}$ ratio analysis is useful for exploring origins. Time-integrated radioactive decay of the rubidium isotope ^{87}Rb to ^{87}Sr , compared to the stable ^{86}Sr isotope, is responsible for the variable $^{87}\text{Sr}/^{86}\text{Sr}$ ratios. Very old and rubidium-rich bedrocks have higher $^{87}\text{Sr}/^{86}\text{Sr}$ ratios, such as granites (up to 0.720 in Britain); whereas younger and low-rubidium bedrocks exhibit lower $^{87}\text{Sr}/^{86}\text{Sr}$ values, for example basalts (approx. 0.705) [46].

The $^{18}\text{O}/^{16}\text{O}$ isotope ratios ($\delta^{18}\text{O}$) are primarily derived from ingested fluids, reflecting local drinking water signals and are therefore useful for investigating origins. Values of $\delta^{18}\text{O}$ vary according to temperature, altitude, latitude, coastal proximity and precipitation and thus relate to particular climatic zones [47]. A global study of human $\delta^{18}\text{O}$ values within archaeological populations by Lightfoot and O'Connell [48] found that the range within most populations is likely to be greater than 3‰. This variation could be due to physiological factors (metabolic isotope fractionation), culturally mediated behaviours (e.g. wine and animal milk consumption), food and drink importation and multiple water sources with different $\delta^{18}\text{O}$ values [49]. Studies have shown that culinary practices such as brewing, boiling and stewing can elevate the $\delta^{18}\text{O}$ values of the water in liquids and food [24, 50, 51]. There are wide-ranging factors that impact on $\delta^{18}\text{O}$ values and therefore it is a complex isotope proxy to interpret, a point returned to in the discussion. However, in this study, comparisons will be made with $\delta^{18}\text{O}$ data from other medieval sites, where the impact of culinary practices is likely to have been similar.

Combined $^{87}\text{Sr}/^{86}\text{Sr}$ and $\delta^{18}\text{O}$ analyses have long been used in parallel to investigate origins (e.g. [52]) but in the last 15 years $\delta^{34}\text{S}$ ($^{34}\text{S}/^{32}\text{S}$ isotope ratio) analysis of bone and dentine collagen has also become useful in both dietary and mobility studies in humans and animals [53-57]. Biosphere $\delta^{34}\text{S}$ has a wide range; relatively low (and negative values are common in inland areas of waterlogged ground and/or impervious lithologies [58]. In contrast, marine resources have relatively high $\delta^{34}\text{S}$ values, and food grown and grazed on coastal soils can be influenced by a 'sea-spray effect' which results in $\delta^{34}\text{S}$ values which approach marine sulphate composition (+20‰) [59], although local geological and environmental factors can also play a role [60]. Therefore, humans and animals feeding at coastal sites generally exhibit higher $\delta^{34}\text{S}$ values than those sourcing their diets from further inland, although estuarial regions also tend

1
2
3 to have high biosphere $\delta^{34}\text{S}$ [61]. Values above +14‰ may indicate origins within 50km of the
4 coast [60], with exposed coasts facing prevailing winds producing the highest values [62].

5
6 Palaeodietary reconstruction using $\delta^{13}\text{C}$ ($^{13}\text{C}/^{12}\text{C}$ isotope ratio) and $\delta^{15}\text{N}$ ($^{15}\text{N}/^{14}\text{N}$ isotope ratio)
7 values is based on the premise that the isotope values in skeletal tissues reflect the average $\delta^{13}\text{C}$
8 and $\delta^{15}\text{N}$ of consumed food (primarily dietary proteins) [47]. The premises behind these
9 methods have been described in detail elsewhere (*see* [47]), so are recounted only briefly (and
10 simplistically) here. Values of $\delta^{13}\text{C}$ principally vary in relation to the consumption of marine
11 resources and different types of plants (C_3 and C_4 photosynthesising plants). Values of $\delta^{15}\text{N}$
12 are closely related to the trophic level of the consumer, with humans consuming more animal
13 protein having higher values. The consumption of fish also tends to result in higher $\delta^{15}\text{N}$ values
14 due to elongated food chains. As an example, humans that consume mostly marine protein are
15 expected to have $\delta^{13}\text{C}$ values up to -12‰ and high $\delta^{15}\text{N}$ values up to 15‰ [63]. However, food
16 produced on manured land will also have higher $\delta^{15}\text{N}$ values (*see* [64]). Dietary composition
17 is not the only driver of $\delta^{13}\text{C}$ and $\delta^{15}\text{N}$ variation. Origins also has an impact, as varied landscape
18 baseline values mean that same foodstuffs will have different values depending on where
they are produced [65]. This grossly oversimplifies the manifold variables that affect $\delta^{13}\text{C}$ and
$\delta^{15}\text{N}$ values, but only a brief summary is provided here, as diet is not a major focus.

More recently, multiple isotope proxies have been employed in combination to investigate
archaeological diet and mobility to increase interpretative potential. However, studies that
integrate five isotope systems, such as this one, remain rare (*see* [53, 56, 66-69]).

2. Materials and methods

2.1. The *Mary Rose* ‘characters’

The *Mary Rose* Trust has attributed professions to the skeletal remains of seven individuals
who are investigated in this project: Cook (FCS-12), Royal Archer (FCS-70), Archer (FCS-
75), Carpenter (FCS-81), Officer (FCS-84), Gentleman (FCS-85) and Purser (FCS-88) (figure
2; electronic supplementary material, table S1). The professions have been ascribed based on
context and artefactual associations by the *Mary Rose* Trust, but are by no means certain as
material could have been displaced [3]. In addition to these characters, an eighth individual
(‘Young Mariner’) was investigated (FCS-09). A photogrammetry study highlighted the
possibility that this individual was potentially of African ancestry [70], which was supported
by further research in this study (see below).

2.2. Sample selection

Enamel samples were extracted from the second molar (M2) for $^{87}\text{Sr}/^{86}\text{Sr}$ and $\delta^{18}\text{O}$ analysis
and dentine was extracted from first molar (M1) roots. Due to the exceptional preservation of
the teeth and the fact that maxillary and mandibular molars should produce similar isotope
ratios [71], teeth/roots were selected based on the ease of sampling to minimise damage.
Bioapatite (in enamel) does not remodel and collagen (in dentine) undergoes very little
remodelling; thus the isotope compositions of teeth represent an archive of childhood diet and
geographic origins [45]. Each tooth provides information from different periods of
childhood [42]. M1 roots and M2 crowns form between 2½ and 9½ years of age, thus sampling
dentine and enamel from these zones provides comparable temporal signals [72] (*but see* [49,
73] for issues relating to temporal resolution). This age bracket should avoid nursing signals,

as historical and isotope studies indicate that weaning in this period was generally complete by 2 years [26, 74]. A whole molar root was sampled for each individual. A c. 4mm strip of enamel was extracted from the lower half (i.e. from the root-enamel junction towards the occlusal surface) of the lingual surface of the M1. The strip was divided into mesial and distal strips, with the larger being analysed for $^{87}\text{Sr}/^{86}\text{Sr}$ and the other for $\delta^{18}\text{O}$.

The duration of incorporation of isotope compositions into dental enamel remains poorly understood, especially in relation to mineralisation [49, 73] and therefore it is not possible to be precise about the temporal range that samples relate to. The same sampling method was followed for all individuals. However, to reduce the impact on display potential, samples were taken from either mandible or maxillae (see electronic supplementary material, table S2). Therefore, temporal ranges should be directly comparable. In addition, all enamel samples cover a period of several years of incorporation, thus minimising any potential seasonal effect on $\delta^{18}\text{O}$ values and to limit issues surrounding intra-individual variation [75]. The youngest remains analysed from the *Mary Rose* were from a 10 year old [3, 6], so this strategy should ensure signals relate to pre-recruitment childhood origins. In addition to the dental samples, a collagen sample from a rib bone (which, unlike dentine, remodels substantially, *see* [76]) was taken from FCS-09 for $\delta^{13}\text{C}$ and $\delta^{15}\text{N}$ isotope analysis to investigate diet in the years before death, prompted by the only instance of severe, bilateral buccal CEJ caries among the individuals recovered from the *Mary Rose*.

2.3. Enamel sample preparation ($^{87}\text{Sr}/^{86}\text{Sr}$ and $\delta^{18}\text{O}$ analysis)

Enamel samples (20-50 mg) were cut from teeth and cleaned using a diamond saw and burr to remove all adhering dentine and $>10\mu\text{m}$ of the enamel surface. Samples for $^{87}\text{Sr}/^{86}\text{Sr}$ (approx. 20-40 mg) and $\delta^{18}\text{O}$ (approx. 2.5-5 mg) analysis were then divided.

For $^{87}\text{Sr}/^{86}\text{Sr}$ ratio analysis, samples were ultrasonically cleaned in deionised water and transferred to a clean working area (class 100, laminar flow) for further preparation and $^{87}\text{Sr}/^{86}\text{Sr}$ analysis at Cardiff Earth Laboratory for Trace Element and Isotope Chemistry (CELTIC). Samples were digested in 8M HNO_3 and heated overnight at 120°C . Strontium extraction from enamel samples used Sr SpecTM resin using a revised version of protocol of Font *et al.* [77]. Samples were loaded into resin columns in 1ml 8M HNO_3 . Matrix elements (including Ca and traces of Rb) were then removed in several washes of 8M HNO_3 before Sr was eluted and collected in 0.05M HNO_3 . Samples were placed on a hotplate (120°C) to dry overnight before this process was repeated for a second pass to remove all traces of remaining Ca. Once dry, purified samples were re-dissolved in 0.3M HNO_3 . The $^{87}\text{Sr}/^{86}\text{Sr}$ ratios were measured using a Nu Plasma II Multi-Collector Inductively Coupled Plasma Mass Spectrometer (MC-ICP-MS) at Cardiff University. Samples were introduced using an Aridus II desolvator introduction system. All data was first corrected for on-peak blank intensities, then mass bias corrected using the exponential law and a normalization ratio of 8.375209 for $^{88}\text{Sr}/^{86}\text{Sr}$ [78]. Residual krypton (Kr) and rubidium (^{87}Rb) interferences were monitored and corrected for using ^{82}Kr and ^{83}Kr ($^{83}\text{Kr}/^{84}\text{Kr} = 0.20175$ and $^{83}\text{Kr}/^{86}\text{Kr} = 0.66474$; without normalization) and ^{85}Rb ($^{85}\text{Rb}/^{87}\text{Rb} = 2.5926$), respectively. Analysis of NIST SRM 987 during the analytical session gave a $^{87}\text{Sr}/^{86}\text{Sr}$ value of 0.710292 ± 0.000007 ($2\sigma_{\text{N}}$, $n=11$) and all data is corrected to NIST SRM 987 values of 0.710248 [79]. Total procedural blanks are typically less than 20pg of Sr, which is negligible relative to the Sr in samples ($>20\text{ng}$). Accuracy was assessed by repeat measurements of $^{87}\text{Sr}/^{86}\text{Sr}$ ratio in NIST SRM 1400 (Bone Ash), giving an average $^{87}\text{Sr}/^{86}\text{Sr}$ ratio of 0.713111 ± 0.000014 ($2\sigma_{\text{N}}$, $n=5$), which is consistent with the published value (0.713126 ± 0.000017 [80]).

For $\delta^{18}\text{O}$ analysis, enamel was powdered using an agate pestle and mortar. The $\delta^{18}\text{O}$ isotope
composition of the structural carbonate within the enamel was measured ($\delta^{18}\text{O}_{\text{carbonate}}$). At the
Cardiff University Stable Isotope Facility, samples were acidified for 5 minutes with $>100\%$
ortho-phosphoric acid at 70°C [81] and analysed in duplicate using a Thermo MAT253 dual
inlet mass spectrometer coupled to a Kiel IV carbonate preparation device. The resultant
isotope values are reported as per mil ($^{18}\text{O}/^{16}\text{O}$) normalized to the VPDB scale using an in-
house carbonate reference material (BCT63) calibrated against NBS19 certified reference
material. The $\delta^{18}\text{O}$ carbonate values are then converted into the VSMOW scale (VSMOW=
$1.0309 \times \delta^{18}\text{O VPDB} + 30.91$ [82]). The long-term reproducibility for $\delta^{18}\text{O}$ BCT63 is ± 0.03
13 per mil (1σ). The $\delta^{18}\text{O}_{\text{carbonate}}$ values were converted to $\delta^{18}\text{O}_{\text{phosphate}}$ ($\delta^{18}\text{O}_p$) values (following
[83]) to allow for comparison with other datasets.

2.4. Dentine and bone collagen sample preparation ($\delta^{13}\text{C}$, $\delta^{15}\text{N}$ and $\delta^{34}\text{S}$ analysis)

Collagen was prepared for $\delta^{13}\text{C}$, $\delta^{15}\text{N}$ and $\delta^{34}\text{S}$ analysis following a modified Longin method
[84]. Approximately 0.4g of root dentine and 1g of rib bone was extracted using a precision
drill with diamond wheel attachment. Collagen was extracted from the whole length of roots
to provide an average for the period of development (see electronic supplementary material,
table S2). Sample surfaces were lightly abraded using a burr to remove contaminants and
$>10\mu\text{m}$ of the cortex. The root and rib samples were then placed in test tubes in 7ml 0.5M HCl,
which was changed weekly until full demineralisation. Each sample was then washed with
deionised water, gelatinised in a pH3 solution at 75°C for 48 hours. The supernatant was then
filtered (8 μm eze-filter, Elkay, Basingstoke), frozen and freeze dried. Samples were weighed
and analysed in duplicate. Ratios of $\delta^{34}\text{S}$, $\delta^{13}\text{C}$ and $\delta^{15}\text{N}$ are reported in per mil (‰) relative to
VCDT, VPDB and AIR standards respectively. The $\delta^{34}\text{S}$ analysis was undertaken at the
National Environmental Isotope Facility, Keyworth, Nottinghamshire. The instrumentation
comprises a ThermoFinnigan EA IsoLink coupled to a Delta V Plus isotope ratio mass
spectrometer via a Conflo IV interface. Values of $\delta^{34}\text{S}$ were calibrated using IAEA S-1 and S-2
with secondary checks comprising in-house standard M1360P (powdered gelatine) with lab
accepted delta value of 2.99‰ and modern cattle bone collagen (lab accepted value of
12.71‰). The average 1σ reproducibility across the standard material was $\pm 0.28\%$. M1360P
was used to calculate %S (0.77% S, calibrated to IAEA S-1 and S-2). The 1σ standard
reproducibility was $\pm 0.25\%$. Both $\delta^{13}\text{C}$ and $\delta^{15}\text{N}$ ratios were measured at the Cardiff
University Stable Isotope Facility using a Flash 1112 Elemental Analyser coupled to
a Thermo Delta V Advantage. Ratios of $\delta^{13}\text{C}$ and $\delta^{15}\text{N}$ were calibrated against caffeine
(laboratory grade, 98.5%, Acros Organics, lot A0342883) and an in-house supermarket
gelatine standard, which is calibrated against IAEA-600 ($\delta^{13}\text{C}$ and $\delta^{15}\text{N}$), IAEA-CH-6 ($\delta^{13}\text{C}$),
and IAEA-N-2 ($\delta^{15}\text{N}$). The long-term reproducibility was ± 0.08 for $\delta^{13}\text{C}$ and ± 0.07 for $\delta^{15}\text{N}$. Caffeine
was used to calculate %N and %C (28.85% N and 49.48% C, calculated from the molecular
formula $\text{C}_8\text{H}_{10}\text{N}_4\text{O}_2$). All samples adhered to published quality control indicators [85].

2.5. Methods of ancestry estimation

During a photogrammetry study of FCS-09 [70], it was suggested that this individual may have
been of African ancestry and additional craniometric and morphological analysis was
conducted to test this. Osteological ancestry estimation is not often applied to archaeological
remains in Britain (*see* [86-88] for exceptions). The authors acknowledge the problematic
history of ancestry estimation stemming from early pursuits of scientific racism [89]. Although
we recognise the methodological and ethical concerns regarding these approaches in forensic

contexts [90], we also agree that not seeking evidence of human diversity and phenotypic
variation in the archaeological record only obscures it further [88], something historians are
actively addressing [16-21]. Ancestry analysis was also performed on FCS-70 and FCS-81,
because of their unusual $\delta^{18}\text{O}$ values and good cranial preservation.

Craniometrics [91] were analysed via Fordisc 3.1 [92], which compares individual samples
with known (recent) skeletal populations. Samples were compared to Black, Hispanic, and
White males from the Forensic Databank. Posterior probabilities, which estimate group affinity
based on relative distance and assumes affinity to one reference group, and typicality
probability (Chi-square), which gives likelihood of group affinity based on absolute distance
of the sample from the group's centroid, are reported alongside group affinity estimates.
Morphoscopic analysis followed Decision Tree and Optimised Summed Scores Attributes
(OSSA) methods [93]. The skeletal populations these methods are based on are also recent.
Therefore, interpretation was conducted with an understanding of the historical, colonial
framework which led to these collections and the ancestry classifications they use [94-96], as
well as evidence for Tudor migration and diversity discussed above. This framing was
particularly important in understanding the designation of 'Hispanic' in a Tudor context. This
classification is used in the USA and refers to individuals with origins or ancestral connections
to Spanish-speaking regions, particularly from the Americas [97, 98]. Applying this linguistic
and social distinction to physical remains is practically and ethically challenging, and requires
an understanding of the colonialist practices in 'Hispanic' regions, which ultimately led to an
admixture of Indigenous populations, (enslaved) African, and European settlers [98]. It is also
important to recognise that the reference populations of Black Americans are the result of
racialisation that follows centuries of enslavement and discrimination which failed to recognise
the substantial phenotypic diversity of individuals with origins or ancestral connections to the
African continent [99, 100]. This cannot be assumed to be the same perception of 'Black' in
Tudor England though its origins can be traced before this period [22].

**3. Isotope results**

**3.1. Provenance isotope results**

The isotope data from $^{87}\text{Sr}/^{86}\text{Sr}$, $\delta^{18}\text{O}$ and $\delta^{34}\text{S}$ analyses are displayed in table 1 with summary
statistics in table 2. All three proxies produced relatively wide-ranging results. The $^{87}\text{Sr}/^{86}\text{Sr}$
values range from 0.70872 to 0.71070 (figure 3), with a mean of 0.70966 (± 0.00072 , 1σ).
Overall, the $\delta^{18}\text{O}_p$ values are high for a British dataset, ranging between 18.4‰ and 21.2‰
(figure 3), with a mean of 19.2‰ ($\pm 1.0‰$, 1σ). The mean is skewed by one clear outlier, FCS-
70 (21.2‰), who has one of the highest values measured in a British man and therefore the
median of 18.9‰ provides a more balanced average. Three individuals (FCS-70, FCS-81 and
FCS-85) have $\delta^{18}\text{O}_p$ values that fall more than two standard deviations outside of the British
mean ($17.7‰ \pm 1.4‰$, 2σ , $n=615$, [42]), the $\delta^{34}\text{S}$ results range widely from -3.3‰ to 17.4‰,
with a mean of 10.5‰ and a large standard deviation ($\pm 8.2‰$, 1σ). Half of the individuals
(FCS-9, FCS-12, FCS-75, FCS-85) have high ($>14‰$) $\delta^{34}\text{S}$ values (15.7 to 17.4‰). One
individual (FCS-88) has an intermediate value (12.6‰), and the others (FCS-84, FCS-70, FCS-
81) have low $\delta^{34}\text{S}$ values (-3.3 to 6.9‰).

**3.2. Dietary isotope results**

The isotope results are presented in tables 1 and 2. The $\delta^{13}\text{C}$ values range between -18.3‰ and
59 -20.1‰, with a mean of -19.3‰ ($\pm 0.6‰$, 1σ) and $\delta^{15}\text{N}$ values range between 9.4‰ and 12.8‰

(figure 4), with a mean of 11.2‰ ($\pm 0.9\%$, 1σ). Two outliers are observed in terms of $\delta^{15}\text{N}$. FCS-88 has the highest $\delta^{15}\text{N}$ value (12.8‰) and FCS-75 has a markedly lower value than the rest of the dataset (9.4‰). FCS-12 has the highest $\delta^{13}\text{C}$ (-18.3‰) and $\delta^{34}\text{S}$ value (17.4‰) as well as a high $\delta^{15}\text{N}$ value (11.3‰). When comparing the $\delta^{34}\text{S}$ and $\delta^{13}\text{C}$ data, the three individuals with the lowest $\delta^{34}\text{S}$ values also have the lowest $\delta^{13}\text{C}$ values (FCS-70, FCS-81 and FCS-84) and the three individuals with the highest $\delta^{34}\text{S}$ values also have the highest $\delta^{13}\text{C}$ values (FCS-09, FCS-12 and FCS-85). Although sample sizes are too small for statistical analysis, $\delta^{34}\text{S}$ and $\delta^{15}\text{N}$ isotope data show no clear relationship.

FCS-09 was the only individual for which different elements were analysed to investigate temporal changes in diet. Only a minor elevation in $\delta^{15}\text{N}$ was observed between the early developing dentine and later developing rib samples. The dietary signals were well within the range of the other crew members and the buccal caries are likely to result from a cultural practice that had a negligible dietary impact.

4. Discussion

The general trend of relatively high $\delta^{15}\text{N}$ and low $\delta^{13}\text{C}$ for all eight of the crew are characteristic of a largely terrestrial protein-rich childhood diet in a C_3 ecosystem, and consistent with isotope data from other late medieval British sites (figure 4; see [101-103] for European sites). Individuals with high $\delta^{15}\text{N}$ with relatively low $\delta^{13}\text{C}$ have been interpreted as representing protein-rich diets involving substantial consumption of freshwater fish, omnivorous animals, or crops and animals raised on manured land [28, 41] (but see [30, 31]).

The eight *Mary Rose* individuals have $^{87}\text{Sr}/^{86}\text{Sr}$ values (0.70872 to 0.71070) compatible with British origins, which Evans *et al.* [61] estimates as ranging between 0.7071 and 0.7183. The $^{87}\text{Sr}/^{86}\text{Sr}$ values are neither wide-ranging, nor particularly diagnostic, with values in this range potentially deriving from various areas across Britain [61]. Furthermore, $^{87}\text{Sr}/^{86}\text{Sr}$ values such as these have been recorded at various locations across Europe, the Mediterranean, Africa and the Middle East and are by no means unique to Britain [46, 104-112]. Interpretation is difficult as there is no local baseline to compare to for distinguishing locals from non-locals and therefore the integration of multiple proxies is key to interpretation.

The approach to $\delta^{18}\text{O}$ analysis was designed to address challenges and ambiguities in interpretation of provenance, but it is not possible to discount all issues of equifinality. Recommended methods were employed [81], sampling locations were standardised to ensure comparability and to minimise seasonal signals [49, 75], drinking water corrections were not used [113] and interpretations were based on comparisons with large British datasets [42, 43]. The *Mary Rose* provides a particular challenge, as it cannot be treated as a settled community from which outliers can be identified statistically, as advocated by Lightfoot and O'Connell [48]. The same authors state that pinpointing specific homelands should not usually be attempted. We therefore take an exploratory approach, suggesting potential areas of origin based on integration with other isotope proxies and artefactual evidence.

The $\delta^{18}\text{O}$ values suggest that as many as three individuals may be from warmer climates than Britain. FCS-70, FCS-81 and FCS-85 have high $\delta^{18}\text{O}_\text{p}$ values (21.2‰, 19.8‰ and 19.3‰, respectively) that are more than two standard deviations from the British mean of 17.7‰ $\pm 1.4\%$ (2σ , $n=615$; see [42, 43]), the dataset compiled for this estimation included data from medieval sites and therefore accounts for possible elevations in $\delta^{18}\text{O}_\text{p}$ of human enamel samples relating to medieval culinary practices, such as brewing, stewing and boiling [24].

Nevertheless, due to the various factors affecting $\delta^{18}\text{O}_p$ values in archaeological populations
$\delta^{18}\text{O}$ analysis remains a complex tool for provenancing humans and must be interpreted with
caution.

Three of the five 'British' individuals (FCS-09, FCS-12 and FCS-75) have high $\delta^{34}\text{S}$ values
($>14\text{‰}$) that suggest coastal origins [60]; FCS-88 has an intermediate value and FCS-84 has a
low 'inland' value. Within the group of likely migrants, FCS-85 has a high 'coastal' $\delta^{34}\text{S}$ value
whereas FCS-70 and FCS-81 have very low $\delta^{34}\text{S}$ values suggesting 'inland' residence,
potentially in areas of wetland and/or impervious lithology [58]. The three individuals with the
highest $\delta^{34}\text{S}$ values (FCS-09, FCS-12 and FCS-85) also had the highest $\delta^{13}\text{C}$ values, supporting
a coastal residence [59].

The two potential sub-sets of the crew are discussed below, with more detailed biographical
reconstruction for individuals with particularly interesting isotope values.

21 **4.1. Individuals with isotope values consistent with British origins**

Five of the individuals (FCS-09, FCS-12, FCS-75, FCS-84 and FCS-88) have isotope values
consistent with a childhood spent in Britain. With an $\delta^{18}\text{O}_p$ value of 18.4‰ and a $^{87}\text{Sr}/^{86}\text{Sr}$
value of 0.71070, FCS-09 sits well within British isotope ranges and is discussed in more detail
in a later section, as ancestry analyses were also undertaken.

All five have $\delta^{18}\text{O}_p$ values (group mean = 18.6‰) closer to the mean value for western Britain
($18.2 \pm 1.0\text{‰}$, 2σ , $n=40$) than for eastern Britain ($17.2 \pm 1.3\text{‰}$, 2σ , $n=83$) [42]. These crew
members were therefore most likely raised in the west of Britain or perhaps the southern coastal
strip, which is also characterised by higher $\delta^{18}\text{O}_p$ values [42].

FCS-09 and FCS-75 have very similar isotope values (FCS-09: $\delta^{18}\text{O}_p = 18.4\text{‰}$, $^{87}\text{Sr}/^{86}\text{Sr} =$
0.71070 , $\delta^{34}\text{S} = 16.7\text{‰}$; FCS-75: $\delta^{18}\text{O}_p = 18.6\text{‰}$, $^{87}\text{Sr}/^{86}\text{Sr} = 0.71055$, $\delta^{34}\text{S} = 15.7\text{‰}$) and may
have originated from the same region. Their $\delta^{18}\text{O}_p$ values suggest a western or southern British
origin, and their $\delta^{34}\text{S}$ values indicate a childhood spent in close proximity ($<30\text{km}$) to the sea
[60, 61]. Values of $^{87}\text{Sr}/^{86}\text{Sr}$ close to 0.7110 are commonly found on Palaeozoic rocks (e.g.
Devonian, Silurian and Ordovician) that are common on the western side of southern Britain,
such as in South Wales, Devon and Cornwall, as well as Carboniferous grits in North Devon
[46, 61, 105]. A coastal settlement in the southwest of Britain is a plausible childhood residence
for FCS-09 and FCS-75. Privateering traditions, commercial trade routes and the long medieval
wars with France encouraged lively maritime activity in the southwest of Britain in the 16th
century [114]. Well-known maritime hubs of the period that are consistent with these values
include Poole, Weymouth (Dorset), Plymouth, Dartmouth (Devon) and Fowey (Cornwall) [61,
115].

Suggesting origins for the remaining three British individuals is more difficult. The $\delta^{18}\text{O}$ values
indicate likely origins in the west or south. The $^{87}\text{Sr}/^{86}\text{Sr}$ values of FCS-12 (0.70965), FCS-84
(0.70915) and FCS-88 (0.70979) relate to rocks of various ages across vast swathes of the
British Isles [46]. All are higher than would be expected for chalk bedrocks and therefore large
areas of Wessex and southeast England can be discounted. The values are also too low to relate
to older, more radiogenic geologies such as Proterozoic and Archaean bedrocks, which cover
large parts of Scotland and pockets of northern England [46, 105]. Most of Wales can be
excluded, as this area is dominated by more radiogenic lithologies [61]. The $^{87}\text{Sr}/^{86}\text{Sr}$ values
may relate to limestone geologies although there are various other possibilities within Britain

(and beyond). Values between 0.709 and 0.710 are very common in the British biosphere, leading Montgomery [116] to describe this range as ‘the strontium of doom’ due to the limited interpretative resolution that can be achieved. Considering the $\delta^{34}\text{S}$ values of these three individuals, FCS-12 has a very high $\delta^{34}\text{S}$ value (17.4‰) clearly indicative of coastal origins, most likely from more exposed western areas. FCS-88 (12.6‰) has a more intermediate value, whereas FCS-84 (6.9‰) had a more inland childhood abode [61]. Despite similar $^{87}\text{Sr}/^{86}\text{Sr}$ values, it is clear that all three came from different areas. A range of western coastal zones provide potential areas of origin for FCS-12. FCS-88 could derive from the Thames estuary or other southern/western areas, though probably not from the most exposed coastal zones of the west. FCS-84 could come from the southern end of the English midlands or inland areas of Wessex, but are not on chalk lithology (e.g. north Wiltshire), but pinpointing origins is not possible.

Due to the size of the *Mary Rose* and the threat of French invasion, she never ventured beyond British coastal waters. Apart from refitting in the River Thames prior to the Third French War, she is believed to have kept to the Channel [5]. Recruits may well have been picked up along the southern coast of Britain, although contingents of men were sent around the country to man ships during times of war. In 1514, for example, the crew of the royal ship *Lizard* comprised contingents from the coastal towns of Beaumaris, Plymouth, Hull and Tynemouth [34]. The historical evidence does not highlight particular regions for Tudor naval recruitment, however the isotope results from this study suggests five western/southern British individuals, four of which grew up near the sea. It should be noted that although the individuals discussed above have isotope values compatible with Britain, other regions of Europe cannot be ruled out. Origins in Britain provide the most parsimonious interpretation, but isotope analysis is an exclusive approach and vast swathes of northern Europe cannot be excluded.

4.2. Individuals with $\delta^{18}\text{O}_p$ values outside the British range

Three individuals (FCS-70, FCS-81 and FCS-85) have $\delta^{18}\text{O}$ values beyond the expected British range [42, 43]. High $\delta^{18}\text{O}_p$ values (>19‰) in humans are common in warmer climates than Britain, such as southern European coasts, southwest Iberia and North Africa [48, 117]. It must be stressed that there are multiple factors, other than migration from a warmer climate, which could cause higher $\delta^{18}\text{O}_p$ values relative to the other individuals (e.g. altitude, latitude, diet, temperature, water source; see [49]). However, considering late medieval and early modern isotope data from the British Isles, the $\delta^{18}\text{O}_p$ values for these three individuals suggest origins in a warmer region than Britain (see [9, 26, 28, 42, 118-120] for late medieval mobility studies conducted in Britain; also see [61]). This is supported by artefactual evidence from the *Mary Rose*.

FCS-85 may well have originated from a southern European coastal site as he has a high $\delta^{18}\text{O}_p$ value of 19.3‰, a ‘coastal’ $\delta^{34}\text{S}$ value of 17.4‰ and a $^{87}\text{Sr}/^{86}\text{
[revised manuscript text omitted]

59 crew of the *Mary Rose*. For five individuals, isotope results suggest western/southern British
60

origins where maritime activity was well established at this time, with most individuals likely raised in coastal communities. Although diet was typical for a British medieval population, provenancing isotope results and artefactual material indicate that as many as three of the eight individuals originated from warmer regions such as southern Europe or possibly North Africa. Results estimate that FCS-70 and FCS-81 may have African ancestry, although this does not preclude a southern European origin. Ancestry estimation results for FCS-09 provided a clearer estimate of African ancestry, but isotope analyses indicated origins in coastal western England. This is the first direct evidence for a Black European mariner in Henry VIII's navy. This contributes to the ever-growing body of evidence for diversity in geographic origins, ancestry and lived experiences in Tudor England. The research highlights the important contributions that individuals of diverse backgrounds and origins made to the English navy. Although isotope analysis is a useful tool for investigating provenance, it should be used with caution and suggested interpretations in this research are supported by artefactual evidence. As drivers of isotope variation become better understood and biosphere maps become higher resolution, the confidence with which origins can be reconstructed will improve. As such, these interpretations may require future refinement. Future isotopic (e.g. [133]), ancestry and aDNA analyses on more of the exceptionally well-preserved human remains of the *Mary Rose* will result in a more complete narrative of both Henry VIII's navy and the wider Tudor world.

Acknowledgements

We are grateful to the Mary Rose Trust for their support for this project and granting permission to undertake sampling of the human remains in their care. We also owe thanks to Nick Owen (Swansea University), Sam Robson and Garry Scarlett (University of Portsmouth) for discussions relating to the *Mary Rose* 'characters', and thanks to Rebecca Redfern (Museum of London) for comments on an earlier draft of this paper. We are also grateful to the Oxford Archaeology South Heritage Burials Team and Ceri Boston for providing access to unpublished osteological data. AL would like to acknowledge Christopher Brodie, Thermo Fisher, for instrumentation support.

Funding Statement

The isotope analysis was funded by the Cardiff University Early Career research fund to RM and FORDISC software was purchased by Avanti Media.

Data Accessibility

The datasets supporting this article have been included within the manuscript and uploaded as part of the Supplementary Material.

Competing Interests

There are no competing interests.

Authors' Contributions

JS and RM designed and instigated the research project. KF designed and delivered the ancestry estimation analysis. JS and RM sampled the remains and JS prepared all enamel, bone and dentine samples for analysis. JS led the production of the manuscript, assisted by RM and KF. All authors read, contributed and approved the manuscript. AN ($\delta^{18}\text{O}$, $\delta^{13}\text{C}$, $\delta^{15}\text{N}$), AL ($\delta^{34}\text{S}$), MAM and MA ($^{87}\text{Sr}/^{86}\text{Sr}$) undertook mass spectrometry. AH provided curatorial support, access and wider information on the *Mary Rose* throughout the project.

References

- 1 Loades, D., Knighton, C. S. 2002 *Letters from the Mary Rose*. Stroud: Sutton.
- 2 Marsden, P. 2003 *Sealed by Time: The Loss and Recovery of the Mary Rose*. Archaeology of the Mary Rose, vol. 1. Portsmouth: Mary Rose Trust.
- 3 Gardiner, J., Allen, M., J. 2005 *Before the mast: Life and death aboard the Mary Rose*. Archaeology of the Mary Rose, vol. 4. Portsmouth: Mary Rose Trust.
- 4 Hildred, A. 2011 *Weapons of Warre: The armaments of the Mary Rose*. Archaeology of the Mary Rose, vol. 3. Portsmouth: Mary Rose Trust.
- 5 Marsden, P., McElvogue, D. 2009 *Mary Rose: Your noblest shippe: Anatomy of a Tudor warship*. Archaeology of the Mary Rose, vol. 2. Portsmouth: Mary Rose Trust.
- 6 Stirland, A. 2000 *Raising the dead: The skeleton crew of Henry VIII's great ship, the Mary Rose*. Chichester: Wiley.
- 7 Stirland, A. J., Waldron, T. 1997 Evidence for Activity Related Markers in the Vertebrae of the Crew of the Mary Rose. *Journal of Archaeological Science*. **24**, 329-335. (10.1006/jasc.1996.0117)
- 8 Kerns, J. G., Buckley, K., Parker, A. W., Birch, H. L., Matousek, P., Hildred, A., Goodship, A. E. 2015 The use of laser spectroscopy to investigate bone disease in King Henry VIII's sailors. *Journal of Archaeological Science*. **53**, 516-520. (10.1016/j.jas.2014.11.013)
- 9 Bell, L. S., Lee Thorp, J. A., Elkerton, A. 2009 The sinking of the Mary Rose warship: a medieval mystery solved? *Journal of Archaeological Science*. **36**, 166-173. (10.1016/j.jas.2008.08.006)
- 10 Watkinson, D., Emmerson, N., Seifert, J. Matching display relative humidity to corrosion rate: Quantitative evidence for marine cast iron cannon balls. *Metal 2016: Proceedings of the Interim Meeting of the ICOM-CC Metals Working Group*. New Delhi, India 2016.
- 11 Woolgar, C. M., Serjeantson, D., Waldron, T. 2006 *Food in medieval England : diet and nutrition*. Oxford: Oxford : Oxford University Press.
- Potter, D. 2004 Britain and the Wider World. In *A Companion to Tudor Britain*. (ed.^eds. pp. 182-202. Malden, MA: Blackwell.
- Sacks, D. H. 1991 *The Widening Gate: Bristol and the Atlantic Economy, 1450-1700*. Berkeley, CA: University of California Press.
- Ormrod, M., McDonald, N. F., David Taylor, C. D. England's Immigrants 1330 – 1550 (www.englishimmigrants.com). 2015.
- Tittler, R. 2004 Society and Social Relations in British Provincial Towns. In *A Companion to Tudor Britain*. (ed.^eds. R. T. a. N. Jones), pp. 363-380. Malden, MA: Blackwell.
- Habib, I. H. 2008 *Black lives in the English archives, 1500-1677 : imprints of the invisible*. Aldershot: Burlington, VT
- Kaufmann, M. 2011 Africans in Britain, 1500-1640 [D.Phil.]: University of Oxford.
- Kaufmann, M. 2015 "Making the beast with two backs" – interracial relationships in early modern England. *Literature Compass* **12**, 22-37.
- Kaufmann, M. 2017 *Black Tudors: The Untold Story*. London: Oneworld.
- Onyeka. 2014 *Blackamoors : Africans in Tudor England, their presence, status and origins*. Revised ed. London: Narrative Eye.
- Onyeka. 2019 *England's Other Countrymen: Black Tudor Society*. London: Zed Books.
- Heng, G. 2018 *The Invention of Race in the European Middle Ages*. Cambridge : Cambridge University Press.
- Ramey, L. T. 2014 *Black Legacies: Race and the European Middle Ages*. Florida: University Press of Florida.
- Brettell, R., Montgomery, J., Evans, J. 2012 Brewing and stewing: The effect of culturally mediated behaviour on the oxygen isotope composition of ingested fluids and the implications for human provenance studies. *Journal of Analytical Atomic Spectrometry*. **27**, 778-785. (10.1039/c2ja10335d)
- Serjeantson, D., Woolgar, C. M. 2006 Fish

Consumption in Medieval
England. In *Food in*
*medieval England : diet and*
*nutrition.* (ed.^eds. C. M.
Woolgar, D. Serjeantson, T.
Waldron), pp. 102-130.
Oxford: Oxford University
Press.
- Müldner, G., Richards,
12 M. P. 2006 Diet in Medieval
England: The Evidence
from Stable Isotopes. In
*Food in medieval England :*
*diet and nutrition.* (ed.^eds.
C. M. Woolgar, D.
Serjeantson, T. Waldron),
pp. 228-238. Oxford:
Oxford University.
- Bennett, H. S. 1937 *Life*
*on the English Manor. A*
*Study of Peasant*
*Conditions, 1150–1400.*
Cambridge: Cambridge
University Press.
- Müldner, G., Richards,
28 M. P. 2005 Fast or feast:
reconstructing diet in later
medieval England by stable
isotope analysis. *Journal of*
*Archaeological Science.* **32**,
39-48.
(10.1016/j.jas.2004.05.007)
- Müldner, G., Richards,
36 M. P. 2007 Diet and
37 diversity at later medieval
fishergate: The isotopic
evidence. *American Journal*
*of Physical Anthropology.*
**134**, 162-174.
(10.1002/ajpa.20647)
- Craig-Atkins, E., Jervis,
B., Cramp, L., Hammann,
S., Nederbragt, A. J.,
Nicholson, E., Taylor, A.
R., Whelton, H., Madgwick,
R. 2020 The dietary impact
of the Norman Conquest: A
multiproxy archaeological
investigation of Oxford,
UK. *PLoS one.* **15**,
e0235005-e0235005.
(10.1371/journal.pone.0235
005)
- Craig-Atkins, E.,
Towers, J., Beaumont, J.
2018 The role of infant life
histories in the construction
of identities in death: An
incremental isotope study of
dietary and physiological
status among children
afforded differential burial.
American Journal of
Physical Anthropology. **167**,
644-655.
(10.1002/ajpa.23691)
- Childs, D. 2007 *The*
Warship Mary Rose: The
Life and Times of King
Henry VIII's Flagship.
London: Greenhill Books in
association with the Mary
Rose Trust.
- Scammell, G. V. 1960
War at sea under the early
Tudors: some Newcastle
upon Tyne evidence.
Archaeologia Aeliana 4th
Series. **38**, 73-97.
- Scammell, G. V. 1961
War at sea under the early
Tudors - Part II.
Archaeologia Aeliana 4th
Series. **39**, 179-205.
- Loades, D. 1992 *The*
Tudor Navy: An
administrative, political and
military history. Volume 1,
660-1649. Aldershot: Scolar
Press.
- Knighton, C. S., Loades,
D. 2000 *The Anthony Roll*
of Henry VIII's Navy: Pepys
Library 2991 and British
Library Additional MS
22047 with Related
Documents. Burlington, Vt.:
Ashgate for the Navy
Records Society in
association with the British
Library and Magdalene
College, Cambridge.
- Millar, G. J. 1980 *Tudor*
mercenaries and auxiliaries
1485-1547. Charlottesville,
VA: University Press of
Virginia.
- Sicking, L. 2010 Naval
warfare in Europe, c. 1330–
c. 1680. In *European*
Warfare, 1350–1750.
(ed.^eds. D. J. B. Trim, F.
Tallett), pp. 236-263.
Cambridge: Cambridge
University Press.
- Millard, A. R.,
Schroeder, H. 2010 'True
British sailors': a comment
on the origin of the men of
the Mary Rose. *Journal of*
Archaeological Science. **37**,
680-682.
(10.1016/j.jas.2009.06.031)
- Bell, L. S., Lee-Thorp, J.
A., Elkerton, A. 2010
Sailing against the wind:
Reply to Millard and
Schroeder: 'True British
sailors': A comment on the
origin of the men of the
Mary Rose. *Journal of*
Archaeological Science. **37**,
683-686.
(10.1016/j.jas.2009.12.021)
- Müldner, G., Richards,
M. P. 2007 Stable isotope
evidence for 1500 years of
human diet at the city of
York, UK. *American*
Journal of Physical
Anthropology. **133**, 682-
697. (10.1002/ajpa.20561)
- Evans, J. A., Chenery, C.
A., Montgomery, J. 2012 A
summary of strontium and
oxygen isotope variation in
archaeological human tooth
enamel excavated from
Britain. *Journal of*
Analytical Atomic
Spectrometry. **27**, 754-764.
(10.1039/c2ja10362a)
- Pellegrini, M., Pouncett,
J., Jay, M., Pearson, M. P.,
Richards, M. P. 2016 Tooth
enamel oxygen "isoscapes"
show a high degree of
human mobility in
prehistoric Britain.
Scientific Reports. **6**,
(10.1038/srep34986)

Veis, A. 1989
Biochemical Studies of
Vertebrate Tooth
Mineralization. In
*Bio mineralization:*
*Chemical and Biochemical*
*Perspectives*. (ed.^eds. S.
Mann, J. Webb, R. J. P.
Williams), pp. 189-222.
New York: VCH.
Montgomery, J. 2010
Passports from the past:
Investigating human
dispersals using strontium
isotope analysis of tooth
enamel. *Annals of Human*
*Biology*. **37**, 325-346.
(10.3109/030144610036492
97)
Evans, J., Montgomery,
24 J., Wildman, G., Boulton,
25 N. 2010 Spatial variations in
biosphere Sr-87/Sr-86 in
Britain. *Journal Of The*
*Geological Society*. **167**, 1-
4. (10.1144/0016-
76492009-090)
Sealy, J. 2001 Body
Tissue Chemistry and
Palaeodiet. In *Handbook of*
*Archaeological Sciences*.
(ed.^eds. D. R. Brothwell,
36 A. M. Pollard), pp. 269-
37 279. Chichester: John Wiley
and Sons.
Lightfoot, E., O'Connell,
40 T. C. 2016 On the Use of
41 Biomineral Oxygen Isotope
Data to Identify Human
Migrants in the
Archaeological Record:
Intra-Sample Variation,
Statistical Methods and
Geographical
Considerations. *PloS one*.
**11**, e0153850.
(10.1371/journal.pone.0153
850)
Pederzani, S., Britton, K.
2019 Oxygen isotopes in
bioarchaeology: Principles
and applications, challenges
and opportunities. *Earth-*
*Science Reviews*. **188**, 77-
107.
(10.1016/j.earscirev.2018.1
1.005)
Daux, V., Lécuyer, C.,
Héran, M.-A., Amiot, R.,
Simon, L., Fourel, F.,
Martineau, F., Lynnerup,
N., Reychler, H., Escarguel,
G. 2008 Oxygen isotope
fractionation between
human phosphate and water
revisited. *Journal of Human*
Evolution. **55**, 1138-1147.
(10.1016/j.jhevol.2008.06.0
06)
Royer, A., Daux, V.,
Fourel, F., Lécuyer, C. 2017
Carbon, nitrogen and
oxygen isotope fractionation
during food cooking:
Implications for the
interpretation of the fossil
human record. *American*
Journal of Physical
Anthropology. **163**, 759-
771. (10.1002/ajpa.23246)
Budd, P., Millard, A.,
Chenery, C., Lucy, S.,
Roberts, C. 2004
Investigating population
movement by stable isotope
analysis: A report from
Britain. *Antiquity*. **78**, 127-
141.
(10.1017/S0003598X00092
98X)
Lamb, A. L., Melikian,
M., Ives, R., Evans, J. 2012
Multi-isotope analysis of the
population of the lost
medieval village of
Auldhame, East Lothian,
Scotland. *Journal of*
Analytical Atomic
Spectrometry. **27**, 765-777.
(10.1039/c2ja10363j)
Madgwick, R., Sykes,
N., Miller, H., Symmons,
R., Morris, J., Lamb, A.
2013 Fallow deer (*Dama*
dama) management
in Roman South-East
Britain. *Archaeological and*
Anthropological Sciences.
5, 111-122.
(10.1007/s12520-013-0120-
0)
Vika, E. 2009 Strangers
in the grave? Investigating
local provenance in a Greek
Bronze Age mass burial
using δ 34S analysis.
Journal of Archaeological
Science. **36**, 2024-2028.
(10.1016/j.jas.2009.05.022)
Worley, F., Madgwick,
R., Pelling, R., Marshall, P.,
Evans, J. A., Lamb, A. L.,
López-Dóriga, I. L., Bronk
Ramsey, C., Dunbar, E.,
Reimer, P., *et al.* 2019
Understanding Middle
Neolithic food and farming
in and around the
Stonehenge World Heritage
Site: An integrated
approach. *Journal of*
Archaeological Science:
Reports. **26**, 101838.
(10.1016/j.jasrep.2019.05.0
03)
Madgwick, R., Grimes,
V., Lamb, A. L.,
Nederbragt, A. J., Evans, J.
A., McCormick, F. 2019
Feasting and Mobility in
Iron Age Ireland: Multi-
isotope analysis reveals the
vast catchment of Navan
Fort, Ulster. *Scientific*
Reports. **9**, 19792-19714.
(10.1038/s41598-019-
55671-0)
Krouse, H. R. 1989
Sulfur Isotope studies of the
Pedosphere and Biosphere.
In *Stable isotopes in*
ecological research.
(ed.^eds. P. W. Rundel, J.
R. Ehleringer, K. A. Nagy),
pp. 424-444. New York:
New York : Springer.
Guiry, E. J., Szpak, P.
2020 Seaweed-eating sheep
show that δ 34S evidence for
marine diets can be fully
masked by sea spray effects.
Rapid Communications in

*Mass Spectrometry*. **34**,
e8868. (10.1002/rcm.8868)
Nehlich, O. 2015 The
application of sulphur
isotope analyses in
archaeological research: A
review. *Earth-Science*
*Reviews*. **142**, 1-17.
(10.1016/j.earscirev.2014.1
2.002)
Evans, J. A., Chenery, C.
14 A., Mee, K., Cartwright, C.
15 E., Lee, K. A., Marchant, A.
P., Hannaford, L. Biosphere
Isotope Domains GB (V1):
Interactive Website. British
Geological Survey.
(Interactive Resource).
2018.
Zazzo, A., Monahan, F.
23 J., Moloney, A. P., Green,
S., Schmidt, O. 2011
Sulphur isotopes in animal
hair track distance to sea.
*Rapid Communications in*
*Mass Spectrometry*. **25**,
2371-2378.
(10.1002/rcm.5131)
Richards, M. P. 2019
Isotope analysis for diet
studies. In *Archaeological*
*Science: An Introduction*.
(ed.^eds. M. P. Richards, K.
Britton), pp. 125-144.
Cambridge: Cambridge
University Press.
Fraser, R. A., Bogaard,
40 A., Heaton, T., Charles, M.,
Jones, G., Christensen, B.
42 T., Halstead, P., Merbach,
I., Poulton, P. R., Sparkes,
D., *et al.* 2011 Manuring
and stable nitrogen isotope
ratios in cereals and pulses:
towards a new
archaeobotanical approach
to the inference of land use
and dietary practices.
*Journal of Archaeological*
*Science*. **38**, 2790-2804.
(10.1016/j.jas.2011.06.024)
Stevens, R., Lightfoot,
E., Hamilton, J., Cunliffe,
B., Hedges, R. 2013 One for
the master and one for the
dame: stable isotope
investigations of Iron Age
animal husbandry in the
Danebury Environs.
*Archaeological and*
*Anthropological Sciences*.
**5**, 95-109.
(10.1007/s12520-012-0114-
3)
Lamb, A. L., Evans, J.
68 A., Buckley, R., Appleby, J.
2014 Multi-isotope analysis
demonstrates significant
lifestyle changes in King
Richard III. *Journal of*
*Archaeological Science*. **50**,
559-565.
(10.1016/j.jas.2014.06.021)
Chenery, C., Lamb, A.,
Evans, J., Sloane, H.,
Stewart, C. 2014 Isotope
Analysis of the Individuals
from the Ridgeway Hill
Mass Grave. In *'Given to*
*the ground': a Viking Age*
*mass grave on Ridgeway*
*Hill, Weymouth*. (ed.^eds. L.
Loe), pp. 23-48. Dorchester:
Dorset County Museum.
Madgwick, R., Lamb, A.
88 L., Sloane, H., Nederbragt,
89 A. J., Albarella, U., Pearson,
90 M. P., Evans, J. A. 2019
Multi-isotope analysis
reveals that feasts in the
Stonehenge environs and
across Wessex drew people
and animals from
throughout Britain. *Science*
*Advances*. **5**, eaau6078.
(10.1126/sciadv.aau6078)
Parker Pearson, M.,
Sheridan, A., Jay, M.,
Chamberlain, A., Richards,
102 M. P., Evans, J., Prehistoric,
S., Prehistoric Society of
East, A. 2019 *The Beaker*
*people : isotopes, mobility*
*and diet in prehistoric*
*Britain*. Oxford : Oxbow
Books.
Owen, N. Associate
Professor in Biomechanics,
Swansea University.
Personal Communication.
28th February 2018.
Slovak, N. M., Paytan,
115 A. 2011 Applications of Sr
Isotopes in Archaeology. In
*Handbook of Environmental*
*Isotope Geochemistry*.
(ed.^eds. M. Baskaran), pp.
743-768. Berlin: Springer-
Verlag.
Alqahtani, S. J., Hector,
123 M. P., Liversidge, H. M.
2010 Brief communication:
The London atlas of human
tooth development and
eruption. *American Journal*
*of Physical Anthropology*.
**142**, 481-490.
(10.1002/ajpa.21258)
Smith, T. M., Tafforeau,
P. 2008 New visions of
dental tissue research: Tooth
development, chemistry,
and structure. *Evolutionary*
*Anthropology*. **17**, 213-226.
(10.1002/evan.20176)
Burt, N. M. 2013 Stable
isotope ratio analysis of
breastfeeding and weaning
practices of children from
medieval Fishergate House
York, UK. *American*
*Journal of Physical*
*Anthropology*. **152**, 407-
416. (10.1002/ajpa.22370)
Plomp, E., von Holstein,
I. C. C., Kootker, L. M.,
Verdegaal-Warmerdam, S.
150 J. A., Forouzanfar, T.,
Davies, G. R. 2020
Strontium, oxygen, and
carbon isotope variation in
modern human dental
enamel. *American Journal*
*of Physical Anthropology*.
**172**, 586-604.
(10.1002/ajpa.24059)
Fahy, G. E., Deter, C.,
Pitfield, R., Miszkiewicz, J.
161 J., Mahoney, P. 2017 Bone
deep: Variation in stable
isotope ratios and
histomorphometric

measurements of bone
remodelling within adult
humans. *Journal of*
*Archaeological Science*. **87**,
10-16.
(10.1016/j.jas.2017.09.009)
Font, L., Nowell, G. M.,
Graham Pearson, D., Ottley,
C. J., Willis, S. G. 2007 Sr
isotope analysis of bird
feathers by TIMS: a tool to
trace bird migration paths
and breeding sites. *Journal*
*of Analytical Atomic*
*Spectrometry*. **22**, 513-522.
(10.1039/b616328a)
78 Nier, A. O. 1938 The
Isotopic Constitution of
Strontium, Barium,
Bismuth, Thallium and
Mercury. *Physical Review*.
**54**, 275-278.
(10.1103/PhysRev.54.275)
Avanzinelli, R., Boari,
E., Conticelli, S.,
Francalanci, L., Guarnieri,
30 L., Perini, G., Petrone, C.,
Tommasini, S., Ulivi, M.
2005 High precision Sr, Nd,
and Pb isotopic analyses
using the new generation
Thermal Ionisation Mass
Spectrometer
ThermoFinnigan Triton-
Ti®. *Periodico di*
*Mineralogia*. **74**, 147-166.
Romaniello, S. J., Field,
41 M. P., Smith, H. B.,
Gordon, G. W., Kim, M. H.,
Anbar, A. D. 2015 Fully
automated chromatographic
purification of Sr and Ca for
isotopic analysis. *Journal of*
*Analytical Atomic*
*Spectrometry*. **30**, 1906-
1912.
(10.1039/c5ja00205b)
Demény, A., Gugora, A.
D., Kesjár, D., Lécuyer, C.,
Fourel, F. 2019 Stable
isotope analyses of the
carbonate component of
bones and teeth: The need
for method standardization.
Journal of Archaeological
Science. **109**, 104979.
(10.1016/j.jas.2019.104979)
Coplen, T. B. 1988
Normalization of oxygen
and hydrogen isotope data.
Chemical Geology: Isotope
Geoscience Section. **72**,
293-297. (10.1016/0168-
9622(88)90042-5)
Chenery, C. A., Pashley,
V., Lamb, A. L., Sloane, H.
J., Evans, J. A. 2012 The
oxygen isotope relationship
between the phosphate and
structural carbonate
fractions of human
bioapatite. *Rapid*
Communications in Mass
Spectrometry. **26**, 309-319.
(10.1002/rcm.5331)
Brown, T. A., Nelson, D.
E., Vogel, J. S., Southon, J.
R. 1988 Improved Collagen
Extraction by Modified
Longin Method.
Radiocarbon. **30**, 171-177.
(10.1017/S00338222000441
18)
Nehlich, O., Richards,
M. 2009 Establishing
collagen quality criteria for
sulphur isotope analysis of
archaeological bone
collagen. *Archaeological*
and Anthropological
Sciences. **1**, 59-75.
(10.1007/s12520-009-0003-
6)
Leach, S., Eckardt, H.,
Chenery, C., Muldner, G.,
Lewis, M. 2010 A Lady of
York: migration, ethnicity
and identity in Roman
Britain. *Antiquity*. **84**, 131-
145.
(10.1017/S0003598X00099
816)
Leach, S., Lewis, M.,
Chenery, C., Müldner, G.,
Eckardt, H. 2009 Migration
and diversity in Roman
Britain: A multidisciplinary
approach to the
identification of immigrants
in Roman York, England.
American Journal of
Physical Anthropology. **140**,
546-561.
(10.1002/ajpa.21104)
Redfern, R., Hefner, J.
T. 2019 “Officially absent
but actually present”:
bioarchaeological evidence
for population diversity in
London during the Black
Death, AD 1348-50. In
Bioarchaeology of
Marginalized People.
(ed. ^eds. M. L. Mant, A. J.
Holland), pp. 69-114. San
Diego, CA: Elsevier
Science & Technology.
Fuentes, A., Ackermann,
R. R., Athreya, S., Bolnick,
D., Lasisi, T., Lee, S. H.,
McLean, S. A., Nelson, R.
2019 AAPA Statement on
Race and Racism. *American*
Journal of Physical
Anthropology. **169**, 400-
402. (10.1002/ajpa.23882)
Bethard, J. D., DiGangi,
E. A. 2020 Letter to the
Editor—Moving Beyond a
Lost Cause: Forensic
Anthropology and Ancestry
Estimates in the United
States. *Journal of Forensic*
Sciences. **65**, 1791-1792.
(10.1111/1556-4029.14513)
Buikstra, J. E., Ubelaker,
D. H. 1994 *Standards for*
Data Collection from
Human Skeletal Remains.
Fayetteville, Ark.: Arkansas
Archeological Survey
research series, no.44.
Jantz, R. L., Ousley, S.
D. 2005 *Fordisc, version*
3.1. Knoxville, TN:
University of Tennessee.
Hefner, J. T., Ousley, S.
D. 2014 Statistical
Classification Methods for
Estimating Ancestry Using
Morphoscopic Traits.
Journal of Forensic

*Sciences*. **59**, 883-890.
(10.1111/1556-4029.12421)
Atalay, S. 2006
Indigenous Archaeology as
Decolonizing Practice.
*American Indian Quarterly*.
**30**, 280-310.
(10.1353/aiq.2006.0015)
Muller, J. L., Pearlstein,
12 K. E., de la Cova, C. 2016
Dissection and Documented
Skeletal Collections:
Embodiments of Legalized
Inequality. In *The*
*Bioarchaeology of*
*Dissection and Autopsy in*
*the United States*. (ed.^eds.
20 K. C. Nystrom), pp. 185-
21 201. Cham: Springer
International Publishing.
Watkins, R., Muller, J.
2015 Repositioning the
Cobb human archive: The
merger of a skeletal
collection and its texts:
Repositioning the Cobb
Human Archive. *American*
*Journal of Human Biology*.
**27**, 41-50.
(10.1002/ajhb.22650)
Spradley, M. K., Jantz,
R. L., Robinson, A.,
Peccerelli, F. 2008
Demographic Change and
Forensic Identification:
Problems in Metric
Identification of Hispanic
Skeletons. *Journal of*
*Forensic Sciences*. **53**, 21-
28. (10.1111/j.1556-
4029.2007.00614.x)
Hughes, C. E., Dudzik,
B., Algee-Hewitt, B. F. B.,
Jones, A., Anderson, B. E.
2019 Understanding
(Mis)classification Trends
of Latin Americans in
Fordisc 3.1: Incorporating
Cranial Morphology,
Microgeographic Origin,
and Admixture Proportions
for Interpretation. *Journal*
*of Forensic Sciences*. **64**,
353-366. (10.1111/1556-
4029.13893)
Ousley, S., Jantz, R.,
Freid, D. 2009
Understanding race and
human variation: Why
forensic anthropologists are
good at identifying race.
American Journal of
Physical Anthropology. **139**,
68-76.
(10.1002/ajpa.21006)
Relethford, J. H. 2009
Race and global patterns of
phenotypic variation.
American Journal of
Physical Anthropology. **139**,
16-22.
(10.1002/ajpa.20900)
Colleter, R., Clavel, B.,
Pietrzak, A., Duchesne, S.,
Schmitt, L., Richards, M. P.,
Telmon, N., Crubézy, É.,
Jaouen, K. 2019 Social
status in late medieval and
early modern Brittany:
insights from stable isotope
analysis. *Archaeological*
and Anthropological
Sciences. **11**, 823-837.
(10.1007/s12520-017-0547-
9)
Inskip, S., Carroll, G.,
Waters-Rist, A., López-
Costas, O. 2018 Diet and
food strategies in a southern
al-Andalusian urban
environment during
Caliphal period, Écija,
Sevilla. *Archaeological and*
Anthropological Sciences.
11, 3857-3874.
(10.1007/s12520-018-0694-
7)
Gismondi, A., Baldoni,
M., Gnes, M., Scorrano, G.,
D'Agostino, A., Di Marco,
G., Calabria, G., Petrucci,
M., Müldner, G., Von
Tersch, M., *et al.* 2020 A
multidisciplinary approach
for investigating dietary and
medicinal habits of the
Medieval population of
Santa Severa (7th-15th
centuries, Rome, Italy).
PloS one. **15**, e0227433-
e0227433.
(10.1371/journal.pone.0227
433)
Voerkelius, S., Lorenz,
G. D., Rummel, S., Quézel,
C. R., Heiss, G., Baxter, M.,
Brach-Papa, C., Deters-
Itzelsberger, P., Hoelzl, S.,
Hoogewerff, J., *et al.* 2010
Strontium isotopic
signatures of natural mineral
waters, the reference to a
simple geological map and
its potential for
authentication of food. *Food*
Chemistry. **118**, 933-940.
(10.1016/j.foodchem.2009.0
4.125)
Montgomery, J., Evans,
J. A., Wildman, G. 2006
87Sr/ 86Sr isotope
composition of bottled
British mineral waters for
environmental and forensic
purposes. *Applied*
Geochemistry. **21**, 1626-
1634.
(10.1016/j.apgeochem.2006.
07.002)
Willmes, M., Bataille,
C. P., James, H. F., Moffat,
I., McMorrow, L., Kinsley,
L., Armstrong, R. A.,
Eggins, S., Grün, R. 2018
Mapping of bioavailable
strontium isotope ratios in
France for archaeological
provenance studies. *Applied*
Geochemistry. **90**, 75-86.
(10.1016/j.apgeochem.2017.
12.025)
Killgrove, K. 2010
Identifying Immigrants to
Imperial Rome using
Strontium Isotope Analysis.
In *Roman diasporas :
archaeological approaches
to mobility and diversity in
the Roman empire*. (ed.^eds.
H. Eckardt, J. L. Barta), pp.
157-174. Portsmouth, R.I.:

- Journal of Roman Archaeology. 108 Ortega, L. A., Guede, I., Zuluaga, M. C., Alonso-Olazabal, A., Murelaga, X., Niso, J., Loza, M., Quirós Castillo, J. A. 2013 Strontium isotopes of human remains from the San Martín de Dulantzi graveyard (Alegria-Dulantzi, Álava) and population mobility in the Early Middle Ages. *Quaternary International*. **303**, 54-63. (10.1016/j.quaint.2013.02.008)
- Nafplioti, A. 2011 Tracing population mobility in the Aegean using isotope geochemistry: a first map of local biologically available $^{87}\text{Sr}/^{86}\text{Sr}$ signatures. *Journal of Archaeological Science*. **38**, 1560-1570. (10.1016/j.jas.2011.02.021)
- Whelton, H. L., Lewis, J., Halstead, P., Isaakidou, V., Triantaphyllou, S., Tzevelekidi, V., Kotsakis, K., Evershed, R. P. 2018 Strontium isotope evidence for human mobility in the Neolithic of northern Greece. *Journal of Archaeological Science: Reports*. **20**, 768-774. (10.1016/j.jasrep.2018.06.020)
- Henderson, J., Evans, J., Barkoudah, Y. 2009 The roots of provenance: glass, plants and isotopes in the Islamic Middle East. *Antiquity*. **83**, 414-429. (10.1017/S0003598X00098525)
- Tafuri, M. A., Bentley, R. A., Manzi, G., Di Lernia, S. 2006 Mobility and kinship in the prehistoric Sahara: Strontium isotope analysis of Holocene human skeletons from the Acacus Mts. (southwestern Libya). *Journal of Anthropological Archaeology*. **25**, 390-402. (10.1016/j.jaa.2006.01.002)
- Pollard, A. M., Pellegrini, M., Lee-Thorp, J. A. 2011 Technical note: some observations on the conversion of dental enamel $\delta^{18}\text{O}(\text{p})$ values to $\delta^{18}\text{O}(\text{w})$ to determine human mobility. *American journal of physical anthropology*. **145**, 499. (10.1002/ajpa.21524)
- Appleby, J. 1992 Devon Privateering from Early Times to 1688. In *The New Maritime History Of Devon Volume 1 – From Early Times to the Late Eighteenth Century*. (ed. ^eds. M. Duffy, S. Fisher, B. Greenhill, D. Starkey, J. Youings), pp. 90-97. London: Conway Maritime Press.
- Kowaleski, M. 2008 Shipping and the Carrying Trade in Medieval Dartmouth. In *Von Nowgorod bis London: Studien zu Handel, Wirtschaft und Gesellschaft im mittelalterlichen Europa. Festschrift für Stuart Jenks zum 60. Geburtstag*. (ed. ^eds. M.-L. Heckmann, J. Röhrkasten), pp. 465-487. Göttingen: V&R Unipress.
- Montgomery, J., Grimes, V., Buckberry, J., Evans, J. A., Richards, M. P., Barrett, J. H. 2014 Finding Vikings with Isotope Analysis: The View from Wet and Windy Islands. *Journal of the North Atlantic*. **7**, 54-70. (10.3721/037.002.sp705)
- Lykoudis, S. P., Argiriou, A. A. 2007 Gridded data set of the stable isotopic composition of precipitation over the eastern and central Mediterranean. *Journal of Geophysical Research: Atmospheres*. **112**, n/a-n/a. (10.1029/2007JD008472)
- Roberts, C. A., Millard, A. R., Nowell, G. M., Gröcke, D. R., Macpherson, C. G., Pearson, D. G., Evans, D. H. 2013 Isotopic tracing of the impact of mobility on infectious disease: The origin of people with treponematoses buried in hull, England, in the late medieval period. *American Journal of Physical Anthropology*. **150**, 273-285. (10.1002/ajpa.22203)
- Müldner, G., Montgomery, J., Cook, G., Ellam, R., Gledhill, A., Lowe, C. 2009 Isotopes and individuals: diet and mobility among the medieval Bishops of Whithorn. *Antiquity*. **83**, 1119-1133. (10.1017/S0003598X00099403)
- Toolis, R., Barrett, J., Boulton, N., Chenery, C., Evans, J., Hall, D., Macsween, A., Melikian, M., Richards, M. 2008 Excavation of medieval graves at St Thomas' Kirk, Hall of Rendall, Orkney. *Proceedings of the Society of Antiquaries of Scotland*. **138**, 239-266.
- Persits, F. M., Ahlbrandt, T. S., Tuttle, M. L., Charpentier, R. R., Brownfield, M. E., Takahashi, K. I. 2002 Maps showing geology, oil and gas fields and geological provinces of Africa. Report. Reston, VA. Report No.: 97-470A. Version 2.0.

- Guede, I., Zuluaga, M. C., Ortega, L. A., Alonso-Olazabal, A., Murelaga, X., Garcia Camino, I., Iacumin, P. 2020 Social structuration in medieval rural society based on stable isotope analysis of dietary habits and mobility patterns: San Juan de Momotio (Biscay, north Iberian Peninsula). *Journal of Archaeological Science, Reports*. **31**, 102300. (10.1016/j.jasrep.2020.102300)
- Guede, I., Ortega, L. A., Zuluaga, M. C., Alonso-Olazabal, A., Murelaga, X., Pina, M., Gutierrez, F. J., Iacumin, P. 2017 Isotope analyses to explore diet and mobility in a medieval Muslim population at Tauste (NE Spain). *PLoS one*. **12**, e0176572-e0176572. (10.1371/journal.pone.0176572)
- MacRoberts, R. A., Barrocas Dias, C. M., Matos Fernandes, T., Santos, A. L., Umbelino, C., Gonçalves, A., Santos, J., Ribeiro, S., Schöne, B. R., Barros, F., *et al.* 2020 Diet and mobility during the Christian conquest of Iberia: The multi-isotopic investigation of a 12th–13th century military order in Évora, Portugal. *Journal of Archaeological Science, Reports*. **30**, 102210. (10.1016/j.jasrep.2020.102210)
- Francisci, G., Micarelli, I., Iacumin, P., Castorina, F., Di Vincenzo, F., Di Matteo, M., Giostra, C., Manzi, G., Tafuri, M. A. 2020 Strontium and oxygen isotopes as indicators of Longobards mobility in Italy: an investigation at Povegliano Veronese. *Scientific Reports*. **10**, 11678-11678. (10.1038/s41598-020-67480-x)
- Munde, M. 2009 An Isotopic Approach to Diet in Medieval Spain. In *Food and Drink in Archaeology 2 : University of Nottingham Postgraduate Conference*. (ed.^eds. S. Baker, A. University of Nottingham. Department of, N. University of), pp. 64-72. Totnes: Totnes : Prospect Books.
- de Luna, K. M. 2008 Agriculture. In *Encyclopedia of Society and Culture of the Medieval World*. (ed.^eds. P. J. Crabtree), pp. 15-20. New York: Facts of File.
- Ajaji, T., Weis, D., Giret, A., Bouabdellah, M. 1998 Coeval potassic and sodic calc-alkaline series in the post-collisional Hercynian Tanncherfi intrusive complex, northeastern Morocco: geochemical, isotopic and geochronological evidence. *LITHOS*. **45**, 371-393. (10.1016/S0024-4937(98)00040-1)
- Edmunds, W. M., Guendouz, A. H., Mamou, A., Moulla, A., Shand, P., Zouari, K. 2003 Groundwater evolution in the Continental Intercalaire aquifer of southern Algeria and Tunisia: trace element and isotopic indicators. *Applied Geochemistry*. **18**, 805-822. (10.1016/S0883-2927(02)00189-0)
- Hanihara, T., Ishida, H. 2001 Os in cae: variation in frequency in major human population groups. *Journal of Anatomy*. **198**, 137-152. (10.1046/j.1469-7580.2001.19820137.x)
- DiGangi, E. A., Hefner, J. T. 2012 Ancestry estimation. In *Research Methods in Human Skeletal Biology*. (ed.^eds. E. A. DiGangi, M. K. Moore), pp. 117-149. San Diego: Elsevier Science & Technology.
- Irish, J. D., Morez, A., Girdland Flink, L., Phillips, E. L. W., Scott, G. R. 2020 Do dental nonmetric traits actually work as proxies for neutral genomic data? Some answers from continental- and global-level analyses. *American Journal of Physical Anthropology*. **172**, 347-375. (10.1002/ajpa.24052)
- Evans, J., Pashley, V., Madgwick, R., Neil, S., Chenery, C. 2018 Tracking natural and anthropogenic Pb exposure to its geological source. *Scientific Reports*. **8**, 1969-1969. (10.1038/s41598-018-20397-y)
- Bownes, J., Clarke, L., Buckberry, J. 2018 The importance of animal baselines: Using isotope analysis to compare diet in a British medieval hospital and lay population. *Journal of Archaeological Science: Reports*. **17**, 103-110. (10.1016/j.jasrep.2017.10.046)
- Fuller, B. T., Richards, M. P., Mays, S. A. 2003 Stable carbon and nitrogen isotope variations in tooth dentine serial sections from Wharram Percy. *Journal of Archaeological Science*. **30**, 1673-1684. (10.1016/S0305-4403(03)00073-6)

Halldórsdóttir, H. H.,
Rogers, B., DiRenno, F.,
Müldner, G., Gröcke, D. R.,
Barnicle, E., Chidimuro, B.,
Evans, M., Morley, R.,
Neff, M., *et al.* 2019
Continuity and individuality
in Medieval Hereford,
England: A stable isotope
approach to bulk bone and
incremental dentine.

*Journal of Archaeological
Science, Reports.* **23**, 800-
809.
(10.1016/j.jasrep.2018.12.0
06)
Brothwell, D. R. 1981
*Digging up bones : the
excavation, treatment, and
study of human skeletal
remains.* Ithaca, NY:
Cornell University Press.

Trotter, M. 1970
Estimation of stature from
intact long bones. In
*Personal identification in
mass disasters.* (ed.^eds. T.
D. Stewart), pp. 71-83.
Washington DC:
Smithsonian Institution.

Tables

Skeleton no.	Character	$^{87}\text{Sr}/^{86}\text{Sr}$	$\delta^{18}\text{O}_c$	$\delta^{18}\text{O}_p$	$\delta^{34}\text{S}$	%S	C:S	N:S	$\delta^{13}\text{C}$	$\delta^{15}\text{N}$	%C	%N	C:N
FCS-09 Rib	Young mariner								-19.0	11.7	40.4	14.7	3.2
FCS-09	Young mariner	0.71070	27.2	18.4	16.7	0.30	373	115	-18.8	11.2	41.8	15.0	3.2
FCS-12	Cook	0.70965	27.7	18.9	17.4	0.30	387	119	-18.3	11.3	42.6	15.3	3.2
FCS-70	Royal Archer	0.70888	30.0	21.2	0.6	0.27	421	130	-20.0	10.9	41.8	15.1	3.2
FCS-75	Archer	0.71055	27.4	18.6	15.7	0.30	392	121	-19.5	9.4	42.6	15.3	3.2
FCS-81	Carpenter	0.70872	28.5	19.8	-3.3	0.28	403	124	-20.0	11.4	42.3	15.2	3.3
FCS-84	Officer	0.70915	27.2	18.4	6.9	0.29	383	119	-20.1	10.8	40.9	14.8	3.2
FCS-85	Gentleman	0.70983	28.1	19.3	17.4	0.29	388	119	-19.0	11.7	41.3	14.8	3.3
FCS-88	Purser	0.70979	27.6	18.8	12.6	0.26	437	134	-19.3	12.8	42.6	15.2	3.3

Table 1. Multi-isotope analysis results.

Isotope proxy	Mean	S.D.	Median	IQR	Min.	Max.
$\delta^{18}\text{O}_c$	27.97	0.93	27.67	0.86	27.18	29.95
$\delta^{18}\text{O}_p$	19.19	0.95	18.88	0.89	18.37	21.23
$^{87}\text{Sr}/^{86}\text{Sr}$	0.709658	0.000724	0.709719	0.000925	0.708718	0.710701
$\delta^{34}\text{S}$	10.50	8.16	14.15	11.55	-3.30	17.40
$\delta^{13}\text{C}$	-19.36	0.66	-19.39	1.03	-20.11	-18.29
$\delta^{15}\text{N}$	11.19	0.95	11.23	0.59	9.40	12.79

Table 2. Summary statistics for all isotope proxies (excl. FCS-09 rib sample). Oxygen values were converted from carbonate to phosphate using the conversion equation set out in Chenery *et al.* [83].

Figures

Figure 1. The *Mary Rose* depicted on the Anthony Roll (1546), an illustrated inventory of King Henry VIII's navy (by permission of the Pepys Library, Magdalene College, Cambridge, UK).

Figure 2. Cutaway diagram of the *Mary Rose* ship indicating the locations of individuals sampled for this study (adapted from Marsden [5], © Mary Rose Trust). FCS-09: the young mariner, FCS-12: the Cook, FCS-70: the Royal Archer, FCS-75: the Archer, FCS-81: the Carpenter, FCS-84: the Officer, FCS-85: the Gentleman, FCS-88: the Purser.

Figure 3. Oxygen and strontium isotope results for the eight crew members along with 2σ ranges for Britain as a whole, and east and west regions, following Evans *et al.* [42].

Figure 4. Carbon and nitrogen isotope results for the eight crew members, along with mean and 1σ ranges for medieval sites in Britain [28, 29, 134-136]. (N.B. comparative data comprises data from both bone and dentine collagen).

Supplemental Material

Skeleton no./ Character	Context	Sex	Age estimate (years)	Stature estimate (cm)	Teeth	Skeleton	Reason for character attribution
FCS-09 Young mariner	H2 (Hold)	M	14-20	170.6 ±3.27	No M3's. Unusual caries (all molars on buccal side). Marked hypoplasia (event at ~4-5 years).	Very complete skeleton. Clearly defined sutures. Basilar suture fusion suggests ~21 years. Skeletal pathologies suggest heavy manual work (spondyloysis of a lumbar vertebra, spur of bone on each tibia in line with the fibular facet, small chunk of the acetabulum is missing due to the left femur being knocked out of the socket antemortem). Ancestry trait tests of the cranium indicate possible Sub-Saharan/African ancestry.	Profession unknown.
FCS-12 Cook	H5 (Hold)	M	26-35	165.07 ±2.99	M3's present. Over-bite. Caries on premolars and molars. Some periodontal disease and considerable calculus. Antemortem loss of right mandibular M1 resulting in more wear on the left side. Hypoplasia (event ~18 months/2 years).	Very complete skeleton. Clearly defined sutures. Muscle marks on the scapula, osteo-arthritis on pubic symphysis and heavily buttressed lumbar.	One of two skeletons found in the galley with cooking utensils and a stove.

FCS-70 Royal Archer	M6 (Main)	M	22-25	177.79 ±3.27	M3's present. Caries on all upper molars and premolars. Hypoplasia (event ~4 years).	Fairly complete skeleton. Sagittal suture fading. Stressed back (Schmorl's nodes and osteophytes on vertebrae). Scoliosis of the thoracic spine and possibly Scheuermann's disease.	Found under a rear axle bronze cannon, with a longbow, sword, scabbard and pomander, and an archer's wristguard (bracer) with the arms of England (hence the attribution 'Royal').
FCS-75 Archer	M3 (Main)	M	18-24	170.18 ±3.27	Skull was associated with the skeletal remains at a later date so no dentition information in the report.	Fairly complete skeleton. Inflamed right elbow prior to fusion (event ~11-13 years). Bent sacrum (90° angle). Humerus capitulum is highly eburnated.	An ivory wristguard remained on the arm.
FCS-81 Carpenter	O9 (Orlop)	M	31-45	171.84 ±3.17	M3's present. Caries on right maxillary M3 and left maxillary premolars and M1. Abscess on left maxillary M1. Maxillary M2's lost antemortem. Quite severe calculus. Hypoplasia (events at ~5 and ~7 years).	Fairly complete skeleton. Sagittal and occipital sutures fading. Button osteoma (?) on R parietal. Lesion across right supraorbital ridge. Schmorl's nodes on some vertebrae. Indications of early osteoarthritis on the sternal end and thoracic vertebrae.	Found in a store area just below the carpenter's cabin with a group of carpentry tools including a wooden tool holder (worn around the waist).

FCS-84 Officer	M4/5 (Main)	M	28-32	162.61 ±2.99	M3's not present. Caries on left maxillary PM2 and mandibular M2. Both mandibular PM2's lost antemortem. Left maxillary M1 lost antemortem and has abscess.	Very complete skeleton. Osteophytic lipping of the calcaneum. Slight roughening at the ends of the humeri. Right radius and ulna slightly larger than the left. Pitting on the femurs and tibiae shafts (periosteal reactions). Osteophytes on some vertebrae.	Previous suggestion that this individual may have been the bosun due to the discovery next to a bosun's call. However, the wealth of this individual, suggested by the silver bosun's call, a belt with a silver buckle and a silk purse, implies a generic officer.
FCS-85 Gentleman	M8 (Main)	M	20-25	161.96 ±2.99	Skull was associated with the skeletal remains at a later date so no dentition information in the report.	Very complete skeleton. PR of femurs and slight PR of tibiae. Os acromiale of the scapula. Thick ulnae and radii. Very developed insertion of biceps below the trochlea.	The only skeletal remains found beside a chest in the small gun-bay. Objects in the chest (coins, seal, dice, lead token, pouch, clothes, comb, Italian bone carving) suggest wealth and status, hence the attribution of generic gentleman. (Although these skeletal remains were also found near a gun and a longbow and arrows, and his skeletal pathologies could be connected to a profession as either a gunner or an archer.)

FCS-88	O10	M	26-35	170.40	Skull was associated with the skeletal remains at a later date so no dentition information in the report.	Fairly complete skeleton. Bilateral Perthes disease confirmed by X-ray. Congenital pathology. Periosteal reactions and plaque on the right tibia. No articular facet for the fibula. Periosteal reactions along the fibula shafts. Marked coxa vara on both femoral heads and both acetabulae are very shallow. The femoral shafts of the are twisted in relation to the heads.
Purser	(Orlop)		±3.29
Found in an area with chests and containers holding various provisions (e.g. barrels with taps) and the only chest found on-board containing a number of coins, four of which were newly minted. The skeletal remains display bilateral Perthes disease which would have made movement around the ship challenging. The proximity to chests and the impeded mobility suggest a purser role.

Table S1. Information on the skeletons of the Mary Rose mariners under investigation, compiled from Human Skeletal Records from the Mary Rose Trust. FCS = fairly complete skeleton. Sex estimations were made using the morphological traits of the skull and pelvis [91]. The age estimate was based on molar wear [137], suture fusion, epiphyseal fusion and the appearance of the pubic symphysis [91]. Stature estimations were made using the equations in Trotter [138].

Skeleton no.	Character	Root sample (M1)	Enamel sample (M2)
FCS-09	Young mariner	R max. distal/lingual	R max. lingual
FCS-12	Cook	R max. lingual	R max. lingual
FCS-70	Royal Archer	R max. distal/lingual	R max. lingual
FCS-75	Archer	L. mand. mesial	L mand. lingual
FCS-81*	Carpenter	R mand. distal	R mand. lingual
FCS-84	Officer	L mand. mesial	L mand. lingual
FCS-85	Gentleman	L mand. distal	L mand. lingual
FCS-88	Purser	R max. distal/lingual	R max. lingual

Table S2. Tooth sample locations.

*N.B. A molar sample from FCS-81 was analysed for oxygen isotopes by Bell *et al.* [9], producing a lower value of 17.7‰. This may represent movement through early life or relate to other forms of intra-individual valuation [75].

Skeleton no.	Fordisc classification	Posterior probability	Chi-square typicality	Decision Tree estimate	OSSA estimate
FCS-09	Hispanic Male	0.611	0.823	Black	Black
FCS-70	White Male	0.640	0.352	Black	-
FCS-81	-	-	-	Black	-

Table S3. Ancestry estimation summary.

Figure S1. Craniometric ancestry estimation of FCS-09 from Fordisc 3.1. The X represents FCS-09, red triangles are Hispanic males, green squares are Black males, and blue diamonds are White males from the Forensic Databank, with the centroids represented by the 'HM', 'BM', 'WM' labels respectively. The ellipses represent the 95% confidence interval.

Figure S2. Craniometric ancestry estimation of FCS-70 from Fordisc 3.1. The X represents FCS-70, green squares are Hispanic males, blue diamonds are Black males, and red triangles are White males from the Forensic Databank, with the centroids represented by the 'HM', 'BM', 'WM' labels, respectively. The ellipses represent the 95% confidence interval.

Figure S3. Decision Tree, reprinted from Hefner and Ousley [93], with the classifications of individuals FCS-09, FCS-70, and FCS-81 annotated on the image. Note the classification accuracies of the reference sample within each terminal.

A note on the ancestry estimation methods

The authors recognise the difficulties in estimating ancestry for past populations using methods designed from modern reference populations. Here we will briefly expand the discussion of how we approached the ancestry estimation data. One of the major issues of estimating ancestry from phenotypic features is that the methods force a classification into discrete categories that a) do not exist in actuality and b) cannot account for probable and varying degrees of ‘mixed heritage.’ Each method will deliver a result with varying degrees of confidence; it is ultimately the discretion of the analyst of how to weigh the ancestry evidence within the context of their sample. We can see this represented in this study graphically, with figures S1-3. Although the ellipses of Figures S1 and S2 represent 95% confidence, this is a theoretical boundary based on the reference sample, and there is considerable overlap between groups. The typicality statistics provided in Table S3 are a measure of the probability of the individual belonging to the group Fordisc estimates they belong to, based on distance from the centroid of that group. In essence, the typicality measure represents how well the individual fits into that group. As highlighted in the main text, although Jantz and Ousley define any typicality over 0.05 as statistically valid, Leach et al. removed any typicality below 0.7 in their study of a Roman population. Due to the small sample size and the gestalt interpretation, this study did not set a typicality threshold, but analyst confidence in the Fordisc classification was lower for FCS-70 than FCS-09. The macromorphoscopic methods reported overall classification accuracies for the OSSA method (86.1%) and the Decision Tree (80.0%). Although the OSSA method has a higher classification accuracy, it is only able to classify an individual as ‘Black’ or ‘White,’ and therefore confidence in the result for an unknown individual is dependent on other contextual information. The overall classification accuracy for the Decision Tree is lower, but when examined group by group, the classification accuracy of Black individuals correctly classified as Black was 91.1%. We therefore afforded greater confidence to the classification of ‘Black’ by the Decision Tree. Confidence was

further improved by looking at the individual classification accuracies of each terminal of the Decision
Tree, reported in Figure S3. The terminal of the 'branch' or 'route' that FCS-70 and FCS-09 did not
include any White individuals, but the terminal of the 'branch' FCS-81 included Black, Hispanic, and
White individuals. We have elaborated upon this information in the supplementary material to provide
transparency in how we incorporated the results of the ancestry estimates. Ancestry estimation remains
challenging to apply to past populations, particularly with a small sample size, but meaningful results
can be reached through careful consideration of the methods used.

Appendix B

Dear Editor,

The authors would like to thank you and the reviewers for your considered and constructive comments. They have brought about a marked improvement in the manuscript. Our point-by-point response is below (with our responses in **blue bold**):

Associate Editor Comments to Author (Professor Matthew Collins):

Comments to the Author:

Thank you for your MS. All three reviewers agree that this is a fascinating paper which should be published and all suggesting only minor changes.

I hope that you would agree that the full data associated with the $^{87}\text{Sr}/^{86}\text{Sr}$ analyses should be included (e.g. in the supplementary section. These data should have ^{88}Sr (V), ^{85}Rb (V), $^{87}\text{Sr}/^{86}\text{Sr}$, $^{87}\text{Sr}/^{86}\text{Sr}$ (1 standard error). Given the methods described in the manuscript for the determination of the $^{87}\text{Sr}/^{86}\text{Sr}$ data. It is best practice to include all the QC data generated with these analyses.

Apologies that this was omitted from our original submission. We have added the extended $^{87}\text{Sr}/^{86}\text{Sr}$ data to the supplementary material (as well the extended $\delta^{18}\text{O}$ data).

Reviewer #1

Overall this is an excellent manuscript based on a very sound study. The authors applied multiple isotope analysis (strontium and oxygen of dental enamel; sulfur, carbon and nitrogen of dentin collagen/+1 rib sample) to construct fairly detailed, yet admittedly incomplete, osteobiographies of eight individuals from the famous shipwreck of the Mary Rose. In its current form the manuscript fulfills the main criteria for publication in this journal (Articles should report work that is scientifically sound, in which the methodology is rigorous and the conclusions are fully supported by the data.) and as such is certainly suitable for publication. I was also very impressed by the clear structure and writing, and particularly how the authors were able to carefully yet successfully incorporate the ancestry estimation results into the broader study. On this point, the way that this potentially problematic data type is used, and the further clarification provided in the supplementary document, provides a useful template for future studies attempting to integrate isotopic data and osteological data sets with ancestry estimations. The authors generally struck a good balance between over- and under-interpretation. The study is also especially important for highlighting the presence of non-Europeans in the British/European archaeological record of the late mediaeval/early modern period.

There are two main critiques of the ms, in its current, that if properly addressed would greatly improve this manuscript.

1) While the authors briefly mention the previous study of Bell et al. (2009) and the subsequent archaeological debate that this initiated (Millard & Schroeder 2010; Bell et al. 2010) there is no explicit effort made to engage in this debate nor even to compare the data in this study with that originally produced by Bell and colleagues. It is unclear why this is the case, and understandable that the authors clearly wish to emphasize their own new data.

Nonetheless, since the original study by Bell et al. included some of the same isotope systems (namely, oxygen, carbon and nitrogen) applied to the same exact skeleton collection, the complete lack of engagement with, or direct comparison to this earlier study is problematic. If the authors have reason to doubt the reliability of the data reported by Bell and colleagues, then they are obligated to be explicit about this. If not, then it really is necessary to make a clear comparison between the isotope data from Bell et al. (2009) and the isotope data reported herein (e.g. how do they compare? do the ranges overlap? are the results consistent with the previous study or differ significantly?, etc...). The data from both studies should also be plotted together in one or more figures to allow the readers to directly compare and interpret the data themselves.

We are grateful for these constructive comments and we agree that we should provide a clearer comparison with Bell et al.'s (2009) data. There are reasons for our not having done this in the original submission. We have some concerns about the comparability of the oxygen isotope data and therefore did not undertake detailed comparison initially. We appreciate that this is an odd omission so have added a caveated comparison and explained our concerns. We have added the following sentences to our discussion section to address this:

'Comparison with existing data from the Mary Rose must also be made with caution. Bell et al.'s [9] data are not directly comparable due to a different sampling methodology and the use of a NaClO pre-treatment, which has been demonstrated to make $\delta^{18}\text{O}$ values lower [114]. Once converted to $\delta^{18}\text{O}_p$ [83], the mean for the previous dataset ($17.6 \pm 0.87\%$, 1σ) is markedly lower than in this study ($19.2 \pm 0.95\%$, 1σ), although there is considerable overlap in the datasets. This difference may derive from chance sampling, as both studies relate to only a small number of individuals, or may relate to varied sampling and pre-treatment methods.'

For the carbon and nitrogen data, comparisons have been made via the biplot (Figure 4).

Also, a more explicit comment on earlier examples of African individuals recovered from (Roman) British archaeological contexts is perhaps merited (Leach et al. 2010).

We appreciate the reviewer's suggestion but do not feel that we need to refer to examples of individuals of African ancestry found in Roman contexts as this is a post-medieval study period; we believe that the reference to historical evidence for people of African ancestry in the Tudor period is sufficient.

2) As is the case with most isotopic provenance studies, the approach used is (by necessity) exclusionary. However, the manner of presenting these types of results (rather long descriptions listing all of the places where each individual could NOT have originated from for each isotope proxy) is less than ideal. In most geographic contexts there is no other option than to present the results this way. However, Britain is the most intensively and extensively mapped areas of the world for most isotope systems, and one of the few regions where high quality isoscapes exist for both strontium and oxygen isotopes (e.g. Evans et al .2012; Pelligrini et a. 2016; see also the British isotope domains dataset and online tool at <https://www.bgs.ac.uk/datasets/biosphere-isotope-domains-gb/>). As such, it is somewhat striking that this study makes no attempt at a more systematic, quantitative,

spatial approach to interpreting and presenting the isotopic provenance data. Such an approach would greatly improve the visualization, interpretation, and presentation of the isotope results by simply and effectively illustrating the areas of potential origin for each individual (or at least the British ones). Such an approach is not very complex and can be accomplished with a fairly simple application in ArcGIS, as demonstrated recently by a similar study combining skeletal isotope data and isoscapes (in a British context!) to trace the origins of individuals buried at Stonehenge (Snoeck et al. 2018).

We appreciate comments from all three reviewers about how best to present and interpret the provenancing data. The reviewers express very different opinions on this issue. We were torn on how best to approach the presentation of the data and how confidently to interpret. This remains a delicate balance to navigate as is demonstrated by the fact that some review comments stated that we need to be more ambitious and others that we need to be more cautious. Britain is indeed the most comprehensively mapped and we used the BGS multi-isotope querying tool to explore origins. On the basis of the three proxies this indicated that only one individual (FCS-09, with African ancestry) was consistent with British origins. This is, in practice, inconsistent with the data and demonstrates that we are not yet (generally, at least) in a position where we can query data to plot origins on a map. The manifold variables that affect these isotope proxies means we are in a situation of providing the most parsimonious explanations of origins and using ArcGIS approaches to plotting origins can over-simplify the complex process of exploring origins. We would like to show greater ambition in refining provenance and take a more solid, quantitative approach, but we are not sure the data can sustain it and it would go against comments of another reviewer. In addition, three of the authors have been involved in a study that has been criticised for overambitious refinement of origins (see Barclay and Brophy 2020, Archaeological Journal), so would rather err on the side of caution.

Based on the concerns detailed above, I recommend that the manuscript be accepted with minor revisions. I hope that the authors seriously consider the proposed suggestions for revision, and look forward to reading the revised and published version of this paper in Royal Society Open Science.

Reviewer #2

The manuscript by Scorrer and colleagues presents a study on the medieval warship Mary Rose. The manuscript attempts an isotope study in order to understand the origins of the crew to the ship. The article is a well written summary and the study scientifically sound, so that the whole manuscript gives a detailed insight into the crews diet and possible origins. However, there are some flaws in the study's design, since the team analysed carbon, nitrogen, sulphur, oxygen and strontium isotopes of eight individuals the sample numbers are very low and discrepancies in the data result in highly biased data. Therefore, there are no strong conclusions and the article is adding another layer of information without revealing origins or deeper understanding of the Tudor's warship. In addition the authors decided to include a fully unrelated craniometric ancestry estimation into the article, which fails to connect to the other results and is only another mosaic puzzle piece in the story, which does not fit with other results.

General remarks:

The data table 1 should be merged with table S1 since they both contain valuable information and only together these information are intuitive assessable. Additionally the table description needs to be more substantial.

We believe that combining table 1 and S1 will create too large a table so would like to keep these separate but agree with the reviewer that the data from both these tables should be better linked. We have signposted this in the table 1 caption (and have added more detail to the caption):

Table 1. Multi-isotope analysis results from dental samples of eight individuals from the Mary Rose (see electronic supplementary material, table S1 for contextual information on these individuals). Oxygen values were converted from carbonate ($\delta^{18}O_c$) to phosphate ($\delta^{18}O_p$) using the conversion equation set out in Chenery et al. [90].

The Figures need a higher resolution, Figures 1,3, and 4 need a higher resolution and better quality, my printouts were bad and even on screen were no distinctions between symbols.

For Figure 1 of the Mary Rose, unfortunately we only have permission from the Pepys Library to have the resolution of the image as 70 dpi. The quality of figures 3 and 4 have been improved for final submission.

The introduction to the Tudor kingdom and the mobility of the navy is excellent. It is well written, substantial yet not lengthy. The isotope background is alright, but lacks some introduction to isotope data from the studied period. There is a multitude of data published and an overview of available data would have been a good start for understanding the setting.

We have added the following to the end of section 1.4 to address this:

There has been little isotope work on post-medieval human remains in Britain (see [9, 53, 70, 71]). There is, however, a wealth of data for late medieval Britain, especially in terms of $\delta^{13}C$ and $\delta^{15}N$ [28, 29, 72-74], but also for $^{87}Sr/^{86}Sr$ and $\delta^{18}O$ [66, 75].

We have also clarified we are referring to the Tudor rather than medieval period when we say that direct evidence of human remains from this period is limited (at the end of section 1.3), but that we are happy to add any studies if we have missed some that relate to this period.

The study design is alright, given the importance and value of the samples, however, seven individuals are very few and therefore the robustness of such data will be limited. This is the major concern of the study, because the results will be only trustworthy, if there are no or limited numbers of outliers, but due to the nature of the historic background we would expect quite a number of outliers. This means there will be no background data for proper interpretation. Therefore, the general literature review of isotopic mobility studies is most

important.

The sample treatment and analytical details are highly detailed and should be cut to a minimum, additionally many of these methods have been published before and these should be acknowledged.

We appreciate that we have a long methods section as multiple methods were used. Other reviewers and the editor have asked for more methodological detail, so we are obliged to keep them in, though we agree that they disrupt the flow of the text.

The ancestry estimation by craniometric and morphoscopic methods should be excluded from the manuscript. Though valid to a certain point the sample number and variability in the skeletal remains are quite big and the interpretation results more in speculations than scientific conclusions. In my opinion the interpretation could be added in a side note, but not a full chapter.

In the absence of aDNA data (though ongoing research will add this in time), we deemed it important to explore ancestry to some degree though we accept that the interpretations are far from certain. However, the results for FCS-09 are convincing and this does add another biographical element to the crew. Both other reviewers praised the ancestry estimation and the value it added. As a result we would like to retain this element. Furthermore, the results of this analysis were included in the museum interpretation and a Channel 4 documentary on the Mary Rose, and so we feel that the methods used should be peer-reviewed and published so others can evaluate interpretations across the data.

The conclusion are just a summary and are relativizing the own data. In my opinion the results are only limited and this needs to be addressed. Additionally the authors have started a data set for the Mary Rose material which needs to be expanded in put into the larger context.

We are grateful for this observation and have made major changes to the conclusion. It is now less of a summary and better emphasises that our sample is small and that more work on the Mary Rose collection needs to be done. We hope that this addresses the reviewer's comment.

Minor comments:

The collagen was not ultrafiltered, this is usually sufficient for carbon and nitrogen isotope results, however, for sulphur isotopic results this could be problematic. Additionally the salt water could have compromised the materials and therefore an additional ultrafiltration step seems wise. My concern is related to the correlation of sulphur isotope values with sulphur content in collagen. The highest sulphur isotope values also revealed the highest sulphur contents. This could be indicative for seawater sulphate intrusion, therefore these data should be questioned and double checked.

This is an interesting point. We are unaware of evidence that ultrafiltration is necessary for sulphur isotope analysis of teeth deposited in marine environments. All quality control criteria were met, as demonstrated in the paper. We are very confident that our results are valid. The range of sulphur values would be difficult to explain if diagenesis had

occurred and therefore we are confident that they are biogenic. In addition, diagenetic alteration would skew C:S and N:S ratios.

The strontium results are nice, but the interpretation and presentation lacks ambition. I would recommend to use literature data for comparison and additional arguments. In my opinion these data have been neglected in the interpretation. Similarly the oxygen isotope data, which are more problematic, but in itself have some value, which needs to be addressed.

We appreciate comments from all three reviewers about how best to present and interpret the provenancing data. The reviewers express very different opinions on this issue. We were torn on how best to approach the presentation of the data and how confidently to interpret. This remains a delicate balance to navigate as is demonstrated by the fact that some review comments stated that we need to be more ambitious and others that we need to be more cautious. Britain is indeed the most comprehensively mapped and we used the BGS multi-isotope querying tool to explore origins. On the basis of the three proxies this indicated that only one individual (FCS-09, with African ancestry) was consistent with British origins. This is, in practice, inconsistent with the data and demonstrates that we are not yet (generally, at least) in a position where we can query data to plot origins on a map. The manifold variables that affect these isotope proxies means we are in a situation of providing the most parsimonious explanations of origins and using ArcGIS approaches to plotting origins can over-simplify the complex process of exploring origins. We would like to show greater ambition in refining provenance and take a more solid, quantitative approach, but we are not sure the data can sustain it and it would go against comments of another reviewer. In addition, three of the authors have been involved in a study that has been criticised for overambitious refinement of origins (see Barclay and Brophy 2020, *Archaeological Journal*), so would rather err on the side of caution.

The suggested changes are not to major in my opinion and therefore should be addressed and the manuscript altered accordingly. After doing so, in my regards to the high quality of the manuscript's style there is no issue with publication.

Reviewer #3

Scorrer et al. have prepared a well-written manuscript that succinctly presents and very adequately interprets multi-isotope and morphometric skeletal data from a relatively small (but well selected) sample of humans from the Mary Rose wreck. This paper is an important contribution to the growing literature involving these datasets, and more specifically helps to add additional scientific value to our understanding about the life-ways (i.e. origins) of individuals from the Tudor time period. I have added specific comments/corrections to the attached .pdf of the manuscript, and mention a few of these items here.

There are no major concerns with the publication of this manuscript providing these minor corrections made and/or considered to help substantiate the interpretation of the isotope data. The authors have been particularly thorough in their realistic evaluation of the morphometric determination using the existing software and databases available for this

purpose (e.g. Fordisc), which are not specific to archaeological populations. The additional information on this part of the study contained in the supplemental was much appreciated and necessary.

Finally, the balance between offering specific (likely) geographic origins for each of the individuals sampled with a recognition of equifinality inherent in individual, or combined, isotope systems ($\delta^{13}\text{C}$, $\delta^{15}\text{N}$, $\delta^{18}\text{O}$, $\delta^{34}\text{S}$, and $^{87}\text{Sr}/^{86}\text{Sr}$) is reasonably done. I have suggested statistical treatment of the isotope data should be attempted/worked through via appropriate non-parametric methods to help further support the author's suggestion of origins within or outside 'Britain'. Given the additional information available on each of the individuals that are part of this study, one could consider a possible bayesian approach to determining 'local' versus 'non-local' in this context.

We are grateful to the reviewer for highlighting the potential. We consulted a statistician at Cardiff University and they advised against tests of difference on such a small sample. The power of the dataset is too poor and it is certainly not standard practice to conduct tests of difference on such small samples. We have added the word 'deemed' to section 3.2 to show that this was our decision.

Overall I enjoyed the paper and how the authors have approached this research, and I look forward to seeing it published in its revised format.

We have made all the minor revisions suggested by reviewer 3 throughout the .pdf of the manuscript. Below are responses to comments which require more detailed replies.

Section 2.3 comments:

Reviewer 3: Just a method point here. Not sure why this particular approach was taken to remove additional Ca from the samples. One could simply include additional washes of 8M HNO₃ through the columns (after the sample was loaded) to remove Ca.

This is an interesting methodological point. From testing and general experience with Eichrom/Triskem organic bead resins, the effectivity of removal of complex matrix does not follow the Kd curves exactly. Particularly for removing major matrix, as in this case Ca, there is a saturation of Ca complexing and adding more 8N HNO₃ will not remove the remaining Ca. However, by drying samples down and reloading on new resin the remaining Ca can be nearly completely eluted. Sentence has been changed to:

Samples were placed on a hotplate (120°C) to dry overnight before this process was repeated for a second pass for the effective removal of any remaining traces of Ca.

Reviewer 3: Presumably SRM 1400 was also prepared in the same way as the enamel samples. If so, please state this. I would suggest that both SRM 987 and 1400 are being used

for accuracy checks here since neither has certified values and previously published values are being reported to help with data quality assurance.

Apologies – we should have clarified this. Yes, 1400 was processed through chemistry in the same way, This has now been made clear in the text. The measured SRM 987 as the primary standard cannot really be used as a test of accuracy but we agree with the reviewer that it may appear incorrect to say we use the 1400 for accuracy check. What we actually are doing is to check the accuracy (as to within published values) and robustness of our sample processing and normalisation to the measured NBS987 primary standard. We have now changed the sentence to reflect this:

Accuracy of the NIST SRM 987 normalisation and the chemistry processing was assessed by repeat measurements of $^{87}\text{Sr}/^{86}\text{Sr}$ ratio in NIST SRM 1400 (Bone Ash, processed through chemistry similar to the unknown samples), giving an average $^{87}\text{Sr}/^{86}\text{Sr}$ ratio of 0.713111 ± 0.000014 ($2\sigma_N$, $n=5$), which is consistent with the published value (0.713126 ± 0.000017 [80]).

Reviewer 3: No pretreatment of the enamel was carried out prior to acid digestion on the Kiel device? Typically, some form of pretreatment is carried out when analysing carbonate in teeth. If this is argued to not be necessary, then appropriate studies/literature should be included here to indicate this.

This is another interesting methodological point and there has been much debate about carbonate pre-treatments. We opted for no pre-treatment to remain in line with Chenery et al 2012 (ref 83), they cleaned their material with DI only. In addition, changes to isotope composition due to pretreatment can be large and variable (Pellegrini and Snoek 2016). In our view we could reduce potentially altering affects by retaining samples untreated material. Pellegrini and Snoek (2016) is now cited in the paper and we mention the potential altering effects of pre-treatment.

Reviewer 3: Were the errors associated with each step of this process ($d18\text{O}$ measurement, conversion from VPDB to VSMOW, and then from $d18\text{Ocarb}$ to $d18\text{Ophos}$) propagated and reported?

Thank you for highlighting this omission. We have added more information about associated error, including the standard deviation of replicate measurements and the error related to Chenery's regression conversion:

The standard deviation of replicate $\delta18\text{O}$ measurements is 0.049‰. The $\delta18\text{Ocarbonate}$ values were converted to $\delta18\text{Ophosphate}$ ($\delta18\text{Op}$) values (following [89]) to allow for comparison with other datasets. The error on the calculated $\delta18\text{Op}$ values is 0.24‰, based on the analytical error and the error in the conversion regression equation [89].

We have also added extended oxygen data in the supplementary material.

Section 2.4 comments:

Reviewer 3: Given the vagaries associated with %N and %C determinations on an EA, I would expect a calibration of materials with different % level contents (i.e. that span the range of values expected in collagen at the very least) to be used in these determinations.

To address this, we have changed the sentence to:

The 1 σ (n=54) reproducibility was ± 0.06 for $\delta^{13}\text{C}$ and ± 0.07 for $\delta^{15}\text{N}$. Different weights of caffeine were analysed to establish a calibration equation for the abundance of C and N against signal intensity in the mass spectrometer, which was used to calculate %N and %C in actual samples. The content of caffeine (28.85%N, 49.48%C) was calculated from its chemical formula ($\text{C}_8\text{H}_{10}\text{N}_4\text{O}_2$).

Reviewer 3: While this comment doesn't require the authors to re-do their analyses, I would suggest for future isotope related analyses they incorporate both calibration and check standards into their analytical $d^{13}\text{C}$, $d^{15}\text{N}$, and concentration determinations. Additionally, calculation of the total uncertainty in their measurement can follow the recommendations and procedure in Szpak et al. (2017) "Best practices for calibrating and reporting stable isotope measurements in archaeology." JAS:R 13:609-616.

Thank you for this comment. We have made the additions above and believe that this essentially follows what the Szpak paper recommends.

Section 3.1 comment:

Reviewer 3: The mean-median difference of 0.3 per mil is really not that significant when the errors associated with measurement of $d^{18}\text{O}_{\text{carb}}$, and the value conversions from VPDB to VSMOW scales, and then to $d^{18}\text{O}_{\text{p}}$ are factored in.

We agree that this is not a marked difference. However, any errors are not associated with the conversion from VPDB to VSMOW, as it's purely a numerical conversion between two different scales; the errors are related to measurement of $^{18}\text{O}_{\text{c}}$ and the conversion to $^{18}\text{O}_{\text{p}}$ and these are reported in the cited literature.

Section 3.2 comments:

Reviewer 3: How were 'outliers' determined? Was this through simply looking at the data, or where statistical tests or evaluations of the dataset carried out?

These outliers were identified through basic observation, as statistical testing was considered inappropriate for such a small dataset.

Reviewer 3: There are appropriate statistical tests that can be performed on small datasets such as these, accounting for non-normal distributions and using post-hoc analyses to mitigate the smaller sample size and distribution. Some examples include Wilcoxon rank and Kruskal-Wallis, for starters.

We are grateful to the reviewer for highlighting the potential. We consulted a statistician at Cardiff University and they advised against test of difference on such a small sample. The power of the dataset is too poor and it is certainly not standard practice to conduct tests of difference on such small samples. We have added the word 'deemed' to section 3.2 to show that this was our decision.

Section 4 comment:

Reviewer 3: I'm sure the authors are aware high d_{34S} values (>14 per mil) would also indicated an origin consistent with living on limestone, which occurs throughout the UK. As such, d_{34S} becomes less useful for determining origins in this particular case.

We are aware of some evidence for a relationship between limestone lithology and sulphur isotope values. However, the data are fairly weak and inconsistent and therefore we do not think they can be relied on yet. Evans et al. (2018). do not map limestone regions as 14+. Zazzo et al. (2011) did not identify this pattern in their study of modern sheep and nor is it mentioned in Nehlich's (2015) review of sulphur isotope research. Guiry and Szpak (2020) and Zazzo et al. (2011) have demonstrated that seaspray (and thus coastal proximity) is the dominant variable in driving high sulphur values. Madgwick et al. 2019 (57) measure S in 6 plant samples from limestone areas and all have values between 6 and 9 (in line with the Evans et al. 2018 map for inland limestone zones). In short, the evidence for high values on limestone is limited.

Section 4.1 comment:

Reviewer 3: Since the baseline areas overlap in d_{18O_p} values, the authors should be more cautious in their interpretation of these data to fall within any specific geographic region in the UK. I would also recommend the authors make some attempt to determine whether or not the overlap (or lack there of) between these samples and the areas indicated (east versus west of Britain) actual have some statistical significance; perhaps even using estimate plots of the median (and distributions of the data) would be helpful here.

The text has been changed to be less firm in interpretation and caveat has been added again. Given the small sample size, we are reluctant to statistically test our data and rely on Evans et al. (2012) and Pellegrini et al. (2016) who both highlight that east and west Britain can be broadly differentiated. We acknowledge this difference is only sufficient for us to state that it is 'likely' that these individuals are from the west/south.